Resource

# Continental-scale associations of *Arabidopsis thaliana* phyllosphere members with host genotype and drought

Talia L. Karasov [1,2] ✉, Manuela Neumann [2,7], Laura Leventhal[3,4], Efthymia Symeonidi[1], Gautam Shirsekar[2,8], Aubrey Hawks[1], Grey Monroe [2,9], Pathodopsis Team*, Moisés Exposito-Alonso[3,4,10,11], Joy Bergelson[5], Detlef Weigel [2,6] ✉ & Rebecca Schwab[2]

Plants are colonized by distinct pathogenic and commensal microbiomes across different regions of the globe, but the factors driving their geographic variation are largely unknown. Here, using 16S ribosomal DNA and shotgun sequencing, we characterized the associations of the *Arabidopsis thaliana* leaf microbiome with host genetics and climate variables from 267 populations in the species' native range across Europe. Comparing the distribution of the 575 major bacterial amplicon variants (phylotypes), we discovered that microbiome composition in *A. thaliana* segregates along a latitudinal gradient. The latitudinal clines in microbiome composition are predicted by metrics of drought, but also by the spatial genetics of the host. To validate the relative effects of drought and host genotype we conducted a common garden field study, finding 10% of the core bacteria to be affected directly by drought and 20% to be affected by host genetic associations with drought. These data provide a valuable resource for the plant microbiome field, with the identified associations suggesting that drought can directly and indirectly shape genetic variation in *A. thaliana* via the leaf microbiome.

The widely different environments in which the cosmopolitan species *Arabidopsis thaliana* is found today[1] have left strong signatures of selection throughout its genome[2]. While geographic differences in abiotic factors are well appreciated, similar differences in the resident microbiota are also likely to influence local plant fitness[3]. A recent survey of *A. thaliana* root microbiomes[4] found regional differentiation, often reflecting the composition of the soil microbiota. Host location was similarly significantly correlated with both

root- and leaf-associated microbial composition of another crucifer, *Boechera stricta*[5].

We already know that host genetics can influence microbiome composition[5-8], and geographic differences in host genetics may in turn structure the resident microbiome, but the two might also be independently affected by physical distance, including abiotic factors that vary geographically[4,5]. For example, pH is a significant predictor of bacteria in the *A. thaliana* rhizosphere[4], consistent with pH as a major

[1]School of Biological Sciences, University of Utah, Salt Lake City, UT, USA. [2]Department of Molecular Biology, Max Planck Institute for Biology Tübingen, Tübingen, Germany. [3]Department of Biology, Stanford University, Stanford, CA, USA. [4]Department of Plant Biology, Carnegie Institution for Plant Science, Stanford, CA, USA. [5]Center for Genomics and Systems Biology, Department of Biology, New York University, New York, NY, USA. [6]Institute for Bioinformatics and Medical Informatics, University of Tübingen, Tübingen, Germany. [7]Present address: Robert Bosch GmbH, Renningen, Germany. [8]Present address: Department of Entomology and Plant Pathology, Institute of Agriculture, University of Tennessee, Knoxville, TN, USA. [9]Present address: Department of Plant Sciences, University of California Davis, Davis, CA, USA. [10]Present address: Department of Integrative Biology, University of California Berkeley, Berkeley, CA, USA. [11]Present address: Howard Hughes Medical Institute, University of California Berkeley, Berkeley, CA, USA. *A list of authors and their affiliations appears at the end of the paper. ✉e-mail: t.karasov@utah.edu; weigel@tue.mpg.de

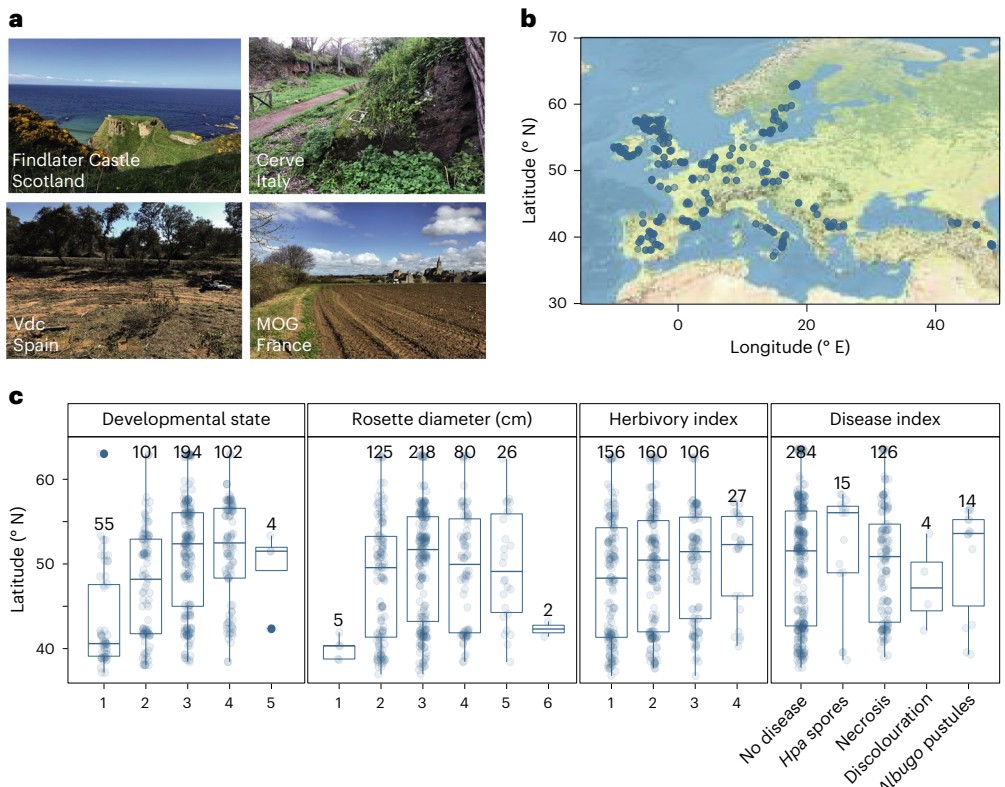

**Fig. 1 | Representative sampling of *A. thaliana* phyllosphere microbiomes across Europe. a,b**, *A. thaliana* plants were collected from distinct ecosystems. **a**, Examples of aspects of collection locations. **b**, Latitude/longitude of all locations. MOG is an acronym for Moguériec, France, and Vdc for Villaviciosa de Córdoba, Spain. **c**, Based on images of individual plants taken at each site, we assessed plant health and development. The *x* axis represents qualitative values (Methods), except for the rosette diameter, which is classified in intervals of (1) 0–1 cm, (2) 1–2 cm and so on. The disease index corresponds to different macroscopic disease symptoms as indicated (*Hpa, Hyaloperonospora arabidopsidis*). The central horizontal line in each box indicates the median, the bounds indicate the upper and lower quartiles and the number above the boxes indicates the individuals in each group.

driver of soil bacterial communities[9]. Similarly, precipitation can be a significant predictor of plant microbiome composition[10].

Because previous studies have typically been limited in the number of populations[4] or the geographic range surveyed[3], it has been difficult to disentangle the effects of host genetics, geography and abiotic factors on the plant-associated microbiome. In this Resource, we use a continental-scale assessment of bacteria that colonize *A. thaliana* leaves to identify environmental and host genetic factors that are strongly associated with distinct microbiome types. We then determine the environmental variables that best predict microbiome composition. Finally, we follow up with a controlled field experiment to test the relative contributions of host genetics and of water availability to these predictable patterns and a direct demonstration that a common bacterial taxon can provide drought protection. Our results indicate that differential plant survival in low-water environments might in part be due to different bacteria colonizing drought-adapted and drought-susceptible plants.

## Results

From February to May 2018, we visited 267 European *A. thaliana* populations around the end of their vegetative growth and close to the onset of flowering[11] (Fig. 1a,b). At each site we collected whole rosettes from two individuals, along with a neighbouring crucifer (family Brassicaceae, primarily *Capsella bursa-pastoris*), if present, and two soil samples. We evaluated *A. thaliana* life history traits (Fig. 1c and Extended Data Fig. 1) and extracted information on climate variables for the collection sites[12]. We assessed the microbial composition of the leaf and soil samples by sequencing the V3–V4 region of the 16S ribosomal RNA locus and identifying amplicon sequence variants (ASV) using DADA[13]. Each ASV was considered a distinct bacterial lineage or phylotype. Host genetics and absolute microbe abundance were assessed by shotgun sequencing plant tissue, which generates reads of host and microbial genomes[14].

### Phyllosphere composition is distinct from the soil and is host species specific

There is considerable debate as to the origin of the microbes that colonize plants, although soil often has a measurable influence[4,15,16]. A study across 17 European *A. thaliana* populations[4] found differentiation between root and non-root-associated microbes, but no significant differences between *A. thaliana* and neighbouring grasses[4]. Intraspecies comparisons in a common garden experiment had suggested that host genetics can explain about 10% of the variance among *A. thaliana* leaf bacteria[17]. At the basis of these comparisons is the question of how much the host influences microbiome assembly, either because of active recruitment of specific microbes, or because of the differential ability of microbes to colonize their hosts.

To explicitly test for enrichment of specific taxa in the phyllosphere, we compared soil and plant leaves across all 267 sites via multi-dimensional scaling (MDS; Hellinger transformation). As expected, there was broad-scale separation between the phyllosphere and the soil (Fig. 2a,b). Modelling[18] the effect of compartment on the microbial core phylotypes in the phyllosphere revealed differential abundance of 91% (524/575) of phylotypes between the *A. thaliana* phyllosphere and soil (False Discovery Rate (FDR) <0.01). Focusing on differences among host species[18], we found 36% (205/575) of phylotypes

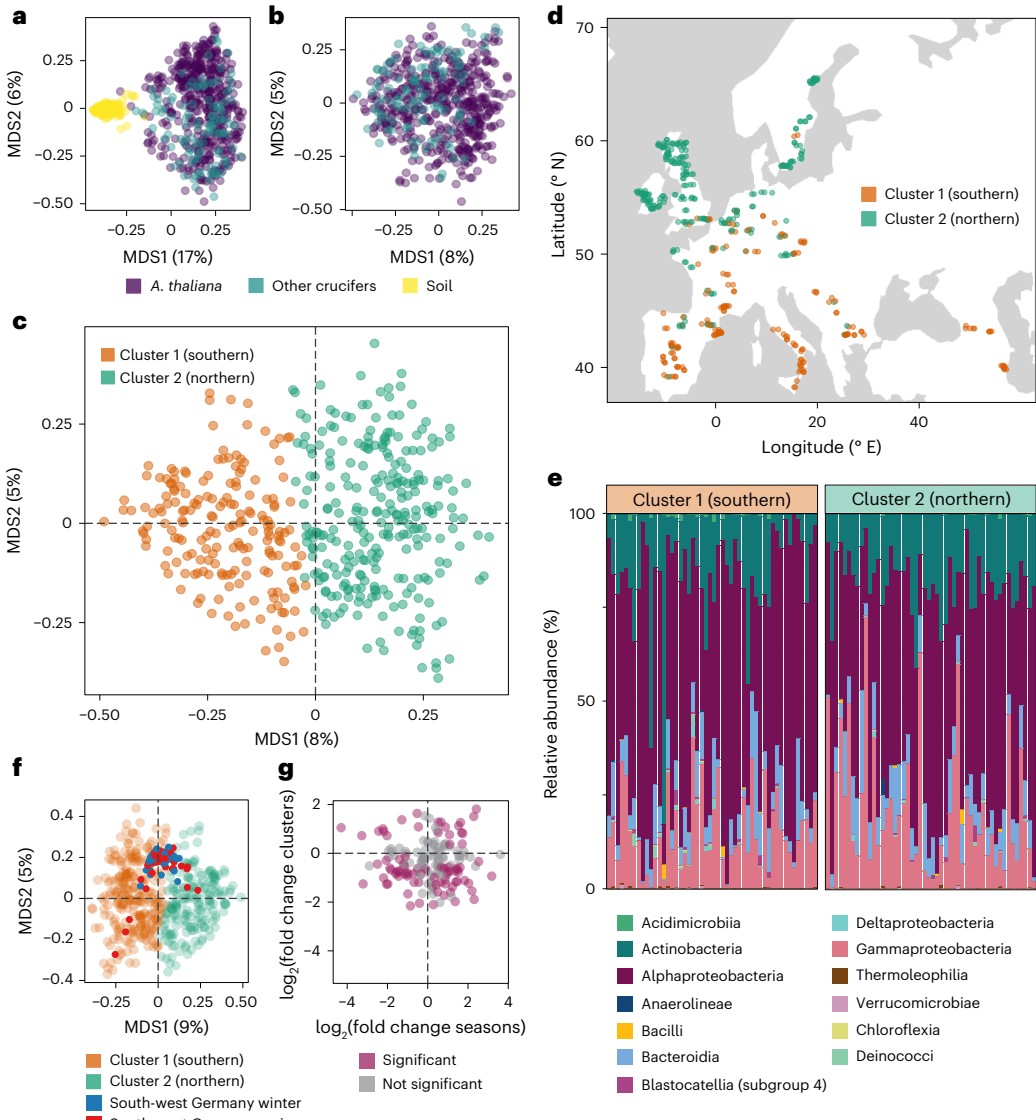

**Fig. 2 | Two distinct microbiome types in *A. thaliana* along a latitudinal cline. a,b**, Ordination on a Hellinger transformation of the samples. *Arabidopsis thaliana* leaf microbiomes are significantly differentiated from that of surrounding soil (**a**) and less so, but still significantly, from surrounding crucifers (Brassicaceae) (**b**). **c,d**, *k*-means clustering (*k* = 2) (**c**) identified two microbiome types that turned out to have a north–south latitudinal cline (**d**). **e**, Distribution of higher taxonomic levels across the southern and northern clusters.

**f**, Comparison of extent of seasonal variation in south-west Germany (winter and spring) with the European geographic variation (clusters 1 and 2). **g**, Absence of correlation in fold changes (FCs) in phylotype abundance between the northern and southern clusters (*y* axis) and between the winter and spring samples from south-western Germany (*x* axis). Colour indicates association with the two north–south clusters 1 and 2.

to distinguish *A. thaliana* from neighbouring crucifers (Extended Data Fig. 2). This indicates that inter-host species differences in genetics or phenology have a strong influence on microbiome composition. On a phylotype-by-phylotype basis, abundance in *A. thaliana* was poorly predicted by a phylotype's abundance in soil or in the surrounding companion plants (Extended Data Fig. 2).

**Phyllosphere microbial composition varies with latitude**

We tested the geographic differentiation of microbiomes using dimensionality reduction for the entire community and assessment of the spatial distribution for each bacterial phylotype. The former reveals global trends in composition, while the latter provides information on individual microbes contributing to such trends. Loadings on both the first and second principal coordinate axes (Fig. 2c) correlated with latitude (Pearson's *r* = 0.75, *P* = 2.2 × 10$^{-16}$, and *r* = −0.24, *P* = 1.35 × 10$^{-7}$, respectively), suggesting geographic structure in the phyllosphere

microbiome. Because silhouette scoring[19] indicated that *A. thaliana* phyllosphere microbiomes were best characterized as two distinct types, we used *k*-means clustering of the Hellinger-transformed counts table to classify our samples (Fig. 2c and Extended Data Fig. 3). We found that the two microbiome types were strongly differentiated by geography, with one dominating in Northern and the other in Southern Europe (Fig. 2d,e). Among individual phylotypes, the relative abundance of one third (33%) was significantly associated with latitude (linear regression, FDR <0. 01), but only a small minority, 2%, was correlated with longitude, confirming that Northern and Southern European *A. thaliana* reproducibly harbour different microbiota. One percent of the plant-associated phylotypes were also significantly correlated in the soil with latitude, suggesting that the latitudinal contrast is formed via colonization.

The phyllosphere changes with plant development and the seasons[20]. To test whether the observed latitudinal phyllosphere

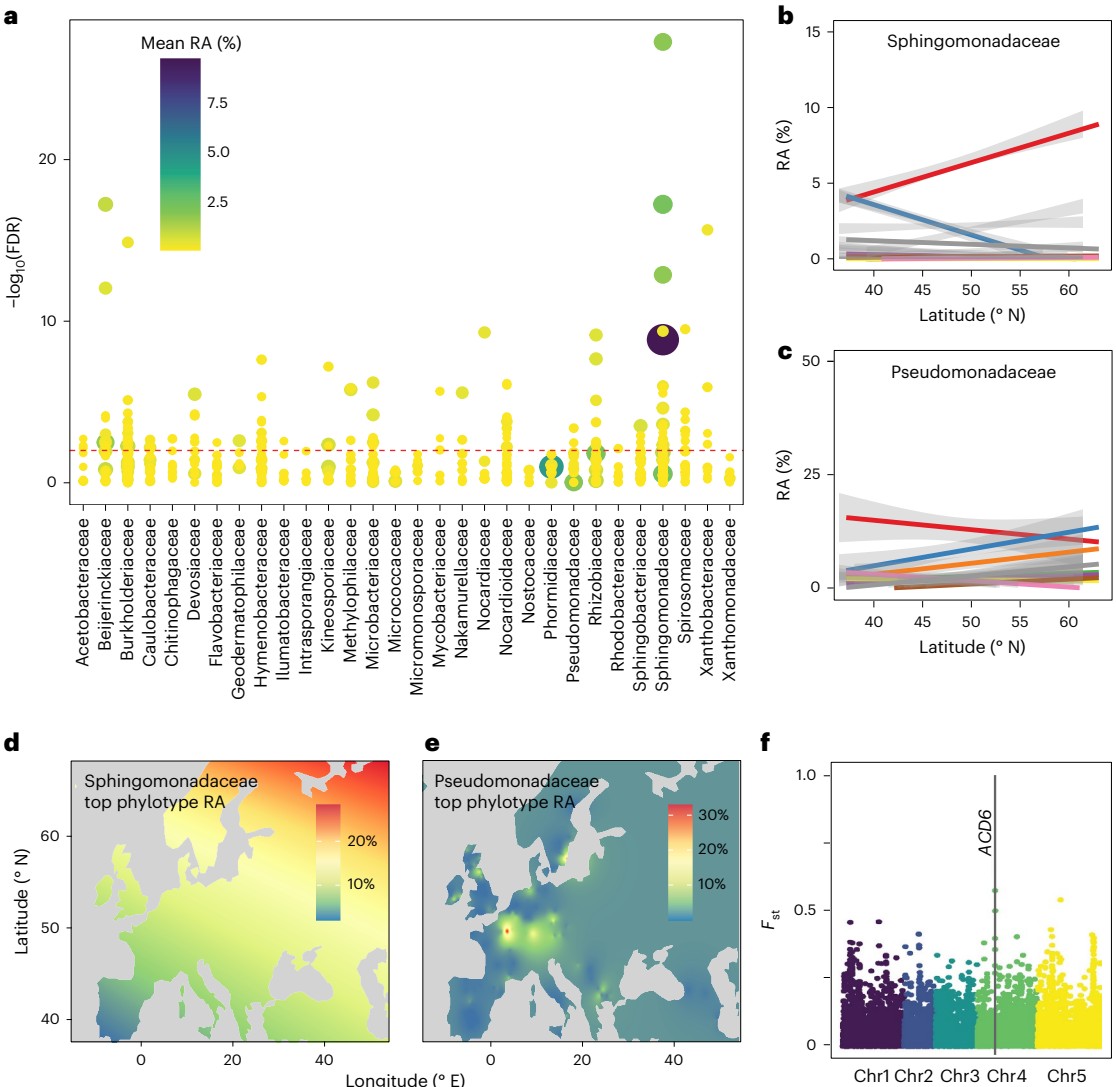

**Fig. 3 | Latitudinal clines in microbial abundances and association of a host immune gene with microbiome type. a**, Linear relationships between relative abundance (RA) of the most common phylotypes. The *y* axis represents $-\log_{10}$-transformed FDR-corrected *P* values obtained when regressing the abundance of a phylotype on latitude (linear regression). Phylotypes are grouped by families, which are indicated on the bottom. **b,c**, There is a strong latitudinal cline for the RA of the most abundant sphingomonads (**b**) but not for the most abundant pseudomonads (**c**; note the difference in RA scale). **d,e**, Interpolation of the abundance of the top sphingomonad phylotype (**d**) and of ATUE5 (**e**), the top pseudomonad phylotype and a known opportunistic pathogen, revealed a continuous spatial gradient for the top sphingomonad (**d**), but a patchy distribution with regional hotspots for the top pseudomonad (**e**). **f**, The relationship between microbiome type and polymorphism in plant immune genes was assessed with the $F_{st}$ population differentiation index. The most extreme $F_{st}$ values were found in the immune regulator *ACD6*. Data in **b** and **c** are presented as the estimated regression value ± s.e.m. Chr, chromosome.

contrast could be explained by seasonal and developmental differences, we compared our samples with a multi-year dataset from a single location in Germany[21]. Projecting seasonal phylotype composition into the MDS biplots of our pan-European samples did not reveal any preferential association of collection season with microbiome type (Fig. 2f). Comparing changes in the abundance of single phylotypes between seasons and between the two major microbiome types (Fig. 2g) similarly did not point to the latitudinal contrast reflecting environmental variation being caused by local seasonal differences (Wald test of multinomial frequency estimates, *P* > 0. 01).

The association between latitude and phylotype abundance was phylotype specific, differing within and between bacterial families (Fig. 3a and Extended Data Fig. 3). *Pseudomonas* and *Sphingomonas* are abundant across *A. thaliana* populations[21–23] and both genera can affect *A. thaliana* health[21,24,25]. Linear regression of each core phylotype onto latitude revealed that four of the five most abundant sphingomonads have latitudinal clines (Fig. 3a,b, FDR <0. 01), while the most

abundant pseudomonad phylotypes did not show long-distance variation (Fig. 3b–e). Rhizobiaceae were also latitudinally differentiated. A consequence of phylotype-specific association with latitude was that the two major microbiome types were significantly differentiated at the phylotype level, but not at higher taxonomic levels (Fig. 2e and Extended Data Fig. 3). Thus, even though *A. thaliana* is colonized by different individual phylotypes in Northern and Southern Europe, the bacterial classes remain broadly the same (Fig. 2e).

**Common phylotypes differ in their geographic distributions**

A single *Pseudomonas* phylotype, ATUE5 (previously OTU5), is a common opportunistic pathogen in local populations in south-west Germany, where it is an important driver of total microbial load[21]. Because ATUE5 was also the most abundant pseudomonad in our study, we wanted to learn how its distribution was geographically structured (Fig. 3c). ATUE5 was the seventh most common phyllosphere phylotype overall, with a relative abundance of up to 64% (mean of 1.8%).

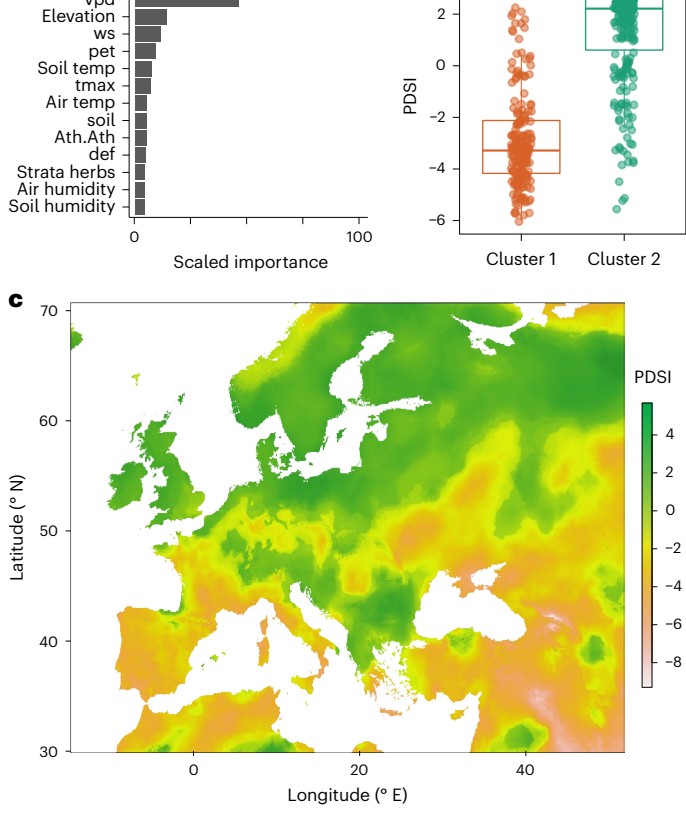

**Fig. 4 | PDSI is the best predictor of phyllosphere microbiome type. a,** Random forest modelling was used to determine environmental variables associated with microbiome type. The abbreviations are explained in Methods. **b,** PDSI of the location was the best predictor of microbiome type, explaining more than 50% of the variance. The upper and lower hinges of the boxes represent the first and third quartiles and the central line the median, with $n = 269$ plants in cluster 1 and $n = 192$ plants in cluster 2. **c,** The mean PDSI throughout Europe for January to April 2018.

ATUE5 was found in 56% of samples, but without significant latitudinal differentiation (Pearson's $r = 0.01$, $P = 0.92$).

Despite ATUE5 being a common phyllosphere member, its distribution was disjoint, and ordinary Kriging interpolation across the sampled range confirmed a very patchy presence (Fig. 3c). In contrast, the most frequent *Sphingomonas* phylotype (and most frequent phylotype overall) showed a significant latitudinal cline (Fig. 3b). High ATUE5 abundance was largely limited to single populations or populations very close to each other, with a spatial autocorrelation restricted to distances of under 50 km (Extended Data Fig. 6). In summary, the *Pseudomonas* pathogen ATUE5 is widely yet very unevenly distributed.

### Drought metrics predict microbiome composition
Common garden experiments have indicated that environmental factors strongly shape bacterial microbiome composition[17]. Our continental-scale data enabled us to test which abiotic factors are most correlated with geographic structure of the phyllosphere microbiome.

We tested for associations between climate variables and microbiome composition, including developmental and health traits as potential confounders[26]. Altogether, we considered 39 covariates that could influence microbiome composition (Extended Data Fig. 7 and Extended Data Table 1). We first removed covariates that were highly correlated with others and then performed random forest classification using the two microbiome types as response variables (Fig. 4

and Extended Data Fig. 8). The covariate with greatest explanatory power was the Palmer Drought Severity Index (PDSI) mean from the six pre-collection months, a metric of recent dryness[27]. PDSI was similarly the best predictor for the loading of a sample on MDS1. In general, environmental covariates were better predictors than were plant traits. In contrast, environmental covariates (including PDS1) had poor predictive power for plant-associated phylotypes in the soil microbiome, explaining less than 1% of the variance in the loading on the first principal coordinate axis.

Because PDSI is correlated with latitude, we tested whether information about both variables improves prediction outcomes. Inclusion of PDSI significantly improved predictive capacity ($P = 4.2 \times 10^{-7}$ for logistic regression with microbiome type and $P = 2.7 \times 10^{-7}$ for linear regression on MDS1), indicating that the association between microbiome type and PDSI extends beyond latitudinal correlation. PDSI was also predictive for microbiome composition within geographic regions and their corresponding sampling tours ($P = 2.3 \times 10^{-7}$ for logistic regression with cluster identity and $P = 0.047$ for linear regression on MDS1).

From mixed-effects modelling, we estimated the marginal $R^2$ for PDSI to be 50%. Together with previous work supporting the importance of water availability in determining host-associated microbiomes[9], we conclude that water availability affects which microbes can access the host plant and/or proliferate on the host. Drought might do so directly by affecting plant physiology, indirectly by shaping host genetics or by a combination of the two. Additionally, drought affects the abundances of microbes in the abiotic environment, and hence which microbes are present for colonization.

### Host genetics is associated with microbiome composition
*Arabidopsis thaliana* exhibits strong population structure across Europe, with a pattern of isolation by distance[28] and greater latitudinal than longitudinal differentiation[1]. Climate-driven selective pressures, particularly water availability and drought[29], along with different groups of insect predators[30] have contributed to the geographic structure of *A. thaliana* genetic diversity.

To determine whether this extends to the phyllosphere microbiome, we extracted heritability estimates for phyllosphere phylotypes from eight common garden experiments in which 200 *A. thaliana* accessions had been grown in four Swedish locations across 2 years[8]. Two thirds (368/575; 64%) of our core phylotypes had been observed in this study[8]. We were able to obtain heritability estimates for 251 of these phylotypes, almost all of which (247; 98.4%) had significant positive heritability in at least one of the eight experiments. Genetic differences are therefore very likely to contribute to the observed geographic differentiation of the *A. thaliana* phyllosphere microbiome across Europe. However, heritability does not necessarily imply direct host control of each phylotype, as it can also be exerted indirectly via microbial hub taxa[8].

To determine how microbiome composition in our study might be influenced by host genetics, which was representative of previous surveys[1] (Extended Data Fig. 4), we fitted a mixed-effects model that included relatedness as a random effect and the loading on the first axis of the decomposition of the microbiome composition as the phenotypic response variable. Plant genotype alone explains 68% of the variance in the loading along MDS1 and 52% of the variance in the MDS2 loading (pseudo $h^2$ 0.68, standard error of the mean (s.e.m.) 0.10 for MDS1 and pseudo $h^2$ 0.52, s.e.m. 0.12 for MDS2). MDS1 explains 8% and MDS2 5% of the variance in microbiome composition, consistent with host genetics probably playing only a subordinate role in structuring the microbiome[8,17,31]. In a mixed-effects model, PDSI was associated with MDS1, whereas several genetic principal components were associated with MDS2 (Extended Data Tables 2–4).

Because immune genes are prime targets for interactions with microbes[32,33], we tested whether specific immune gene alleles are

associated with the two microbiome types. Among a generous, though not exhaustive, list of 1,103 genes with connection to pathogen response and defense[34], the top single-nucleotide polymorphism (SNP) was in *ACD6* (empirical $P = 0.0001$) (Fig. 3f and Extended Data Fig. 5). *ACD6* alleles can differentially impact pathogen resistance through constitutive effects on immunity[35]. The full *ACD6* haplotypes associated with each microbiome type have not yet been reconstructed, as the short reads used for genotypic comparisons did not allow for resolution of full-length alleles. Nonetheless, our results demonstrate a striking association between microbiome type and polymorphisms in a central regulator of immune activation. Whether resident microbiota select for *ACD6* allele type, or instead *ACD6* allele type influences microbiome type, remains to be determined.

Are genetic alleles responsible for microbiome variation across geography? For defense genes such as *R* genes, this is probably not the case as variation tends to be maintained within local populations of *A. thaliana*[36,37]. We do not know whether this extends to genes that control the non-pathogenic microbiota. A previous study found ~150 SNPs to be significantly associated with heritable microbiome composition in *A. thaliana*[31]. When we tested the geographic differentiation of these SNPs across Europe (Extended Data Fig. 5), we found that they had significantly higher global $F_{st}$ values than the genome-wide background, consistent with different *A. thaliana* populations selecting for different microbiota.

### Host adaptation to drought influences microbial abundance

To disentangle the impact of drought from that of plant genetics, we conducted a common garden field experiment in California. Using a setup similar to our previous work in Europe[29], we grew *A. thaliana* accessions (Extended Data Table 5) under a high- and low-watering regimen. Focusing on accessions that had previously been identified as drought adapted or susceptible based on genetic loci associated with adaptation to drought[29], we assessed differences in phyllosphere composition after drought stress. Of the 575 core phylotypes in the European field collections, 154 were present in California and 20 were sufficiently common to enable us to determine the relative influences of genetics and drought treatment on their relative abundances (Extended Data Tables 2–4). Of these 20 phylotypes, 3 were significantly influenced by host genetic classification of drought-adapted versus susceptible accessions, and 3/20 showed a significant interaction between drought treatment and host genotype (Extended Data Table 6). Two out of 20 showed a significant response to the abiotic drought treatment alone. The phylotypes that were significantly associated with plant genotype in the California field experiment accounted for an appreciable fraction of the total microbiome in the European wild collections—an average of 13.2% of the total microbial community in a plant and as high as 71.9% total relative abundance in a plant (Extended Data Fig. 9). The most abundant phylotype across the European collection (Extended Data Fig. 9) was significantly associated with plant genotypic classification. In total, these results indicate that genetic adaptation to drought has an impact on some of the most abundant bacteria that colonize a plant.

### Common phylotypes alter drought effects on *A. thaliana*

Finally, we tested whether water availability can influence the abundance of a common phylotype, the opportunistic pathogen ATUE5. In growth chambers, we exposed 5-week-old plants of the Col-0 reference accession to a week-long drought, followed by syringe inoculation with the ATUE5 p25.c2 strain[21]. Three days after infection, we compared bacterial growth and green tissue in drought-stressed and well-watered plants. Drought significantly reduced the ability of ATUE5 to proliferate in planta (Extended Data Fig. 10; two-sided Wilcoxon rank-sum test, $P = 0.003$), a result consistent with *Pseudomonas* pathogens relying on water availability to spread and multiply[38]. Drought also significantly reduced the green, photosynthetically active leaf area (Extended Data Fig. 10), with ATUE5 infection blunting this negative effect of drought.

These results indicate that infection by an opportunistic pathogen may be conditionally beneficial, conferring drought tolerance under specific conditions. ATUE5 was previously shown to influence *A. thaliana* growth in a genotype-specific manner[39], indicating that the interaction between drought and ATUE5 infection is likely to differ between plant populations. This is reminiscent of viral infection reducing drought-based mortality[40] and in agreement with plant growth promoting effects of microbes under drought[41], as discussed in a recent review[42] of the diverse mechanisms of microbe-mediated drought tolerance. Moreover, there is precedence for cryptic *A. thaliana* pathogens providing environment-specific fitness benefits[43].

## Discussion

Our results reveal several robust trends. Firstly, colonization of *A. thaliana* leaves imposes a strong bottleneck on the microbes that arrive from the surrounding soil and other plants, with most microbes differing in abundance between the soil and *A. thaliana* leaves and more than a quarter differing between *A. thaliana* and companion plants from the same family. Host genetics clearly matters for determining which microbes manage to establish in and on the plant. Our results indicate that these trends, observed before over small regions[4,7,8], are reproducible and ubiquitous on a continental scale. Secondly, geography and associated abiotic factors significantly influence the microbes on *A. thaliana*: a plant in Spain will very probably be colonized by a different suite of microbes than a plant in Sweden. Our field experiment begins to disentangle the direct contribution of geography-dependent climate differences on the microbiome from those that are mediated by adaptive differences in host genetics. We note, however, that both genetic population structure and environmental variables exhibit autocorrelation, hence the variance explained by plant genotype is invariably confounded by correlated environmental factors, with the exact extent being difficult to discern. We identify genetic variation in an immunity gene, *ACD6*, to be associated with microbiome type and with PDSI. Specific alleles of *ACD6* confer drought tolerance[44], adding further complexity to our understanding of the relationship between drought, microbes and plant genetics. Lastly, our analyses suggest that microbial colonization of plants is strongly dictated by water availability and the attendant microbiota. This again raises the question of how different microbial communities influence plant phenotype. Drought not only plays a major selective role in *A. thaliana* populations[29], but it is also known to affect the ability of plants to withstand pathogen attack. An important question will be whether different background microbiomes in plants that are more likely to experience drought in the wild will help or hamper defense against pathogens[45].

## Methods

### Sample collection

*Arabidopsis thaliana* and other crucifers were sampled during local springtime in 2018. Most crucifer companion samples were *Capsella bursa-pastoris*, and the rest were *Cardamine hirsuta*. A full list of sampling locations and dates is provided in Extended Data Table 1. Rosettes were separated from the roots using alcohol wipe-sterilized scissors and forceps, then washed with water and ground with a sharp disposable spatula (Roth) in RNAlater (Sigma, now Thermo Fisher). For each *A. thaliana* plant for which soil was accessible, one to three tablespoons of soil were collected from the location where the plant had been removed and placed in a clean airtight bag. Samples were then maintained in electrical coolers (Severin Kühlbox KB2922) until the end of the sampling trip (which were 1–12 days long). In the lab, samples were stored at 4 °C. Within 0–3 days, RNAlater was removed from plant samples. Samples were centrifuged for 1 min at 1,000 g, the supernatant was removed and samples were washed with 1 ml autoclaved water. For storage at −80 °C, plant tissue was transferred with ethanol sterilized forceps to screw cap freezer tubes containing 1.0 mm Garnet Sharp Particles (BioSpec Products, Cat. No. 11079110GAR). A ~200 mg aliquot

from each soil sample was transferred to a screw cap freezer tube using an ethanol sterilized spatula, with great effort to exclude plant and insect pieces. Before aliquoting, soil bags were kept at −80 °C and defrosted at 4 °C overnight, unless aliquoting was done immediately upon arrival in the lab at the end of the sampling trip.

## Nagoya Protocol Compliance

Respective national authorities of all sampled countries that are party to the Nagoya Protocol were contacted. Where needed, advised measures were taken and resulted in sampling and export permits: KC3M-160/11. 04. 2018 (Bulgaria), ABSCH-IRCC-FR-253846-1 (France) and ABSCH-IRCC-ES-259169-1 (Spain).

## Plant phenotyping

Scores presented in Fig. 1 and Extended Data Fig. 1 are

- Developmental state: vegetative (1), just bolting (2), flowering (3), mature (4) and drying (5)
- Herbivory index: no (1), weak (2), strong (3) and very strong (4) herbivory
- For rosette diameter, a 1 cm rosette diameter classification corresponds to any rosette diameter ≤1 cm.

## DNA extraction

DNA was extracted from plant samples according to the protocol from ref. 21. Soil DNA was extracted using Qiagen MagAttract PowerSoil DNA EP Kit (384) (cat. 27100-4-EP). On dry ice, soil samples were transferred from tubes to PowerBead DNA plates using sterile individual funnels. Plates were stored up to 2 weeks at −80 °C until processing. The Qiagen protocol was adapted to a 96-well-pipette (Integra Viaflo96). Power-Bead solution and SL Solution were pre-warmed at 55–60 °C to avoid precipitation. RNase A was added to the PowerBead solution just before use. From step 17 of the protocol, instead of starting epMotion protocol, the following steps were performed: to each well of the 2 ml deep-well plate containing maximum 850 µl of supernatant, 750 µl of Bead Solution was added and mixed with Eppendorf MixMate at 650 rpm for 10–20 min. Plates were placed on a magnet for 5 min, the supernatant solution discarded and the beads washed three times with 500 µl wash solution. Beads were eluted with 100 µl elution buffer. The eluate was transferred to PCR plates and stored at −20 °C until library preparation.

## Drought treatment with infection

Plants of the *A. thaliana* Col-0 reference accession were grown for 35 days at 23 °C under short day conditions (8 h light:16 h dark) with normal watering (approximately 1 l water per tray once soil moisture dropped below a reading of 3; XLUX Soil Moisture Meter). At 35 days, plants were randomized into new trays and watering treatments started. Soil moisture was measured every day. Control plants were watered normally once the soil moisture readings were between 2 and 3. Drought-stressed trays were dried down to an average soil moisture reading of 1, kept ≤1 for a full day, then maintained between a reading of 1 and 2 with minimal watering. The plants were exposed to these contrasting water conditions for seven days before infection. On day 7, control trays were watered normally (until soil moisture averaged a reading of 5–6 per tray) and drought trays were watered at 0.4× normal water per tray (reaching an average soil moisture reading of 2–3). After having been watered, two leaves per plant were syringe-infiltrated with either $MgSO_4$ (control) or ATUE5 p25.c2 at an $OD_{600}$ of 0.0002. Each treatment had approximately 96 plants, divided over four trays. Plants were photographed every other day, starting at 35 days after planting. Plant growth and health were estimated by measuring green pixel area per plant using plantCV[46] (Supplementary Data Table 1). At 3 days after infection, hole punches were taken from two leaves per plant, ground and resuspended in dilutions 10 mM $MgSO_4$. Colonies were counted after 2–3 days of growth on selective lysogeny broth agar plates with

100 µg ml$^{-1}$ nitrofurantoin to select for *Pseudomonas* (Supplementary Data Table 2). No statistical methods were used to pre-determine sample sizes but sample sizes are similar to or greater than those reported in previous publications[47].

## Field experiment

**Accessions.** A total of 110 *A. thaliana* accessions were planted in a common garden experiment with water manipulation in a common garden field site at the Carnegie Institution for Science (37.42857020996903° N, 122.17944689424299° W) in Stanford in the spring of 2023 (Extended Data Table 5). We selected two groups of accessions based on their predicted contrasts in ability to survive drought in two consecutive field experiments at two locations. Based on survival data under low watering in Spain[29], polygenic scores were trained on 515 accessions following state-of-the-art methods[48] using PLINK v2.00a2.3[49]. Conducting polygenic scores with different sets of SNPs (varying *P* value of their association with survival from $10^{-3}$ to $10^{-9}$), we verified a broad overlap of accessions in the top 30 and bottom 30 of the rank distribution. We utilized a threshold of 0.001 to select such 30 top and 30 bottom accessions. In a second round of experiments in California, a pilot study for the current work, polygenic scores were trained on total fitness (survival and fruit production) under drought conditions in 245 accessions. Polygenic score analyses used the software GEMMA and the Bayesian Sparse Linear Mixed Model[50]. This approach utilized genome-wide SNP information and their estimated parameters (probability of causal effect and the effect size) to make polygenic score predictions. We again selected 30 accessions with the highest and lowest polygenic scores. Finally, from the two polygenic score prediction rounds we identified 57 accessions with a high score in drought survival and 59 with a low score to conduct field experiments and microbiome analyses (3 and 1 accessions, respectively, did not have enough seeds for our experiment size). As there was some overlap in selected accessions from the first to the second year, only a total of 110 unique accessions were sown.

**Experiment.** We planted seeds from selected accessions in 464 individual, randomized pots on 16 November 2022 in a common garden field site at the Carnegie Institution for Science. Five to ten seeds were planted in each pot within a 60-pot tray with Nutrient Ag Solutions PROMIX PGX Biofungicide Plug & Germination mix. The trays were gently watered for 2 weeks until germinants were established. We thinned each pot to have a single plant, before imposing a high and low precipitation treatment. For the well-watered treatment, the plants received an additional 144 min of rainfall every 2 days from December 2022 to May 2023 (about 600 additional mm for the entire growing season) on top of the natural rainfall at this location. The drought treatment consisted of only natural rainfall, which in California typically leads to water stress and visible mortality of *A. thaliana* plants.

**Microbiome study.** On 5 April 2023, we collected two true leaves from every plant that had not begun to senesce or decay (386 plants in total). All tools were sterilized between plant sampling. Tubes with tissue were immediately submerged in liquid nitrogen and transferred to a −80 °C freezer.

## 16S rDNA ASV identification

Oligonucleotide primers targeting the consensus V3–V4 ribosomal DNA (rDNA) region from 341 bp (5′-CCTACGGGAGGCAGCAG-3′) to 806 bp (5′-GGACTACNVGGGTWTCTAAT-3′) were used to amplify 16S rDNA sequences with the protocol described in ref. 21. Briefly, amplification was achieved with a two-step PCR protocol in which 100 µM peptide nucleic acid was used in the initial PCR to block amplification of chloroplasts. Amplicons were sequenced on the MiSeq (Illumina) platform using the MiSeq Reagent Kit v3 (600 cycle). Samples with lower coverage were preferentially sequenced to greater depth in

subsequent runs in a total of four runs of the Miseq. Output from all runs was pooled for downstream analysis. Primer sequences were removed before analysis with a combination of usearch (version 11, ref. [51]) and custom bash scripting. The 16S rDNA sequences were quality trimmed using DADA2[13] (version 1.10.1). The forward read was truncated at position 260 and the reverse read at position 210 due to decreased quality of the second read. Reads were truncated when the quality score dropped to less than or equal to 2 (trunQ=2). Chimeras were removed with the removeBimeraDenovo function (method='consensus') and ASVs called de novo using DADA2. The resulting reads were then aligned using AlignSeqs from the DECIPHER package[52] (version 2.8.1). A phylogenetic tree of the de novo called ASVs was constructed using fasttreeMP[53] (version 2.1.11). Taxonomic assignment of reads was performed with comparisons of 16S rDNA sequences to the Silva database[54] (nr v132 training set).

Only samples with at least 1,000 reads after filtering for mitochondria and chloroplast reads were included. We began with 939 samples (including soil samples and neighbouring non-*A. thaliana* plants), in which we found 195,545 ASVs. A total of 918 samples had a sufficient number of reads (>1,000 reads) and after removing ASVs that were not found in any single sample with more than 50 reads, we were left with 10,566 ASVs. We identified a core set of 575 ASVs by filtering for those ASVs that were present in at least 5% of *A. thaliana* samples. The ASVs classified as belonging to the taxonomic class Cyanobacteria were removed from the dataset to eliminate possible misassignment of plant chloroplast DNA that can vary between plant genotypes and skew subsequent analyses.

For the Californian field experiment, we sequenced the 16S rDNA amplicons as above and processed ASVs with the same pipeline used for the European wild samples. In the Californian ASV table, we identified ASVs present in 10% or more of the samples, and merged these ASV identifiers with those of the European collections to call the intersection of observed ASVs.

## Climate variables

The majority of climate variables were obtained from Terraclimate[12] using the data for 2018 (http://www.climatologylab.org/terraclimate.html), a dataset with approximately 4 km spatial resolution. For random forest modelling and climate associations, we calculated the average value of each climate metric over the 6 months preceding the date of collection. The following variables were included in the random forest modelling from the Terraclimate dataset: tmax, maximum temperature; tmin, minimum temperature; vp, vapour pressure: ppt, precipitation accumulation; srad, downward surface shortwave radiation; ws, wind speed; pet, reference evapotranspiration (ASCE Penman–Montieth); *q*, runoff; aet, actual evapotranspiration; def, climate water deficit soil and soil moisture; swe, snow water equivalent; PDSI; and vpd, vapour pressure deficit.

We further analysed associations with Koeppen–Geiger climatic zones[55,56], which were inferred in R using the package kgc and the regional classifications from ref. 57. Initial assessments of the density of microbes throughout Europe were calculated via ordinary Kriging using the R package automap[58] (version 1.0-14). Four models were tested during variogram fitting, namely 'Sph', 'Exp', 'Gau' and 'Ste'. Interpolation was performed either on the abundance data untransformed or on $\log_{10}$-transformed values with 0. 0001 added to allow for zero counts to be included. Global information on the major vegetation types was obtained using the Globcover 2009 map (released December 2010) from the European Space Agency (http://due.esrin.esa.int/page_globcover.php). Measures of soil properties were obtained using the International Soil Reference and Information Centre (ISRIC, global gridded soil information) Soil Grids (https://soilgrids.org/#!/?layer=geonode:taxnwrb_250m).

At the time of collection we took several measurements of the soil and air temperature and humidity (Soil temp, Air temp, Soil hum and

Air hum), the surrounding plant community and the location type: distance between the focal and the closest neighbouring *A. thaliana* plant (Ath.Ath), distance between the focal and the closest other plant (Ath.other), immediate plant density (Ground cover), visible *H. arabidopsidis* infection on focal plant (Hpa plant) or at site (Hpa site), visible *Albugo* spp. infection on focal plant (Albugo tour), fraction of herbal plants in the surrounding (Strata herbs), and estimated sun exposure (Sun), slope (Slope) and ground humidity (Humidity ground). Measurements are listed and detailed in Extended Data Table 1.

## Feature selection and random forest modelling

Features of interest were first identified by feature selection in the R package caret[59] (version 6.0-86) using repeated cross-validation (three repeats). Prediction variables were preprocessed by centring, scaling and nearest-neighbour imputation for samples that lacked data for a variable. A training set was generated with 75% of the data. Random forest regression was performed to minimize the root mean squared error with repeated cross-validation. Variable importance was assessed via generalized cross-validation in the package caret[59].

## ASV differential abundance analysis

Differential abundance of ASVs in the soil versus *A. thaliana*, and *A. thaliana* versus other Brassicaceae was assessed using the edgeR[18] package in R (version 3.28.1). We estimated a common negative binomial dispersion parameter, and abundance-dispersion trends by Cox–Reid approximated profile likelihoods[60]. We then fit a quasi-likelihood negative binomial generalized log-linear model to the count data. We tested for differential abundance by a likelihood ratio test.

## Phylotype classification and regression

Phylotypic clusters were identified by *k*-means clustering of Hellinger-transformed ASV count matrices. The optimal number of clusters was determined through both partitioning around medioids[61] using the pamk function in the R package fpc[62] (version 2.2.9) and through silhouette analysis[19] in the cluster (version 2.1.2) package in R[63].

To determine the relative effect sizes of drought, latitude and plant identity on MDS loadings, phenotypes were modelled using restricted expectation maximum likelihood with the lmekin package in R with kinship as a random effect[64]. The kinship matrix was constructed using several methods including the R package gaston[64] as well as the centred kinship matrix in gemma (version 0.98.3)[65]. The different methods yielded unstable estimates of kinship, probably due to the low coverage of the plant genomes. To account for the low coverage, we employed a method designed for kinship estimation in low coverage data, SEEKIN[66] using the homogeneous parameter. Mixed-effects modelling with a kinship matrix was computed both with lmekin[67] and with GEMMA. The data distribution was assumed to be normal but this was not formally tested. The proportion of phenotypic variance explained by the environmental covariates was estimated with the function 'r.squaredLR' from the package MuMIn (version 1.43.1) and the pseudo-heritability was estimated using the kinship matrix and lmekin as well as in GEMMA (-gk = 1, maf = 0.1). In the paper we report the lower estimate for pseudo-heritability as estimated in GEMMA with the centred kinship matrix also estimated in GEMMA.

To test for the relative effects of genotype, latitude and PDSI in a single model, we estimated the first five principal components of the plant genotype relatedness matrix[68] and included the eigenvectors as covariates in our models for microbiome type and the loading on MDS1 and MDS2 (Fig. 2). The data distribution was assumed to be normal but this was not formally tested. Regressions used the lm and glm functions (logit link) in the R stats package. The relative importance of PDSI and Tour ID were tested with the models in glm glmer(cluster identity) ~ PDSI + 1|Tour_ID, family = 'binomial') or with lmer(MDS1 ~ PDSI + 1|Tour_ID).

## Plant polymorphism calling and filtering

Raw reads were mapped to the TAIR10 reference genome of *A. thaliana* with bwa-mem (bwa 0.7.15)[69]. SNP calling was performed using GATK (version 3.5) HaplotypeCaller using recommended best practices[70] with some modifications. Filtering for individuals with greater than 25% missing data (across all the SNPs) and bi-allelic SNPs with greater than 25% missing data (across all the individuals) resulted in a final set of 527 individuals with 409,850 bi-allelic SNPs for further analysis.

## Population structure analysis of *A. thaliana*

Wright's fixation index ($F_{st}$) was calculated using the method of Cockeram and Weir[71]. The 1001 Genomes[1] dataset (without individuals from North America) was merged with the dataset from this study to perform principal component analysis. Genotypes from this study were projected into the principal component space of the 1001 Genomes genotypes using the SmartPCA tool of EIGENSOFT (version 6)[72].

## Heritability comparisons

For comparison of ASV distributions and heritability estimates, we identified related OTUs from four microbiome common garden experiments in Sweden[8]. OTU sequences from ref. 8 were downloaded from https://forgemia.inra.fr/bbrachi/microbiota_paper, as were heritability estimates for the OTUs. Correspondence between Swedish OTUs (called from sequenced V5–V7 region of 16S rDNA) and the ASVs in our study (identified from sequenced V3–V4 regions of the 16S rDNA locus) was established using the Qiime2[73] fragment insertion method using sepp-refs-gg-13-8 as the reference database. Correspondence between the OTUs and ASVs was established with divergence of less than 1% on the Green Genes tree.

## Reporting summary

Further information on research design is available in the Nature Portfolio Reporting Summary linked to this article.

## Data availability

The V3–V4 16S rDNA sequence data and metagenomic sequencing data of plants were deposited in the European Nucleotide Archive (ENA) under the Primary Accession ENA: PRJEB44379. Metadata and processed read data sets including phyloseq objects are available at Zenodo via https://doi.org/10.5281/zenodo.11187761 (ref. 74).

## Code availability

Scripts for data processing, analyses and figure generation can be accessed at GitHub via https://github.com/tkarasov/pathodopsis.

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

## Acknowledgements

We thank J. Keck, T. Hagmaier, A. Rütten, T. Vaupel, K. Poersch, N. Vasilenko, H. Vo-Gia, J. Elis, C. Tahtsidou, T. Schlegel and F. Vogt

for aliquoting soil. We thank F. Roux, H. Burbano, A. Duque and M. Collenberg for comments on the paper. We thank M. Horton for providing global SNP $F_{st}$ values. We thank H. Burbano, M. Horton and B. Brachi for discussion. This work was funded by an HFSPO Long-term Fellowship LT000348/2016-L EMBO LRTF 1483-2015 and NIH grant R35 GM150722-01 (T.L.K.), ERC-SyG PATHOCOM (J.B. and D.W.) and the Max Planck Society (D.W.).

## Author contributions

T.L.K., R.S., G.S., J.B., M.E.-A. and D.W. devised the study. T.L.K., R.S., G.S., M.N., L.L., E.S. and the PATHODOPSIS collection team collected and prepared the samples. T.L.K., G.S. and M.N. processed the samples. T.L.K., R.S. and G.S. analysed the data. G.M. provided climate data. A.H. performed infection experiments. T.L.K., D.W. and R.S. wrote the paper.

## Funding

## Competing interests

D.W. holds equity in Computomics, which advises plant breeders. D.W. consults for KWS SE, a plant breeder and seed producer. The other authors declare no competing interests.

## Additional information

**Extended data** is available for this paper at https://doi.org/10.1038/s41564-024-01773-z.

**Correspondence and requests for materials** should be addressed to Talia L. Karasov or Detlef Weigel.

## Pathodopsis Team

A. Cristina Barragán[2], Ilja Bezrukov[2], Claudia Friedemann[2], Alba González Hernando[2], Anette Habring[2], Julia Hildebrandt[2], Sonja Kersten[2], Patricia Lang[2], Sergio M. Latorre[2], Miriam Lucke[2], Derek S. Lundberg[2], Ulrich Lutz[2], Fiona Paul[2], Fernando A. Rabanal[2], Julian Regalado[2], Thanvi Srikant[2], Bridgit Waithaka[2], Anjar T. Wibowo[2] & Wei Yuan[2]

**Extended Data Table 1 | Variables included in random forest modelling; ANOVA Tables, Accessions used in field experiment, ASVs tracked in field experiment**

| Variable | Description of variable |
|---|---|
| ClimateZ | Köppen Climate Classification derived from R package kgc (version 1.0.0.2 in 9/2000) |
| HpA_plant | Observation of HpA sporulation on the collected plant: yes (1), no (0) |
| HpA_where | Location of visible HpA sporulation on the collected plant: rosette leaf (RL), cauline leaf (CL) |
| HpA_site | Observation of HpA sporulaton on plants at the collection site as a whole: yes (1), no (0) |
| Albugo_tour | Observation of Albugo sporulation on the collected plant: yes (1), no (0) |
| Developmental_state | vegetative (1), just bolting (2), flowering (3), mature (4), drying (5) |
| R_diameter | Diameter of rosette of collected plant estimated from photos including ruler. <1 cm (1), 1–2 cm (2), 2–3 cm (3), 3–4 cm (4), 4–5 cm (5), >5 cm (6) |
| Herbivory | no (1), weak (2), strong (3), very strong (4) herbivory |
| Ath.Ath | Distance between the collected and the closest other *Arabidopsis thaliana* plant. Touching (1), <1 cm (2), 1–3 cm (3), 3–5 cm (4), 5–10 cm (5), >10 cm (6) |
| Ath.other | Distance between the collected *Arabidopsis thaliana* plant and the closest other mono- or dicot. Touching (1), <1 cm (2), 1–3 cm (3), 3–5 cm (4), 5–10 cm (5), >10 cm (6) |
| Humidity_ground | Humidity of the surface on which the collected *Arabidopsis thaliana* plant grew as estimated from photos taken during collection. Very dry (1), dry (2), moist (3), wet (4), very wet (5) |
| Slope | Slope at collection point scored from photos taken during collection. Flat (1), mild slope (2), medium slope (3), very steep (4), in wall (5) |
| Sun | Estimate of average sun exposure at collection site scored from photos taken during collections. Full sun (1), mostly sun (2), sun and shade (3), mostly shaded (4) |
| Site_type | Classification of collection point based on photos taken during collections and virtual re-visits via Google Maps. Roadside (1), garden/park (2), railway (3), parking (4), wineyard/orchard (5), cemetery/church (6), wall (7), riverbank (8), beach (9), rock/cliff (10), field/meadow (11), sidewalk (12), dirthill (13), forest (14) |
| Site_category | Categorization of collection site based on virtual re-visits via Google Maps. Town (1), agricultural (2), visited/sightseeing (3), nature (4) |
| Aspect | North (N), East (E), South (S), West (W), flat (A), unknown (cannot_say) |
| Ground_cover | Estimated %age of ground covered with plants in a 20x30 cm rectangle around the collected plant. 1–25% (1), 25–50% (2), 50–75% (3), 75–90% (4), >90% (5) |
| Strata_herbs | Based on a 360˚C photo taken at the collection site: percent estimate of herbal plants covering the field of view. Other strata variables considered: shrubs, trees, wall of shrub height, wall of tree height, road, water. |
| Elevation | Elevation at collection site |
| Air_temp | Estimated air temperature at time of collection |
| Air_hum | Estimated air humidity at time of collection |
| Soil_temp | Estimated soil temperature at time of collection |
| Soil_hum | Estimated soil humidity at time of collection |
| Pop_size | Estimated size of visible *A. thaliana* population |
| tmax | Terraclim variable: averaged over six months prior to collection date. |
| tmin | Terraclim variable: averaged over six months prior to collection date. |
| vap | Terraclim variable: averaged over six months prior to collection date. |
| ppt | Terraclim variable: averaged over six months prior to collection date. |
| srad | Terraclim variable: averaged over six months prior to collection date. |
| soil | Terraclim variable: averaged over six months prior to collection date. |
| ws | Terraclim variable: averaged over six months prior to collection date. |
| aet | Terraclim variable: averaged over six months prior to collection date. |
| def | Terraclim variable: averaged over six months prior to collection date. |
| PDSI | Terraclim variable: averaged over six months prior to collection date. |
| vpd | Terraclim variable: averaged over six months prior to collection date. |
| pet | Terraclim variable: averaged over six months prior to collection date. |

Extended Data Tables 1–4 in .xlsx format with tabs.

**Extended Data Table 2 | Anova Table (Type II Test) for results of logistic regression associating genetic relatedness of plant with microbiome cluster type**

| | LR Chisq | Df | Pr(>Chisq) |
|---|---|---|---|
| PC1[a] | 0.653 | 1 | 0.418944 |
| PC2[b] | 0.064 | 1 | 0.800149 |
| PC3[c] | 0.67 | 1 | 0.413121 |
| PC4[d] | 0.468 | 1 | 0.494094 |
| PC5[e] | 0.009 | 1 | 0.926298 |
| Lat[f] | 32.004 | 1 | 1.538e-08 *** |
| PDSI[g] | 7.775 | 1 | 0.005298 ** |
| --- | | | |

Signif. codes: 0 '***' 0.001 '**' 0.01 '*' 0.05 '.' 0.1 ' ' 1 [a]Principal Component 1 in Genetic Relatedness Matrix of *A. thaliana* plants [b]Principal Component 2 in Genetic Relatedness Matrix of *A. thaliana* plants [c]Principal Component 3 in Genetic Relatedness Matrix of *A. thaliana* plants [d]Principal Component 4 in Genetic Relatedness Matrix of *A. thaliana* plants [e]Principal Component 5 in Genetic Relatedness Matrix of *A. thaliana* plants [f]Latitude of collection point [g]PDSI of collection point.

**Extended Data Table 3 | Anova Table (Type II Test) for results of linear model associating genetic relatedness of plant with microbial loading on MDS1**

| | Sum Sq | Df | F value | Pr(>F) | Variance Explained |
|---|---|---|---|---|---|
| PC1[a] | 0.0169 | 1 | 0.9824 | 0.3222973 | 0.002503444 |
| PC2[b] | 0.0972 | 1 | 5.6382 | 0.018112* | 0.014398507 |
| PC3[c] | 0.2232 | 1 | 12.9508 | 0.0003659*** | 0.033063238 |
| PC4[d] | 0 | 1 | 0.0003 | 0.9859697 | 0 |
| PC5[e] | 0.0099 | 1 | 0.5759 | 0.4484379 | 0.001466515 |
| Lat[f] | 0.226 | 1 | 13.1138 | 0.0003364*** | 0.03347801 |
| PDSI[g] | 0.1461 | 1 | 8.4807 | 0.0038196** | 0.0216422 |
| Residuals | 6.0314 | 350 | | | 0.893448087 |

---

Signif. codes: 0 '***' 0.001 '**' 0.01 '*' 0.05 '.' 0.1 ' ' 1 [a]Principal Component 1 in Genetic Relatedness Matrix of *A. thaliana* plants [b]Principal Component 2 in Genetic Relatedness Matrix of *A. thaliana* plants [c]Principal Component 3 in Genetic Relatedness Matrix of *A. thaliana* plants [d]Principal Component 4 in Genetic Relatedness Matrix of *A. thaliana* plants [e]Principal Component 5 in Genetic Relatedness Matrix of *A. thaliana* plants [f]Latitude of collection point [g]PDSI of collection point.

**Extended Data Table 4 | Anova Table (Type II Test) for results of linear model associating genetic relatedness of plant with microbial loading on MDS2**

|  | Sum Sq | Df F | value | Pr(>F) | Explained Variance |
|---|---|---|---|---|---|
| PC1[a] | 1.6712 | 1 | 57.9316 | 0.0000000000002543*** | 0.127433412 |
| PC2[b] | 0.9664 | 1 | 33.4999 | 0.00000001584*** | 0.073690552 |
| PC3[c] | 0.0547 | 1 | 1.8947 | 0.16955 | 0.004171019 |
| PC4[d] | 0.1292 | 1 | 4.4779 | 0.03504* | 0.009851841 |
| PC5[e] | 0.1336 | 1 | 4.6303 | 0.0321* | 0.010187353 |
| Lat[f] | 0.0623 | 1 | 2.1586 | 0.14267 | 0.004750539 |
| PDSI[g] | 0 | 1 | 0 | 0.99481 | 0 |
| Residuals | 10.0969 | 350 |  |  | 0.769915283 |

---

Signif. codes: 0 '***' 0.001 '**' 0.01 '*' 0.05 '.' 0.1 ' ' 1 [a]Principal Component 1 in Genetic Relatedness Matrix of *A. thaliana* plants [b]Principal Component 2 in Genetic Relatedness Matrix of *A. thaliana* plants [c]Principal Component 3 in Genetic Relatedness Matrix of *A. thaliana* plants [d]Principal Component 4 in Genetic Relatedness Matrix of *A. thaliana* plants [e]Principal Component 5 in Genetic Relatedness Matrix of *A. thaliana* plants [f]Latitude of collection point [g]PDSI of collection point.

**Extended Data Table 5 | Arabidopsis thaliana accessions used in field experiment, their collection location information and ABRC stock number**

| id | name | country | latitude | longitude | ABRC |
|---|---|---|---|---|---|
| 159 | MAR2-3 | France | 47.35 | 3.93333 | CS77070 |
| 403 | Zdarec3 | Czech Republic | 49.3667 | 16.2667 | CS78873 |
| 765 | Sus-1 | Kyrgyzstan | 42.1833 | 73.4 | CS76607 |
| 766 | Dja-1 | Kyrgyzstan | 42.5833 | 73.6333 | CS76473 |
| 768 | Zal-1 | Kyrgyzstan | 42.8 | 76.35 | CS76634 |
| 772 | Neo-6 | Tajikistan | 37.35 | 72.4667 | CS76560 |
| 5349 | UKSE06-639 | UK | 51.1 | 0.4 | CS78807 |
| 5486 | UKNW06-233 | UK | 54.6 | −3.3 | CS78794 |
| 5577 | UKNW06-403 | UK | 54.7 | −3.4 | CS78797 |
| 5768 | UKID63 | UK | 54.1 | −1.5 | CS78786 |
| 5772 | Set-1 | UK | 54.1 | −2.3 | CS78787 |
| 5811 | UKID107 | UK | 52.9 | −3.1 | CS78778 |
| 6008 | Duk | Czech Republic | 49.1 | 16.2 | CS76824 |
| 6094 | T1040 | Sweden | 55.6494 | 13.2147 | CS77290 |
| 6098 | T1080 | Sweden | 55.6561 | 13.2178 | CS77292 |
| 6099 | T1090 | Sweden | 55.6575 | 13.2386 | CS77293 |
| 6108 | T480 | Sweden | 55.7989 | 13.1206 | CS77300 |
| 6112 | T540 | Sweden | 55.7967 | 13.1044 | CS77303 |
| 6125 | T710 | Sweden | 55.8403 | 13.3106 | CS77310 |
| 6126 | T720 | Sweden | 55.8411 | 13.3047 | CS77311 |
| 6131 | T780 | Sweden | 55.8369 | 13.3181 | CS77315 |
| 6133 | T800 | Sweden | 55.8364 | 13.2906 | CS77317 |
| 6137 | T850 | Sweden | 55.9419 | 13.5603 | CS77320 |
| 6142 | T900 | Sweden | 55.9428 | 13.5558 | CS77323 |
| 6173 | TÄD 05 | Sweden | 62.8717 | 18.3419 | CS77336 |
| 6180 | TÄL 07 | Sweden | 62.6322 | 17.6906 | CS77339 |
| 6201 | TDr-16 | Sweden | 55.7719 | 14.1211 | CS77348 |
| 6202 | TDr-17 | Sweden | 55.7717 | 14.1206 | CS77349 |
| 6217 | TFÄ 07 | Sweden | 63.0169 | 18.3283 | CS77363 |
| 6218 | TFÄ 08 | Sweden | 63.0172 | 18.3283 | CS77364 |
| 6911 | Cvi-0 | Cape Verde | 15.1111 | −23.6167 | CS76789 |
| 6929 | Kondara | Tajikistan | 38.48 | 68.49 | CS76532 |
| 6938 | Ms-0 | Russia | 55.7522 | 37.6322 | CS76555 |
| 6940 | Mz-0 | GER | 50.3 | 8.3 | CS76557 |
| 6963 | Sorbo | Tajikistan | 38.35 | 68.48 | CS78917 |
| 7003 | Bs-1 | SUI | 47.5 | 7.5 | CS78888 |
| 7026 | Boot-1 | UK | 54.4 | −3.2667 | CS76452 |
| 7063 | Can-0 | Spain | 29.2144 | −13.4811 | CS76740 |
| 7081 | Co | POR | 40.2077 | −8.42639 | CS78895 |
| 7106 | Dr-0 | GER | 51.051 | 13.7336 | CS78897 |
| 7164 | Hau-0 | DEN | 55.675 | 12.5686 | CS76915 |
| 7186 | Kn-0 | Lithuania | 54.8969 | 23.8924 | CS76969 |
| 7208 | Lan-0 | UK | 55.6739 | −3.78181 | CS76539 |
| 7255 | Mh-0 | Poland | 50.95 | 20.5 | CS76550 |
| 7282 | Or-0 | GER | 50.3827 | 8.01161 | CS76568 |
| 7323 | Rubezhnoe-1 | Ukraine | 49 | 38.28 | CS76594 |
| 7337 | Si-0 | GER | 50.8738 | 8.02341 | CS76601 |

**Extended Data Table 5 (continued) | Arabidopsis thaliana accessions used in field experiment, their collection location information and ABRC stock number**

| id | name | country | latitude | longitude | ABRC |
|---|---|---|---|---|---|
| 7394 | Wa-1 | Poland | 52.3 | 21 | CS76626 |
| 8236 | HSm | Czech Republic | 49.33 | 15.76 | CS76941 |
| 8242 | Lillö-1 | Sweden | 56.1494 | 15.7884 | CS77039 |
| 8244 | PHW-34 | France | 48.6103 | 2.3086 | CS77174 |
| 8312 | Is-0 | GER | 50.5 | 7.5 | CS78904 |
| 9399 | Hamm-1 | Sweden | 55.4234 | 13.9905 | CS76910 |
| 9408 | Kal 1 | Sweden | 56.047 | 13.9519 | CS76959 |
| 9512 | IP-Vid-1 | POR | 38.22 | −7.84 | CS78842 |
| 9519 | IP-Ang-0 | Spain | 41.94 | 2.64 | CS78886 |
| 9529 | IP-Cap-1 | Spain | 36.97 | −3.36 | CS76741 |
| 9542 | IP-Fun-0 | Spain | 40.79 | −4.05 | CS76872 |
| 9544 | IP-Gua-1 | Spain | 39.4 | −5.33 | CS76894 |
| 9545 | IP-Her-12 | Spain | 39.4 | −5.78 | CS76920 |
| 9549 | IP-Hum-2 | Spain | 42.23 | −3.69 | CS76943 |
| 9555 | IP-Mar-1 | Spain | 39.58 | −3.93 | CS77068 |
| 9571 | IP-Pro-0 | Spain | 43.28 | −6.01 | CS78914 |
| 9577 | IP-Ria-0 | Spain | 42.34 | 2.17 | CS77216 |
| 9583 | IP-Sne-0 | Spain | 37.09 | −3.38 | CS77258 |
| 9599 | IP-Vin-0 | Spain | 42.8 | −5.77 | CS78846 |
| 9615 | Parti-1 | Russia | 52.99 | 52.16 | CS77163 |
| 9619 | Basta-1 | Russia | 51.84 | 79.48 | CS76691 |
| 9625 | Kolyv-2 | Russia | 51.31 | 82.59 | CS76977 |
| 9629 | K-oze-1 | Russia | 51.35 | 82.18 | CS76957 |
| 9631 | Lebja-1 | Russia | 51.65 | 80.79 | CS77015 |
| 9634 | Masl-1 | Russia | 54.13 | 81.31 | CS77073 |
| 9637 | Noveg-2 | Russia | 51.77 | 80.85 | CS77132 |
| 9641 | Rakit-2 | Russia | 51.9 | 80.06 | CS77203 |
| 9642 | Rakit-3 | Russia | 51.84 | 80.06 | CS77204 |
| 9643 | Sever-1 | Russia | 52.1 | 79.31 | CS77245 |
| 9653 | Giffo-1 | Italy | 38.44 | 16.13 | CS76878 |
| 9657 | Melic-1 | Italy | 38.45 | 16.04 | CS77078 |
| 9659 | Pigna-1 | Italy | 41.18 | 14.18 | CS77177 |
| 9660 | Sarno-1 | Italy | 40.84 | 14.57 | CS77236 |
| 9661 | Cimin-1 | Italy | 39.58 | 16.21 | CS76771 |
| 9697 | Dolen-1 | BUL | 41.62 | 23.94 | CS76802 |
| 9701 | Ivano-1 | BUL | 43.7 | 25.91 | CS76954 |
| 9716 | Leska-1-44 | BUL | 41.54 | 24.98 | CS77030 |
| 9718 | Smolj-1 | BUL | 41.55 | 24.75 | CS77256 |
| 9723 | Slavi-2 | BUL | 41.42 | 23.67 | CS77252 |
| 9726 | Faneronemi-3 | Greece | 37.07 | 22.04 | CS76853 |
| 9744 | Iasi-1 | Romania | 47.16 | 27.59 | CS76944 |
| 9759 | Anz-0 | Iran | 37.47 | 49.47 | CS76439 |
| 9761 | Bik-1 | Lebanon | 33.92 | 35.7 | CS76449 |
| 9764 | Qar-8a | Lebanon | 34.1 | 35.84 | CS76581 |
| 9779 | Bai-10 | GER | 48.5 | 8.78 | CS76682 |
| 9826 | IP-Bor-0 | Spain | 42.49 | −6.71 | CS76717 |
| 9830 | IP-Bus-0 | Spain | 36.97 | −3.28 | CS76736 |

**Extended Data Table 5 (continued) | Arabidopsis thaliana accessions used in field experiment, their collection location information and ABRC stock number**

| id | name | country | latitude | longitude | ABRC |
|----|------|---------|----------|-----------|------|
| 9832 | IP-Cat-0 | Spain | 40.54 | −3.69 | CS76759 |
| 9843 | IP-Elp-0 | Spain | 40.53 | −3.92 | CS76840 |
| 9846 | IP-Ezc-2 | Spain | 42.31 | −3.02 | CS76849 |
| 9871 | IP-Nac-0 | Spain | 40.75 | −3.99 | CS77117 |
| 9881 | IP-Pie-0 | Spain | 40.46 | −5.32 | CS77176 |
| 9882 | IP-Pil-0 | Spain | 40.46 | −4.26 | CS77178 |
| 9885 | IP-Prd-0 | Spain | 41.14 | −3.68 | CS77189 |
| 9890 | IP-Rib-1 | Spain | 43.16 | −5.07 | CS77217 |
| 9892 | IP-Sam-0 | Spain | 42.68 | −6.96 | CS77231 |
| 9901 | IP-Urd-1 | Spain | 42.27 | −2.98 | CS78824 |
| 9903 | IP-Val-0 | Spain | 42.31 | −3.1 | CS78829 |
| 9933 | VED-10 | France | 43.74 | 3.89 | CS78839 |
| 9941 | Fei-0 | POR | 40.92 | −8.54 | CS76412 |
| 9947 | Ped-0 | Spain | 40.74 | −3.9 | CS76415 |
| 10020 | Jl-2 | Czech Republic | 49.17 | 16.5 | CS76956 |

**Extended Data Table 6 | ASVs tracked in field experiment and the associated p-values in regression modelling with genotype x environment interactions**

| ASV_seqID_from_large_experiment[a] | p_water_treatment[b] | p_interaction_genotype_environment[c] | p_genetic_risk[d] | p_cluster_association[e] | ASV_Genus[f] | ASV_sequence[g] |
|---|---|---|---|---|---|---|
| seq_01 | 0.27 | 0.02 | 0.03 | 0.00 | Sphingomonas | TAAGGAATATTGGTCAATGGAGGCAACTCTG AACCAGCCATGCCGCGTGCAGGAAGACGGCCC TATGGGTTGTAAACTGCTTTTATCCGGGAATAAA CCTTTCTACGTGTAGAGAGCTGAATGTACCGGA AGAATAAGGATCGGCTAACTCCGTGCCAGCAGC CGCGGTAATACGGAGGATCCAAGCGTTATCCGG ATTTATTGGGTTTAAAGGGTGCGTAGGCGGCCT GTTAAGTCAGAGGTGAAAGACGGTAGCTCAACT ATCGCAGTGCCTTTGATACTGACGGGCTTGAAT GAACTAGAGGTAGGCGGAATGAGACAAGTAGCG GTGAAATGCATAGATATGTCTCAGAACACCGAT TGCGAAGGCAGCTTACTATGGTTTTATTGACGC TGAGGCACGAAAGCGTGGGGATCAAACAGG |
| seq_05 | 0.12 | 0.74 | 0.16 | 1.88E-05 | Allorhizobium-Neorhizobium-Pararhizobium-Rhizobium | TAGGGAATATTGGACAATGGGGGGCAACCCTGATCC AGCAATGCCGCGTGAGTGATGAAGGCCTTAGGGTT GTAAAGCTCTTTTACCCGAGATGATAATGACAGTAT CGGGAGAATAAGCTCCGGCTAACTCCGTGCCAGC AGCCGCGGTAATACGGAGGGAGCTAGCGTTGTT CGGAATTACTGGGCGTAAAGCGCACGTAGGCGG CGATTTAAGTCAGAGGTGAAAGCCCGGGGCTCA ACCCCGGAACTGCCTTTGAGACTGGATTGCTAGA ATCTTGGAGAGGCGGGTGGAATTCCGAGTGTAGA GGTGAAATTCGTAGATATTCGGAAGAACACCAGTG GCGAAGGCGGCCCGCTGGACAAGTATTGACGCT GAGGTGCGAAAGCGTGGGGAGCAAACAGG |
| seq_06 | 0.89 | 0.51 | 0.40 | 0.38 | Sphingomonas | TAGGGAATATTGGGCAATGGGCGAGAGCCTGACC CAGCCATGCCGCGTGCAGGAAGAAGGCTTTCT GAGTCGTAAACTGCTTTTGACAGGGAAGAATAA GCACTACGTGTAGTGCGATGACGGTACCTGCA GAATAAGCACCGGCTAACTCCGTGCCAGCAGC CGCGGTAATACGGAGGGTGCGAGCGTTGTCCG GATTTATTGGGTTTAAAGGGTGCGTAGGCGGCCG TTTAAGTCTGGGGTGAAAGCCCGCTGCTCAACA GCGGAACTGCCCTGGATACTGGATGGCTTGAG TACAGACGAGGTTGGCGGAATGGACTGAGTAGCG GTGAAATGCATAGATACAGTCCAGAACCCCGAT TGCGAAGGCAGCTGACTAGGCTGTTACTGACG CTGAGGCACGAAAGCGTGGGGAGCGAACAGG |
| seq_07 | 0.93 | 0.74 | 0.23 | 6.82E-11 | Sphingomonas | TGGGGAATATTGGACAATGGGCGAAAGCCTGATC CAGCAATGCCGCGTGAGTGATGAAGGCCTTAGGGT TGTAAAGCTCTTTTACCAGGGATGATAATGACAGTA CCTGGAGAATAAGCTCCGGCTAACTCCGTGCCA GCAGCCGCGGTAATACGGAGGGAGCTAGCGTTG TTCGGAATTACTGGGCGTAAAGCGTACGTAGGC GGTTATTCAAGTCAGAGGTGAAAGCCTGGAGCTC AACTCCAGAACTGCCTTTGAAACTAGATAGCTA GAATCTTGGAGAGGTGAGTGGAATTCCGAGTGT AGAGGTGAAATTCGTAGATATTCGGAAGAACAC CAGTGGCGAAGGCGACTCACTGGACAAGTATTG ACGCTGAGGTACGAAAGCGTGGGGAGCAAACAGG |
| seq_08 | 0.24 | 0.74 | 0.07 | 2.27E-16 | Sphingomonas | TGGGGAATATTGGACAATGGGCGAAAGCCTGATCC AGCAATGCCGCGTGAGTGATGAAGGCCTTAGGG TTGTAAAGCTCTTTTACCCGGGAAGATAATGACTGT ACCGGGAGAATAAGCCCCGGCTAACTCCGTGCC AGCAGCCGCGGTAATACGGAGGGGGCTAGCGTT GTTCGGAATTACTGGGCGTAAAGCGTACGTAGGC GGTTTTGTAAGTTAGAGGTGAAAGCCCGGAGCTCA ACTTCGGAATTGCCTTTAAGACTGCATCACTTGAAC GTCGGAGAGGTGAGTGGAATTCCGAGTGTAGAGG TGAAATTCGTAGATATTCGGAAGAACACCAGTGGC GAAGGCGGCTCACTGGACGACTGTTGACGCTGAG GTACGAAAGCGTGGGGAGCAAACAGG |
| seq_10 | 0.24 | 0.92 | 0.81 | 2.77E-24 | Pseudomonas | TGGGGAATATTGGACAATGGGCGAAAGCCTGATCC AGCAATGCCGCGTGAGTGATGAAGGCCTTAGGGT TGTAAAGCTCTTTTACCCGGGATGATAATGACAGT ACCGGGAGAATAAGCTCCGGCTAACTCCGTGCC AGCAGCCGCGGTAATACGGAGGGAGCTAGCGTTA TTCGGAATTACTGGGCGTAAAGCGCACGTAGGCG GCTTTGTAAGTAAGAGGTGAAAGCCCAGAGCTCA ACTCTGGAATTGCCTTTTAGACTGCATCGCTTGAA TCATGGAGAGGTCAGTGGAATTCCGAGTGTAGAG GTGAAATTCGTAGATATTCGGAAGAACACCAGTGG CGAAGGCGGCTGACTGGACATGTATTGACGCTGA GGTGCGAAAGCGTGGGGAGCAAACAGG |

**Extended Data Table 6 (continued) | ASVs tracked in field experiment and the associated p-values in regression modelling with genotype x environment interactions**

| ASV_seqID_from_large_experiment[a] | p_water_treatment[b] | p_interaction_genotype_environment[c] | p_genetic_risk[d] | p_cluster_association[e] | ASV_Genus[f] | ASV_sequence[g] |
|---|---|---|---|---|---|---|
| seq_11 | 0.01 | 0.00 | 0.01 | 3.93E-09 | Sphingomonas | TGGGGAATATTGGACAATGGGCGAAAGCCTGATCCAGCAATGCCGCGTGAGTGATGAAGGCCTTAGGGTTGTAAAGCTCTTTTACCCGGGATGATAATGACAGTACCGGGAGAATAAGCTCCGGCTAACTCCGTGCCAGCAGCCGCGGTAATACGGAGGGAGCTAGCGTTATTCGGAATTACTGGGCGTAAAGCGCACGTAGGCGGCTTTGTAAGTAAGAGGTGAAAGCCTGGTGCTCAACACCAGAACTGCCTTTTAGACTGCATCGCTTGAATCCAGGAGAGGTGAGTGGAATTCCGAGTGTAGAGGTGAAATTCGTAGATATTCGGAAGAACACCAGTGGCGAAGGCGGCTCACTGGACTGGTATTGACGCTGAGGTGCGAAAGCGTGGGGAGCAAACAGG |
| seq_12 | 0.70 | 0.74 | 0.00 | 9.77E-71 | Methylobacterium | TGGGGAATATTGGACAATGGGCGAAAGCCTGATCCAGCAATGCCGCGTGAGTGATGAAGGCCTTAGGGTTGTAAAGCTCTTTTACCCGGGATGATAATGACAGTACCGGGAGAATAAGCTCCGGCTAACTCCGTGCCAGCAGCCGCGGTAATACGGAGGGAGCTAGCGTTATTCGGAATTACTGGGCGTAAAGCGCACGTAGGCGGCTTTGTAAGTTAGAGGTGAAAGCCTGGAGCTCAACTCCAGAATTGCCTTTGATACTGCATGGCTTGAATCCAGGAGAGGTGAGTGGAATTCCGAGTGTAGAGGTGAAATTCGTAGATATTCGGAAGAACACCAGTGGCGAAGGCGGCTCACTGGACTGGTATTGACGCTGAGGTGCGAAAGCGTGGGGAGCAAACAGG |
| seq_15 | 0.95 | 0.14 | 0.56 | 2.31E-45 | Sphingomonas | TGGGGAATATTGGACAATGGGCGAAAGCCTGATCCAGCAATGCCGCGTGAGTGATGAAGGCCTTAGGGTTGTAAAGCTCTTTTACCCGGGATGATAATGACAGTACCGGGAGAATAAGCTCCGGCTAACTCCGTGCCAGCAGCCGCGGTAATACGGAGGGAGCTAGCGTTGTTCGGAATTACTGGGCGTAAAGCGCACGTAGGCGGCTTTGTAAGTTAGAGGTGAAAGCCTGGAGCTCAACTCCAGAATTGCCTTTAAGACTGCATCGCTTGAATCCAGGAGAGGTGAGTGGAATTCCGAGTGTAGAGGTGAAATTCGTAGATATTCGGAAGAACACCAGTGGCGAAGGCGGCTCACTGGACTGGTATTGACGCTGAGGTGCGAAAGCGTGGGGAGCAAACAGG |
| seq_17 | 4.48E-10 | 0.39 | 0.48 | 2.55E-12 | Sphingomonas | TGGGGAATATTGGACAATGGGCGCAAGCCTGATCCAGCCATGCCGCGTGAATGATGAAGGCCTTAGGGTTGTAAAGTTCTTTCACCGGAGAAGATAATGACGGTATCCGGAGAAGAAGCCCCGGCTAACTTCGTGCCAGCAGCCGCGGTAATACGAAGGGGGCTAGCGTTGTTCGGAATTACTGGGCGTAAAGCGCACGTAGGCGGATCGATCAGTCAGGGGTGAAATCCCAGAGCTCAACTCTGGAACTGCCTTTGATACTGTCGGTCTAGAGTATGGAAGAGGTGAGTGGAATTCCGAGTGTAGAGGTGAAATTCGTAGATATTCGGAGGAACACCAGTGGCGAAGGCGGCTCACTGGTCCATTACTGACGCTGAGGTGCGAAAGCGTGGGGAGCAAACAGG |
| seq_18 | 0.82 | 0.17 | 0.03 | 2.06E-07 | Allorhizobium-Neorhizobium-Pararhizobium-Rhizobium | TGGGGAATATTGGACAATGGGCGCAAGCCTGATCCAGCCATGCCGCGTGAGTGATGAAGGCCTTAGGGTTGTAAAGCTCTTTCAGTGGGGAAGATAATGACGGTACCCACAGAAGAAGCCCCGGCTAACTTCGTGCCAGCAGCCGCGGTAATACGAAGGGGGCTAGCGTTGTTCGGATTTACTGGGCGTAAAGCGCACGTAGGCGGATTGTTAAGTGAGGGGTGAAATCCTGGAGCTCAACTCCAGAACTGCCTTTCATACTGGCAATCTAGAGTCCGGAAGAGGTAAGTGGAACTCCTAGTGTAGAGGTGGAATTCGTAGATATTAGGAAGAACACCAGTGGCGAAGGCGGCTTACTGGTCCGGTACTGACGCTGAGGTGCGAAAGCGTGGGGAGCAAACAGG |
| seq_19 | 0.94 | 0.51 | 0.56 | 0.00 | Duganella | TGGGGAATATTGGACAATGGGCGCAAGCCTGATCCAGCCATGCCGCGTGAGTGATGAAGGCCTTAGGGTTGTAAAGCTCTTTTGTCCGGGACGATAATGACGGTACCGGAAGAATAAGCCCCGGCTAACTTCGTGCCAGCAGCCGCGGTAATACGAAGGGGGCTAGCGTTGCTCGGAATCACTGGGCGTAAAGGGCGCGTAGGCGGCCATTCAAGTCGGGGGTGAAAGCCTGTGGCTCAACCACAGAATTGCCTTCGATACTGTTTGGCTTGAGTATGGCAGAGGTCAGTGGAACTGCGAGTGTAGAGGTGAAATTCGTAGATATTCGCAAGAACACCAGTGGCGAAGGCGGCTGACTGGGCCATTACTGACGCTGAGGCGCGAAAGCGTGGGGAGCAAACAGG |

**Extended Data Table 6 (continued) | ASVs tracked in field experiment and the associated p-values in regression modelling with genotype x environment interactions**

| ASV_seqID_from_large_experiment[a] | p_water_treatment[b] | p_interaction_genotype_environment[c] | p_genetic_risk[d] | p_cluster_association[e] | ASV_Genus[f] | ASV_sequence[g] |
|---|---|---|---|---|---|---|
| seq_20 | 0.24 | 0.74 | 0.81 | 2.22E-42 | Methylobacterium | TGGGGAATATTGGACAATGGGCGCAAGCCTGATCCAGCCATGCCGCGTGAGTGATGAAGGCCTTAGGGTTGTAAAGCTCTTTTGTCCGGGACGATAATGACGGTACCGGAAGAATAAGCCCCGGCTAACTTCGTGCCAGCAGCCGCGGTAATACGAAGGGGGCTAGCGTTGCTCGGAATCACTGGGCGTAAAGGGCGCGTAGGCGGCCATTCAAGTCGGGGGTGAAAGCCTGTGGCTCAACCACAGAATTGCCTTCGATACTGTTTGGCTTGAGTTTGGTAGAGGTTGGTGGAACTGCGAGTGTAGAGGTGAAATTCGTAGATATTCGCAAGAACACCAGTGGCGAAGGCGGCCAACTGGACCAATACTGACGCTGAGGCGCGAAAGCGTGGGGAGCAAACAGG |
| seq_21 | 0.70 | 0.00 | 0.23 | 1.35E-10 | Methylobacterium | TGGGGAATATTGGACAATGGGCGCAAGCCTGATCCAGCCATGCCGCGTGAGTGATGAAGGTCTTAGGATTGTAAAGCTCTTTCACCGGGGACGATAATGACGGTACCCGGAGAAGAAGCCCCGGCTAACTTCGTGCCAGCAGCCGCGGTAATACGAAGGGGGCTAGCGTTGTTCGGAATTACTGGGCGTAAAGCGCACGTAGGCGGACATTTAAGTCAGGGGTGAAATCCCGGGGCTCAACCCCGGAACTGCCTTTGATACTGGTGTCTTGAGTGTGGTAGAGGTGAGTGGAATTGCGAGTGTAGAGGTGAAATTCGTAGATATTCGCAGGAACACCAGTGGCGAAGGCGGCTCACTGGACCACAACTGACGCTGAGGTGCGAAAGCGTGGGGAGCAAACAGG |
| seq_22 | 0.24 | 0.74 | 0.17 | 1.15E-25 | Aureimonas | TGGGGAATATTGGACAATGGGCGCAAGCCTGATCCAGCCATGCCGCGTGAGTGATGAAGGTCTTAGGATTGTAAAGCTCTTTCAGTGGGGACGATAATGACGGTACCCACAGAAGAAGCCCCGGCTAACTTCGTGCCAGCAGCCGCGGTAATACGAAGGGGGGCTAGCGTTGTTCGGAATTACTGGGCGTAAAGCGCACGTAGGCGGATATTTAAGTCGGGGGGTGAAATCCCGGGGCTCAACCCCGGAACTGCCTTCGATACTGGGTATCTTGAGTTCGGAAGAGGTGAGTGGAATTGCGAGTGTAGAGGTGAAATTCGTAGATATTCGCAGGAACACCAGTGGCGAAGGCGGCTCACTGGTCCGATACTGACGCTGAGGTGCGAAAGCGTGGGGAGCAAACAGG |
| seq_23 | 0.29 | 0.74 | 0.63 | 0.10 | Aureimonas | TGGGGAATATTGGACAATGGGCGCAAGCCTGATCCAGCCATGCCGCGTGTGTGATGAAGGCCTTAGGGTTGTAAAGCACTTTCACCGGTGAAGATAATGACGGTAACCGGAGAAGAAGCCCCGGCTAACTTCGTGCCAGCAGCCGCGGTAATACGAAGGGGGCTAGCGTTGTTCGGAATTACTGGGCGTAAAGCGCACGTAGGCGGATATTTAAGTCAGGGGGTGAAATCCCAGAGCTCAACTCTGGAACTGCCTTTGATACTGGGTATCTTGAGTATGGAAGAGGTGAGTGGAATTCCGAGTGTAGAGGTGAAATTCGTAGATATTCGGAGGAACACCAGTGGCGAAGGCGGCTCACTGGTCCATAACTGACGCTGAGGTGCGAAAGCGTGGGGAGCAAACAGG |
| seq_24 | 0.90 | 0.92 | 0.92 | 0.06 | Bradyrhizobium | TGGGGAATATTGGACAATGGGCGGAAGCCTGATCCAGCAACGCCGCGTGAGGGATGACGGCCTTCGGGTTGTAAACCTCTTTCAGTATCGACGAAGCGCCCGTGTGGGTGGTGACGGTAGGTACAGAAGAAGCACCGGCCAACTACGTGCCAGCAGCCGCGGTAATACGTAGGGTGCGAGCGTTGTCCGGAATTATTGGGCGTAAAGGGCTCGTAGGCGGTTTGTCGCGTCGGGAGTGAAAACACTGGGCTTAACCGAGTGCTTGCTTTCGATACGGGCAGACTTGAGGCATTGAGGGGAGAACGGAATTCCTGGTGTAGCGGTGAAATGCGCAGATATCAGGAGGAACACCGGTGGCGAAGGCGGTTCTCTGGCAATGTTCTGACGCTGAGGAGCGAAAGTGTGGGGAGCGAACAGG |
| seq_26 | 0.24 | 0.17 | 0.23 | 0.00 | Pseudarthrobacter | TGGGGAATCTTAGACAATGGGCGCAAGCCTGATCTAGCCATGCCGCGTGAGCGATGAAGGCCTTAGGGTTGTAAAGCTCTTTCAGTGGGGAAGATAATGACTGTACCCACAGAAGAAGCCCCGGCTAACTCCGTGCCAGCAGCCGCGGTAATACGGAGGGGGCTAGCGTTGTTCGGAATTACTGGGCGTAAAGCGCACGTAGGCGGACTGGAAAGTCAGAGGTGAAATCCCAGGGCTCAACCTTGGAACTGCCTTTGAAACTCCCGGTCTTGAGGTCGAGAGAGGTGAGTGGAATTCCGAGTGTAGAGGTGAAATTCGTAGATATTCGGAGGAACACCAGTGGCGAAGGCGGCTCACTGGCTCGATACTGACGCTGAGGTGCGAAAGCGTGGGGAGCAAACAGG |

**Extended Data Table 6 (continued) | ASVs tracked in field experiment and the associated p-values in regression modelling with genotype x environment interactions**

| ASV_seqID_from_large_experiment[a] | p_water_treatment[b] | p_interaction_genotype_environment[c] | p_genetic_risk[d] | p_cluster_association[e] | ASV_Genus[f] | ASV_sequence[g] |
|---|---|---|---|---|---|---|
| seq_27 | 0.24 | 0.74 | 0.31 | 3.60E-37 | Methylotenera | TGGGGAATCTTGCGCAATGGGCGAAAGCCTGACGCAGCCATGCCGCGTGTATGATGAAGGTCTTAGGATTGTAAAATACTTTCACCGGTGAAGATAATGACTGTAGCCGGAGAAGAAGCCCCGGCTAACTTCGTGCCAGCAGCCGCGGTAATACGAAGGGGGCTAGCGTTGCTCGGAATTACTGGGCGTAAAGGGAGCGTAGGCGGACATTTAAGTCAGGGGTGAAATCCCAGAGCTCAACTCTGGAACTGCCTTTGATACTGGGTGTCTTGAGTGTGATAGAGGTATGTGGAACTCCGAGTGTAGAGGTGAAATTCGTAGATATTCGGAAGAACACCAGTGGCGAAGGCGACATACTGGATCATTACTGACGCTGAGGCTCGAAAGCGTGGGGAGCAAACAGG |
| seq_28 | 0.24 | 0.17 | 0.16 | 3.11E-07 | Kineosporia | TGGGGAATTTTGGACAATGGGCGCAAGCCTGATCCAGCAATGCCGCGTGCAGGAAGAAGGCCTTCGGGTTGTAAACTGCTTTTGTACGGAACGAAACGGTCCTGGTTAATACCTGGGGCTAATGACGGTACCGTAAGAATAAGCACCGGCTAACTACGTGCCAGCAGCCGCGGTAATACGTAGGGTGCAAGCGTTAATCGGAATTACTGGGCGTAAAGCGTGCGCAGGCGGTTTTGTAAGACAGGCGTGAAATCCCCGGGCTTAACCTGGGAATGGCGCTTGTGACTGCAAAGCTGGAGTGCGGCAGAGGGGGATGGAATTCCGCGTGTAGCAGTGAAATGCGTAGATATGCGGAGGAACACCGATGGCGAAGGCAATCCCCTGGGCCTGCACTGACGCTCATGCACGAAAGCGTGGGGAGCAAACAGG |

[a]The ASV sequence ID from the large international collection described in this study [b]Adjusted p-value for significance of water treatment in generalized linear model [c]Adjusted p-value for significance of polygenic risk score classification (genotype class) in generalized linear model [d]Adjusted p-value for significance of interaction term between treatment andpolygenic risk score genotype classification [e]Adjusted p-value for significance of water treatment in generalized linear model [f]Adjusted p-value for significance of cluster classification in large study [g]ASV sequence associated with sequence ID.

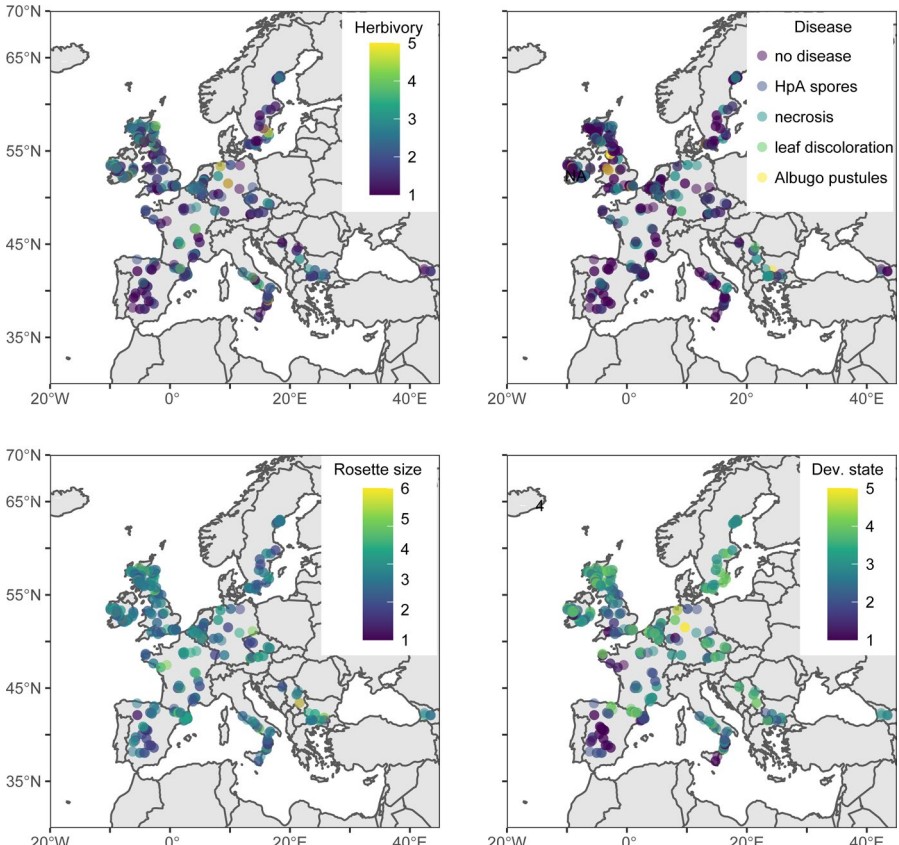

**Extended Data Fig. 1 | Distribution of sampled *A. thaliana* plants with various developmental and health states.** Arbitrary scales (see Methods) except for rosette size (cm).

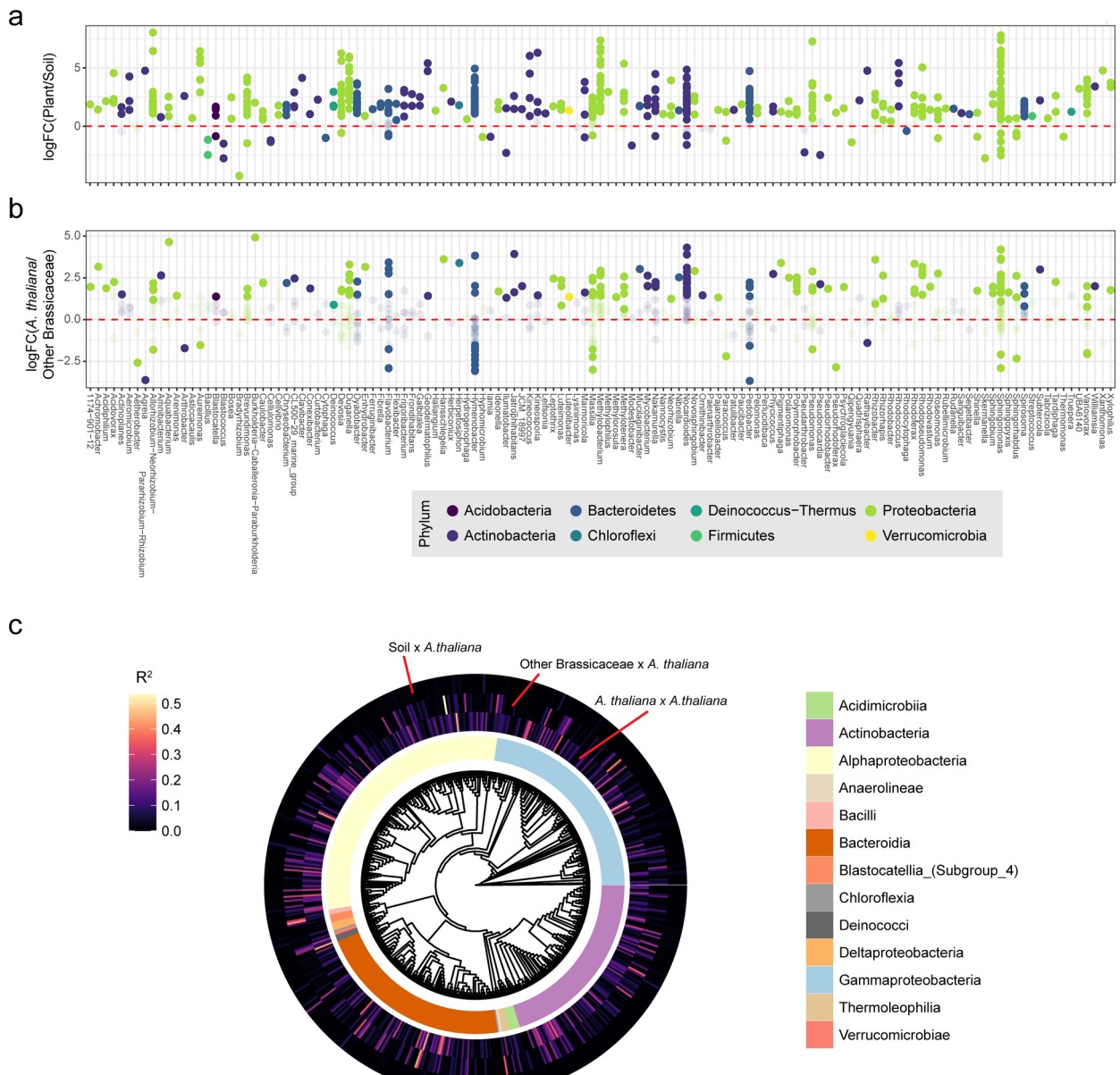

**Extended Data Fig. 2 | Differential abundance of phylotypes in soil, *A. thaliana* phyllospheres, and phyllospheres of other Brassicaceae. a**, 91% of phylotypes were differentially abundant between *A. thaliana* and soil. **b**, 36% of phylotypes were differentially abundant between *A. thaliana* and other Brassicaceae. Bold points indicate significance with an FDR ≤ 0.01. **c**, Within-site correlation of phylotype abundance. Correlation coefficients (scale on top left) were calculated for the co-occurrence of a phylotype within a site between the two *A. thaliana* plants collected at the site, *A. thaliana* x *A. thaliana* (third ring from the outside), other Brassicaceae x *A. thaliana* (second ring from the outside), and soil x *A. thaliana* (outermost ring). The central tree in the Circos plot represents the maximum likelihood tree of phylotypes, plotted without inferred branch lengths[75].

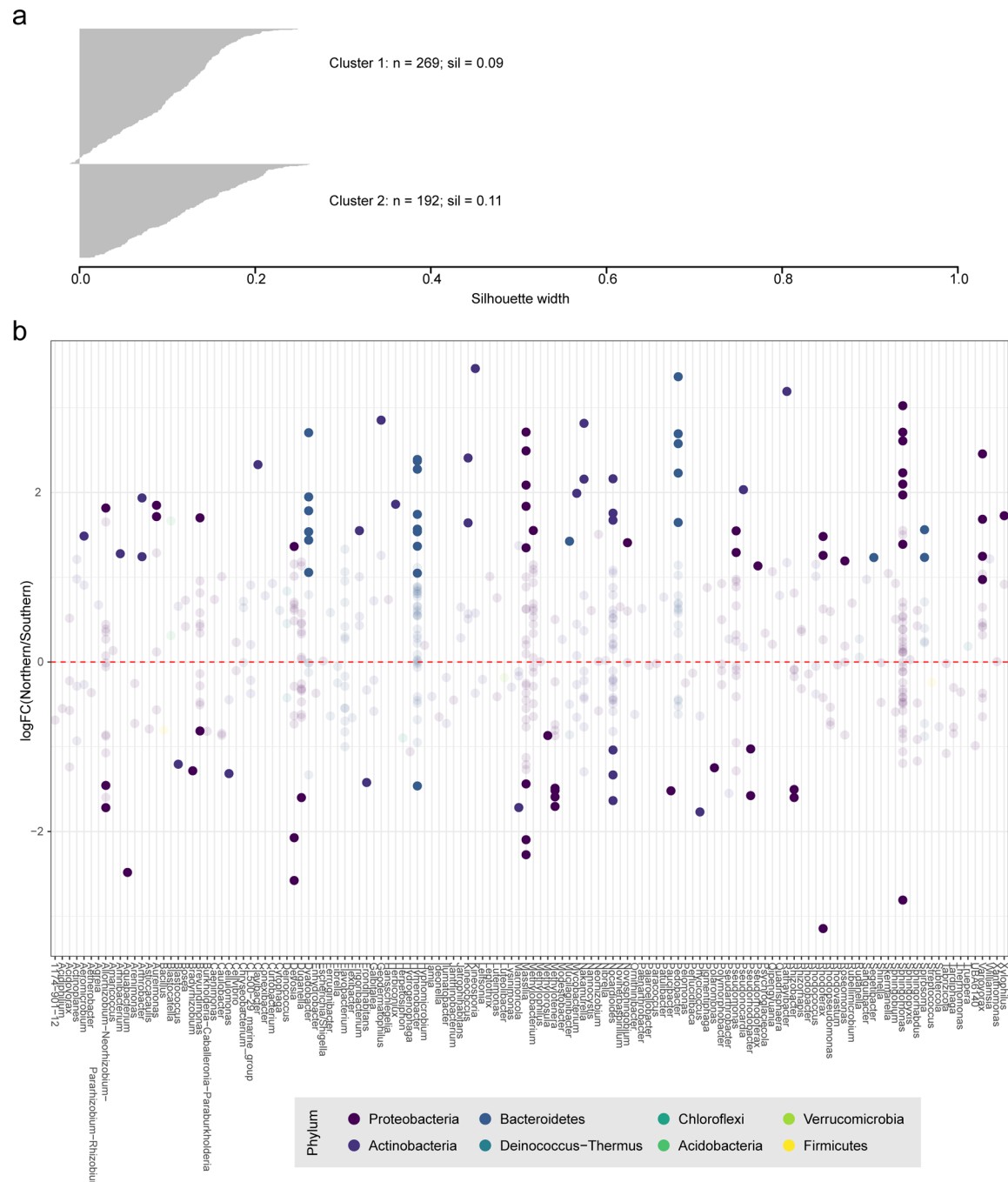

**Extended Data Fig. 3 | Contrasts in phylotype abundances between Southern and Northern microbiome clusters. a**, Silhouette scores for membership assignment to either of the two main microbiome types, cluster 1 (Southern) and cluster 2 (Northern). For each cluster, number of individuals and average distance between a sample and members of the other cluster is indicated. **b**, Differential abundance of phylotypes between Southern and Northern microbiome clusters. y-axis shows the $\log_2$(Fold Change) for the relative abundance difference of a phylotype between clusters. Bold points indicate significance with an FDR ≤ 0.01.

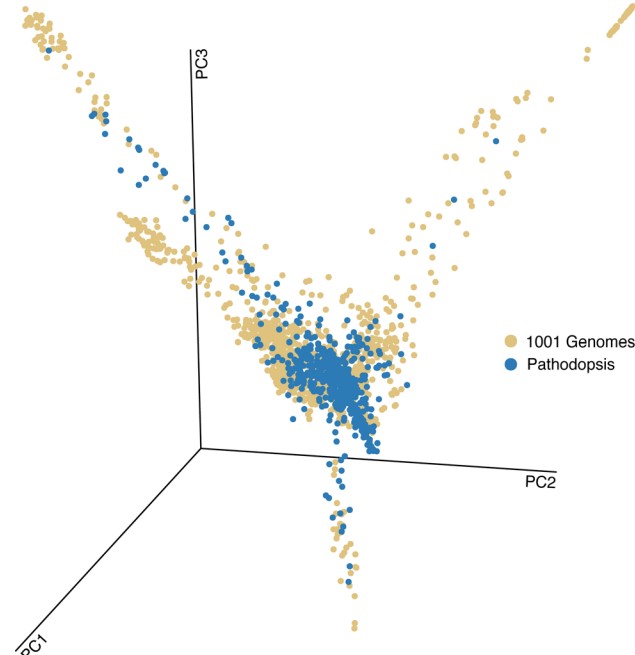

**Extended Data Fig. 4 | Projection of *A. thaliana* genotypes from this study into genotypic PC space from the 1001 Genomes Project.** Individuals from this study ('Pathodopsis') align well with the broader 1001 Genomes (https://1001genomes.org) collection of primarily Eurasian accessions.

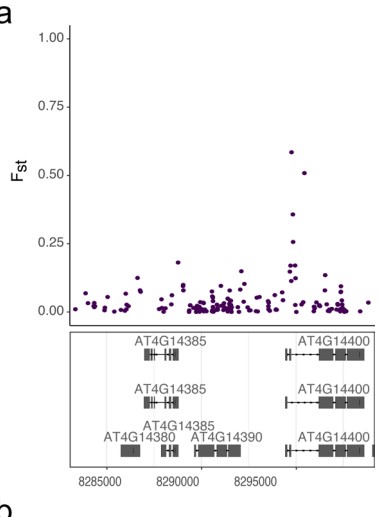

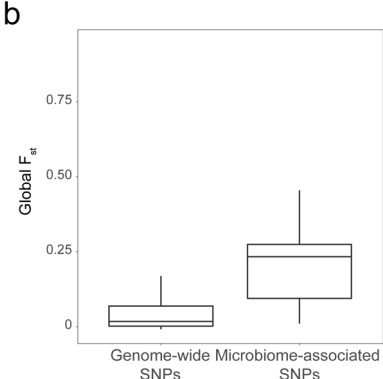

**Extended Data Fig. 5 | F$_{st}$ around *ACD6* and globally. a**, Cockerham and Weir's fixation index F$_{st}$ was estimated for SNPs in a list of known immune-associated genes. The genome-wide most extreme F$_{st}$ values are concentrated in a region on chromosome 4 that includes the immune regulator *ACD6* (At4g14400). Reference genome positions (in nt) on chromosome 4 given at the bottom. **b**, Bergelson and colleagues[32] identified *A. thaliana* SNPs associated with (and likely to influence) microbiome composition. We compared the geographic differentiation of these SNPs (F$_{st}$) to the genome-wide distribution. Microbiome-associated SNPs exhibit significantly higher F$_{st}$ values than the remainder of the genomic SNPs (Wilcoxon rank-sum test $p < 2.2 \times 10^{-16}$). The central horizontal line in each box indicates the median, the bounds indicate the upper and lower quartiles and the whiskers indicate 1.5*inner quartile range.

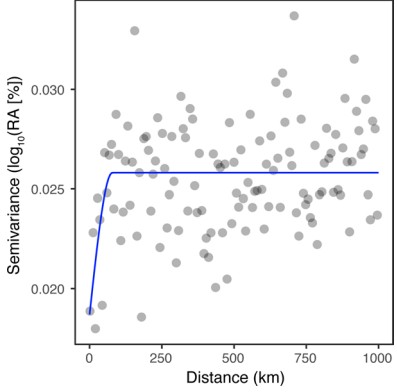

**Extended Data Fig. 6 | Distance-Semivariance plot for ATUE5.** Relationship between the geographic distance between two sampled *A. thaliana* plants, and the correlation of the abundance of ATUE5 between these two plants.

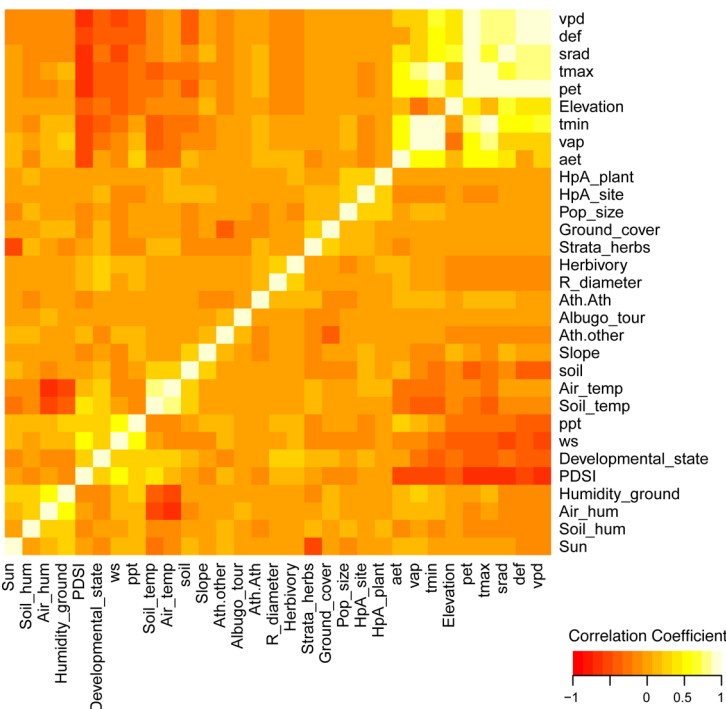

**Extended Data Fig. 7 | Correlogram of relationship between environmental and developmental covariates used in random forest modeling.** Covariates are detailed in Methods.

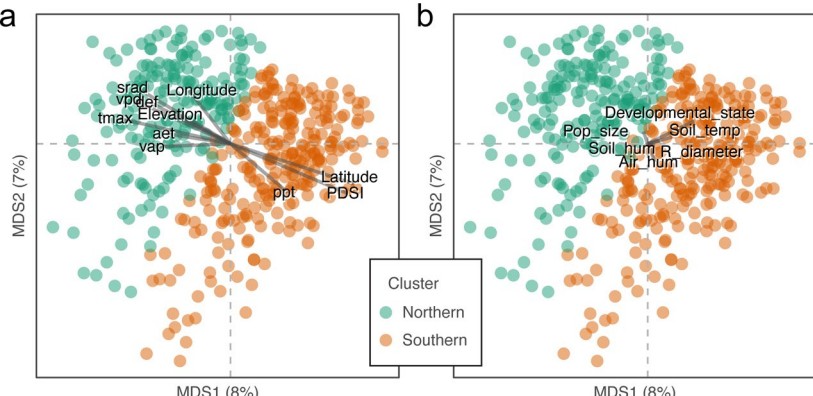

**Extended Data Fig. 8 | Biplots of the correlation of environmental and physiological variables on the MDS axes in Fig. 2. a**, Environmental variables derived from Terraclimate. **b**, Environmental variables measured at time of collection. Correlations were assessed with the envfit() function in vegan, and vector length corresponds to strength of correlation. Long-term climate variables (**a**) are better predictors of microbiome composition than are more temporary weather and physiological variables measured at the time of collection (**b**).

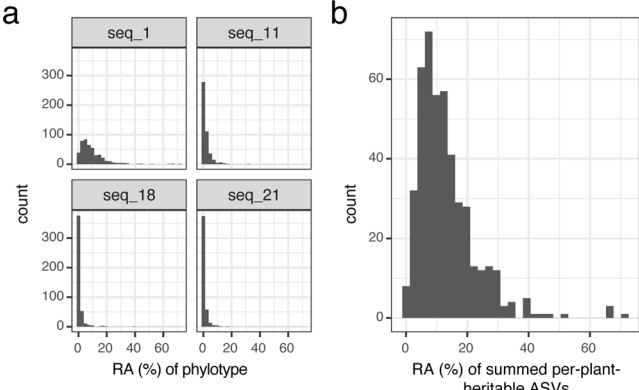

**Extended Data Fig. 9 | Relative abundance of phylotypes is significantly associated with plant genotype classification.** Four out of 20 phylotypes that were shared between the Eurasian collections and California field experiment were significantly associated with plant genotype. **a**, Histograms for the relative abundance of each of the four significant phylotypes across all plants collected in Eurasia. **b**, Histogram of the total relative abundance per plant of the sum of all four phylotypes (mean = 13.2% RA).

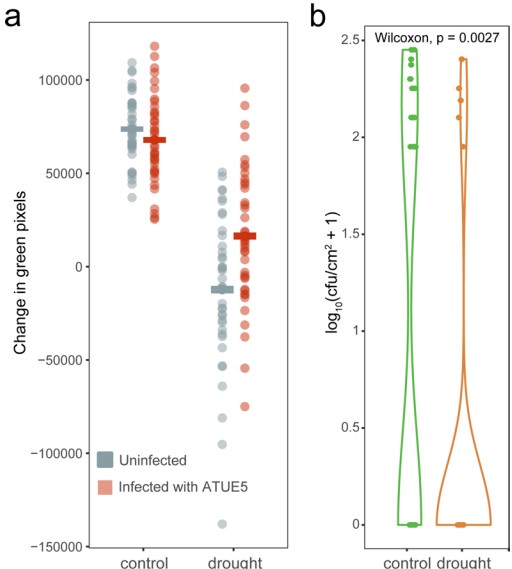

**Extended Data Fig. 10 | Impact of a common phylotype on plant growth as a function of drought status.** *Arabidopsis thaliana* plants were exposed to combinations of drought and infection with ATUE5 strain p25.c2. **a**, The change in plant leaf area from day 0 to day 10 as calculated based on daily images and extracting green pixels from images. Infection with ATUE5 reduced the negative effect of drought on plant growth (ANOVA, p = 0.0063 in ANOVA). **b**, measured colony forming units (cfu) on day 3 post infection. 5/41 (12%) drought-treated plants had established infection on day 3, whereas 17/23 (42%) of control plants had established infection at this same timepoint (Wilcoxon rank-sum p = 0.0027).

# nature research

# Reporting Summary

Nature Research wishes to improve the reproducibility of the work that we publish. This form provides structure for consistency and transparency in reporting. For further information on Nature Research policies, see our Editorial Policies and the Editorial Policy Checklist.

## Statistics

For all statistical analyses, confirm that the following items are present in the figure legend, table legend, main text, or Methods section.

| n/a | Confirmed | |
|---|---|---|
| ☐ | ☒ | The exact sample size (*n*) for each experimental group/condition, given as a discrete number and unit of measurement |
| ☐ | ☒ | A statement on whether measurements were taken from distinct samples or whether the same sample was measured repeatedly |
| ☐ | ☒ | The statistical test(s) used AND whether they are one- or two-sided *Only common tests should be described solely by name; describe more complex techniques in the Methods section.* |
| ☐ | ☒ | A description of all covariates tested |
| ☐ | ☒ | A description of any assumptions or corrections, such as tests of normality and adjustment for multiple comparisons |
| ☐ | ☒ | A full description of the statistical parameters including central tendency (e.g. means) or other basic estimates (e.g. regression coefficient) AND variation (e.g. standard deviation) or associated estimates of uncertainty (e.g. confidence intervals) |
| ☐ | ☒ | For null hypothesis testing, the test statistic (e.g. $F$, $t$, $r$) with confidence intervals, effect sizes, degrees of freedom and $P$ value noted *Give P values as exact values whenever suitable.* |
| ☒ | ☐ | For Bayesian analysis, information on the choice of priors and Markov chain Monte Carlo settings |
| ☒ | ☐ | For hierarchical and complex designs, identification of the appropriate level for tests and full reporting of outcomes |
| ☐ | ☒ | Estimates of effect sizes (e.g. Cohen's *d*, Pearson's *r*), indicating how they were calculated |

*Our web collection on statistics for biologists contains articles on many of the points above.*

## Software and code

Policy information about availability of computer code

| Data collection | usearch (version 11), DADA2 (version 1.10.1), DECIPHER (version 2.8.1), gemma (version 0.98.3),  GATK (version 3. 5), SEEKIN (version 3.0), R packages used: fpc (version 2.2.9), cluster (version 2.1.2), MuMIn (version 1.43.1), EIGENSOFT (version 6), edgeR (version 3.28.1), caret47 (version 6.0-86), automap (version 1.0-14), fasttreeMP (version 2.1.11) |
|---|---|
| Data analysis | Scripts for data processing, analyses and figure generation can be accessed at https://github. com/tkarasov/pathodopsis. |

For manuscripts utilizing custom algorithms or software that are central to the research but not yet described in published literature, software must be made available to editors and reviewers. We strongly encourage code deposition in a community repository (e.g. GitHub). See the Nature Research guidelines for submitting code & software for further information.

## Data

Policy information about availability of data

All manuscripts must include a data availability statement. This statement should provide the following information, where applicable:
 - Accession codes, unique identifiers, or web links for publicly available datasets
 - A list of figures that have associated raw data
 - A description of any restrictions on data availability

v3-v4 16S rDNA sequence data were deposited in the European Nucleotide Archive (ENA) under the Primary Accession ENA: PRJEB44379. Metadata and processed read data sets including phyloseq objects are available at Zenodo under DOI 10.5281/zenodo.5140512. The Silva database is available at https://www.arb-silva.de/ documentation/release-132/.

# Field-specific reporting

Please select the one below that is the best fit for your research. If you are not sure, read the appropriate sections before making your selection.

☐ Life sciences          ☐ Behavioural & social sciences          ☒ Ecological, evolutionary & environmental sciences

For a reference copy of the document with all sections, see nature.com/documents/nr-reporting-summary-flat.pdf

# Ecological, evolutionary & environmental sciences study design

All studies must disclose on these points even when the disclosure is negative.

| | |
|---|---|
| Study description | A. thaliana plants (and their associated microbiomes) were collected across Europe |
| Research sample | The research sample was the rosette of Arabidopsis thaliana (processed for host genotype and microbiome composition). |
| Sampling strategy | Previous work characterized the population structure and distribution of A. thaliana in Europe (Platt et al. 2010, TG Consortium, 2016). We sought |
| Data collection | Each collection team went to a specific region in Eurasia, and followed a systematic pipeline for sterile collection of plant tissue and processing. |
| Timing and spatial scale | Samples were collected in Winter-Spring 2018 across Europe |
| Data exclusions | Samples were excluded from analysis if we could not verify that the host of isolation was A. thaliana or Brassicaceae, or if the read output did not reach the specified standard of 1,000 16S rDNA reads per sample. |
| Reproducibility | Significant results were tested over several spatial scales, and confounding covariates were included in regression models (such as collection group or plate for processing). |
| Randomization | Samples were collected in regions by specific collection groups. While this confounds a perfectly random collection design, we conducted 17 collection trips and are able to include collection trip as a cofactor in models. |
| Blinding | Investigators had no prior knowledge of the distribution of microbes prior to this study. |

Did the study involve field work?     ☒ Yes     ☐ No

## Field work, collection and transport

| | |
|---|---|
| Field conditions | These parameters differed across field sites (information provided in metadata supplementary material) |
| Location | We collected from several hundred locations. Information provided in metata files. |
| Access & import/export | Respective national authorities of all sampled countries Party to the Nagoya Protocol were contacted ahead of collections. Where needed, advised measures were taken and resulted in sampling and export permit KC3M-160/11. 04. 2018 (Bulgaria), ABSCH-IRCC-FR-253846-1 (France) and ABSCH-IRCC-ES-259169-1 (Spain). |
| Disturbance | Only the soil within 5cm of the plant was collected with the plant. |

# Reporting for specific materials, systems and methods

We require information from authors about some types of materials, experimental systems and methods used in many studies. Here, indicate whether each material, system or method listed is relevant to your study. If you are not sure if a list item applies to your research, read the appropriate section before selecting a response.

## Materials & experimental systems

| n/a | Involved in the study |
|---|---|
| ☒ | ☐ Antibodies |
| ☒ | ☐ Eukaryotic cell lines |
| ☒ | ☐ Palaeontology and archaeology |
| ☒ | ☐ Animals and other organisms |
| ☒ | ☐ Human research participants |
| ☒ | ☐ Clinical data |
| ☒ | ☐ Dual use research of concern |

## Methods

| n/a | Involved in the study |
|---|---|
| ☒ | ☐ ChIP-seq |
| ☒ | ☐ Flow cytometry |
| ☒ | ☐ MRI-based neuroimaging |

