## [Peer Review File · Nature Microbiology]

Peer Review Information

Journal: Nature Microbiology

Manuscript Title: Drought selection on Arabidopsis populations and their microbiomes

Corresponding author name(s): Dr Detlef Weigel

Editorial Notes:

This manuscript has been previously reviewed at another journal. This document only contains reviewer comments, rebuttal and decision letters for versions considered at Nature Microbiology. Mentions of prior referee reports have been redacted

Reviewer Comments & Decisions:

Decision Letter, initial version:

Message: 16th June 2022

Dear Detlef,

Thank you for your patience while your manuscript "Drought selection on Arabidopsis populations and their microbiomes" was under peer-review at Nature Microbiology. It has now been seen by 3 referees, whose expertise and comments you will find at the end of this email. Although they find your work of some potential interest, they have raised a number of concerns that will need to be addressed before we can consider publication of the work in Nature Microbiology.

In particular, referees #1 and #3 both ask for additional experimental evidence to support the main conclusion. These referees give some suggestions to test causation. We also asked referee #2 for further comments and specific advice regarding the suggestion from referees #1 and #3 for an additional causation experiment. They said "The causation experiment is feasible but not easy (establishing microbiome inoculation system in the lab needs a lot of time and experience). Two major points may be clarified by experiments : 1) dissecting influences on leaf microbiome by environmental factors and plant genetics; 2) demonstrating the function of certain leaf bacteria in Arabidopsis drought tolerance. For point 1, if the authors did not harvest Arabidopsis seeds when collecting leaf samples, they could perform the experiment using representative ecotypes (e.g. 10~20) across longitudes from the Arabidopsis resources in Weigel lab. For point 2, the authors could examine Arabidopsis growth (one or two ecotypes) with or without bacterial inoculation (e.g. ASV5). If the authors could manage one of these experiments, the manuscript would be much stronger." In addition, the referees ask for some additional bioinformatics analyses and some rewriting of the paper to make it more focused.

Should further experimental data allow you to address these criticisms, we would be happy to look at a revised manuscript.

We strongly support public availability of data. Please place the data used in your paper into a public data repository, if one exists, or alternatively, present the data as Source Data or

2Supplementary Information. If data can only be shared on request, please explain why in your Data Availability Statement, and also in the correspondence with your editor. For some data types, deposition in a public repository is mandatory - more information on our data deposition policies and available repositories can be found at <https://www.nature.com/nature-research/editorial-policies/reporting-standards#availability-of-data>.

Please include a data availability statement as a separate section after Methods but before references, under the heading "Data Availability". This section should inform readers about the availability of the data used to support the conclusions of your study. This information includes accession codes to public repositories (data banks for protein, DNA or RNA sequences, microarray, proteomics data etc...), references to source data published alongside the paper, unique identifiers such as URLs to data repository entries, or data set DOIs, and any other statement about data availability. At a minimum, you should include the following statement: "The data that support the findings of this study are available from the corresponding author upon request", mentioning any restrictions on availability. If DOIs are provided, we also strongly encourage including these in the Reference list (authors, title, publisher (repository name), identifier, year). For more guidance on how to write this section please see:

<http://www.nature.com/authors/policies/data/data-availability-statements-data-citations.pdf>

* If you have not done so already we suggest that you begin to revise your manuscript so that it conforms to our Article format instructions at <http://www.nature.com/nmicrobiol/info/final-submission>. Refer also to any guidelines provided in this letter.

When submitting the revised version of your manuscript, please pay close attention to our [href="https://www.nature.com/nature-portfolio/editorial-policies/image-integrity">Digital Image Integrity Guidelines](https://www.nature.com/nature-portfolio/editorial-policies/image-integrity). and to the following points below:

Note: This url links to your confidential homepage and associated information about manuscripts you may have submitted or be reviewing for us. If you wish to forward this e-mail to co-authors, please delete this link to your homepage first.

Nature Microbiology is committed to improving transparency in authorship. As part of our efforts in this direction, we are now requesting that all authors identified as 'corresponding author' on published papers create and link their Open Researcher and Contributor Identifier (ORCID) with their account on the Manuscript Tracking System (MTS), prior to acceptance. This applies to primary research papers only. ORCID helps the scientific community achieve unambiguous attribution of all scholarly contributions. You can create and link your ORCID from the home page of the MTS by clicking on 'Modify my Springer Nature account'. For more information please visit www.springernature.com/orcid.

If you wish to submit a suitably revised manuscript we would hope to receive it within 6 months. If you cannot send it within this time, please let us know. We will be happy to consider your revision, even if a similar study has been accepted for publication at Nature Microbiology or published elsewhere (up to a maximum of 6 months).

Yours sincerely,

Reviewer Expertise:

Referee #1: plant microbiome, multi-omics, bioinformatics

Referee #2: plant-microbe interactions, multi-omics

Referee #3: plant genetics and microbiome

Reviewer Comments:

Reviewer #1 (Remarks to the Author):

Karasov and colleagues present in this manuscript a continental-scale characterization of the leaf microbiome of 300 *Arabidopsis thaliana* populations across Europe. They relate changes in community composition estimated from 16S rRNA gene amplicon profiling to variation in host genetics and environmental factors. They identify latitude and drought-associated readouts as the factors with the highest explanatory power, tightly correlated with the

4abundances of some (but not all) of the most abundant bacterial phylotypes (ASVs). Although previous studies have explored the variation in plant microbiome composition across populations, the depth and breadth of the dataset presented, as well as some potentially interesting new findings, raises the interest of this manuscript. As far as I can judge, the data analysis and interpretation are more than appropriate, and I think the authors should be praised for making code and intermediate data publicly available at this stage. Although I can clearly see the relevance of this manuscript, I have a lingering concern that this study might represent only an incremental step towards improving our understanding of microbiome variation in nature and its link to host genetics. Below I list a few points for the authors to consider and suggest an additional experiment which could help address this concern and that I believe might improve the study.

Major points:

1. The authors find that, among abundant bacterial phylotypes, some of them appear to have a patchy distribution, while others correlate with climate and geographical variables such as latitude or PDSI. One possible interpretation is that the first group of bacteria are differentially selected by genetic variation in immune genes, which are not geographically structured (e.g., ATU5). In contrast, members of the second group (e.g., Sphingomonadaceae or Rhizobiaceae) may be selected by environmental variation directly, or indirectly by geographically-structured and climate-driven variation in host genetics.

The authors do an excellent job in presenting these alternative hypotheses, but ultimately the answer is unsatisfactory since the data from this continental survey cannot differentiate between a direct impact of climate on abundant, not pathogenic microbes, and the effect of climate-driven variation in host genetics (e.g., drought). To address this important question, perhaps the authors could consider a greenhouse experiment, where a few ecotypes from representative populations along a latitude/PDSI gradient are grown under controlled conditions and drought treatment. These plants can be inoculated with an ATU5 and/or an Sphingomonas isolate(s) from an abundant phylogroup that shows latitudinal variation (or a close relative from the *A. thaliana* leaf culture collection if not available). If the final abundances of the Sphingomonas isolate(s) show variation across ecotypes in the control condition, an effect of host genetics is likely. If they only show variation in comparison with the drought-treated plants, this could be taken as evidence of direct influence of the environment on the microbes, independently of the host.

2. I find the clustering into $k=2$ 'microbiome types' somewhat problematic. Firstly, I think that clustering approaches that use pairwise distances or dissimilarities are better suited for compositional data than methods that need an embedding into a metric space such as k -means. Why do k -means on projected coordinates in 2 components (which capture only a fraction of all variance) instead of simply doing hierarchical clustering on the pairwise dissimilarity matrix? Secondly, it is not at all clear to me that $k=2$ is a good choice, by looking at Fig. 2c and Fig. S4, the latter of which I don't find very informative. What are the silhouette scores for $k=3, 4, \dots$? I understand one of the main reasons for this analysis is to map Fig. 2c onto 2d, but this could be easily done by coloring 2c according to latitude, for instance. I think fitting the data into two microbiome types is rather artificial, given what looks to me like a rather smooth distribution of samples along MDS1. I would worry that the authors might be repeating the same error made in gut microbiota field, when the concept of

5discrete 'enterotypes' was introduced (Arumugam, et al., 2011), and then proven to be somewhat flawed years later, when more data showed a smooth gradient in the abundances a small subset of key taxa, rather than well-defined clusters (Koren et al., 2013). My suggestion would be to remove this clustering analysis altogether, color Fig. 2c according to latitude (or show a biplot), and perhaps use hierarchical clustering as an alternative (e.g., on top of Fig. 2e).

Minor points:

3. Soil pH doesn't seem to be among the climate variables used for the random forest classification and feature ranking. However, pH was reported to be one of the largest contributors to (root) microbiota variation across European *A. thaliana* populations (Thiergart et al., 2020). Is this also an important factor for predicting leaf community composition?
4. Abstract: "52-68% of variance in the first two principal coordinates of microbiome type explained by host genotype" If I get this right, this is a percentage of a percentage – what is the fraction of overall variance explained by genotype?
5. "500 major bacterial phylotypes" – use the term 'amplicon variant' or define 'bacterial phylotype' the first time is used. I was unsure at this point of whether any clustering or abundance filtering had been done to group sequences into phylotypes.
6. The first four paragraphs of the 'Results' section read more as an Introduction. The results should probably start in the middle of page three instead.
7. Page 3, paragraph 4: "We assessed the microbial composition of the leaf and soil samples by sequencing the v3-v4 region of the 16S rRNA locus (hereafter rDNA) and classifying distinct 16S sequences as alternative sequence variants (ASV) using DADA2 (ref. 12)" As far as I know, ASV stands for 'amplicon sequence variant' (not 'alternative'). Also, "ref. 12" can be removed.
8. Page 4: "Modeling the effect of host versus soil on the abundance of core microbial phylotypes revealed that 91% (524/575) of phylotypes were differentially abundant between the *A. thaliana* phyllosphere and soil". Surely the analysis yielded more than 575 ASVs and these are abundant or prevalent ones? If so, this might be good to mention when defining how the authors use the term bacterial phylotype. In addition, perhaps it would clearer to write "modeling the effect of compartment" or "microhabitat" or "leaf versus soil" rather than "host versus soil".
9. Figure S3: This is a very minor issue but I would say that phylogenies built from partial DNA sequences such as this one should not be ultrametric, unless a model with a clock has been used (which I don't think was the case here).
10. Page 5: "indicating extensive filtering of microbes that colonize the plant, whether from soil or from another environmental source" and again in the discussion "colonization of *A. thaliana* leaves serves as a strong filter from the surrounding environment". With the current data one can only conclude this for neighboring soil and plant species, not for other environmental sources (from which they might be less or partial filtering) such as

6root/rhizosphere, leaves of nearby trees, rainwater, etc. In fact, roots might be an important environmental reservoir of leaf-colonizing bacteria (see for example the observation from Wagner et al., 2016, that a significant fraction of the leaf microbiota of another Brassicaceae is found in roots of the same plant).

11. Figure 2: what dissimilarity measure was used for these beta-diversity analyses? Not stated in the figure caption or methods, as far as I can tell.

12. Fig. 3a. Also Rhizobiaceae, an abundant and ubiquitous bacterial taxon in the leaf, seems to have a relatively strong latitude component. I find this intriguing, but not mentioned in the text.

13. The analysis based on Fst in gene alleles linked to immunity is very interesting, and the fact that the top SNPs were identified in the ACD6 gene I guess is reassuring. Perhaps a similar analysis on drought-related genes might be relevant as well?

14. Page 6: "While ATUE5 was ubiquitously present, its distribution was not uniform." It is not clear from Fig. 3c that this ASV was ubiquitous. To me, it appears almost completely absent from Spain, England and southern Italy.

15. Analysis of climate variables as predictors of community composition. I think the random forest analysis (e.g, Fig. 4a) is very informative but I think biplots on a projection such as the one shown on Fig. 2c might be useful as well.

16. Page 11 (methods): "918 samples had a sufficient number of reads". How many reads was considered a sufficient number?

Reviewer #2 (Remarks to the Author):

The manuscript by Karasov et al investigated the composition and driving force of the leaf microbiome of *Arabidopsis thaliana* populations in their native sites across Europe. First, the authors found a remarkable pattern that the leaf microbiomes separated along a latitude gradient, southern and northern parts of the European continent. Second, association analysis revealed that host genetics, especially SNPs of an immune regulator ACD6 in *Arabidopsis* correlated with leaf microbiome composition. In the end, the statistical model revealed that drought indices were the strongest predictors for leaf microbiome, but not for soil microbiome. This is a comprehensive sequencing study covering diverse environmental and plant genetic factors. Although it is a pity that the authors did not show functions leaf microbiome in *Arabidopsis* drought tolerance, the continent-scale data alone provide valuable and solid insight on leaf microbiome assembly and potential co-adaption of *Arabidopsis* accessions and leaf microbiome. However, several aspects are needed to be further explored or clarified. My concerns are as below:

1) it is unclear why the microbiome clusters along latitude showed taxonomic patterns only at the phylotype (ASV) level, but not at higher levels (family or genus). The Southern cluster and Northern clusters significantly separate in the first dimension of MDS analysis (Figure

72c), which is very likely to be driven by a group of bacteria in closely related taxa (e.g. *Bacteroides* and *Prevotella* genera determine the enterotype in human gut) or some bacterial ASVs with high-abundance. The authors need to perform the association in Figure 2e at the level of family or genus, and showed all biomarker ASVs to distinguish the Southern and Northern clusters. The results would not only help to explain the data, but also serve as the basis to speculate or test leaf microbiome's function for *Arabidopsis* drought tolerance.

2) A large proportion of the manuscript shows data on distribution analysis of *Arabidopsis* pathogens and association of immune genes with leaf microbiome composition, but the results do not explain host genetic regulation on leaf microbiome clustering on latitude, because common *Arabidopsis* pathogens are patchily distributed and do not correlated with latitude. I would suggest the authors explore the data with more thoughts on commensals. The interactions between commensals and host plants are key questions in microbiome study, and may drive the plant evolution. The authors could search the association of latitude with genes that are reported to regulate *Arabidopsis* leaf or root commensal bacteria, which may improve the result of host genetic regulation on leaf microbiome in this large-scale data.

3) In the last part of the story, the authors focus on drought indexes PDSI as the strongest predictors of leaf microbiome composition. What about others, such as *srad* and *vpd*? They also show a considerable contribution in the statistic model (Figure 4a). I understand that it may be easy to focus on the strongest predictor, but it is also very important show a model using a few (e.g. three) predictors to obtain a largely correct leaf microbiome. I think, this is the power of such large-scale data. It would be great if the authors could extend the modeling part of the manuscript, and explain the influence of environmental factors on leaf microbiome in a more systematical way.

4) The authors should test the association of leaf microbiome with drought-related SNPs with their geographic distribution. Since environmental factors are strong drivers for both microbiome and plant genetics, it is likely that drought-related genes evolve with the environment and plant microbiome. A similarly analysis with the immune genes would be helpful to elucidate potential co-adaptation of *Arabidopsis* genomes and leaf microbiome under variable drought conditions in nature.

Reviewer #3 (Remarks to the Author):

The manuscript by Karasov et al. describes the results of a large-scale sampling of *Arabidopsis thaliana* rosette microbiomes across Europe. A strong latitudinal pattern was observed, where microbiome composition clustered into two distinct groups roughly corresponding to "north" and "south". Multiple environmental variables associated with each sample were either measured directly or acquired from databases, and a few plant traits (size, disease status, herbivory status) were also estimated. Shotgun sequencing reads were also used to characterize plant genotype. Overall this is a rich dataset with high potential value.

While the sampling effort and some of the analyses were quite impressive, and the paper was well-written and presented, in my opinion the data do not sufficiently support the authors' conclusion stated in the title, that drought pressure has been an agent of natural selection

8shaping the phyllosphere microbiome. The paper demonstrates correlations between several sets of variables - microbiome composition, latitude, environment (esp. drought), and host genotype - but ultimately, causal relationships between these factors cannot be identified from this type of sampling design. Instead, the data simply demonstrate that all of these things are spatially structured by latitude - a pattern with numerous possible alternative explanations. To convincingly show a role for leaf microbiomes in host adaptation to drought, I would expect to see this dataset bolstered by manipulative experiments of some kind, such as a greenhouse experiment with inoculated and uninoculated plants under simulated drought. Fitness measurements and/or much more thorough population genomic analyses are needed to invoke natural selection as the cause of the observed patterns.

Even the claim that host genotype is a major factor shaping microbiome composition, is difficult to defend without a way to eliminate the environmental contribution. I was surprised that the authors did not attempt to fit models of leaf microbiome composition that included latitude, environmental variables, and host genotype as predictors, simultaneously. This model would have been very informative, and increased my confidence that some or all of these factors have a marginal effect on the microbiome. Instead there is a series of separate analyses using subsets of these variables, such that it is not clear whether any of them has an effect independent of the others.

Another criticism is that the paper is unfocused. Ostensibly the paper is about the leaf microbiome's role in plant adaptation to drought. However, a very large proportion of the paper actually is focused on spatial distribution of pathogens (ATUE5), disease, and immunity. Granted, allelic variation at immunity genes such as ACD6 may well be important for microbiome composition. Nevertheless, this paper contains no actual experimental evidence of causal links between disease, drought, and leaf microbiomes.

Finally, it is unclear what contribution this paper makes to the broader field. What do these results teach us about the role of leaf microbiomes in adaptation to drought, or even plant-microbiome interactions in general? References to the pre-existing literature are quite sparse. There is no discussion of possible mechanisms linking phyllosphere microbes to drought resistance, in any direction of causality. It is unclear if links between pathogens, drought, and phyllosphere microbiome have been reported in other systems.

Other comments:

The dataset includes information about microbial load, but it is unclear how this was used.

It is also unclear how the plant phenotype data were used - for example, developmental stage was recorded but not used as a predictor or covariate in the models. Other studies have shown strong changes in leaf microbiomes with developmental stage. Is it possible that genetic variation for developmental rate is at the core of observed correlations between genotype and microbiome?

Is PDSI during the 6 months prior to collection a reliable proxy for drought pressure over evolutionary time (which is when genomic signatures of plant-microbiome adaptation would have been shaped)?

What value is added by repeating analyses using logistic regression with microbiome clusters, and also linear regression with latitude?

“Previous work on non-host associated soils supports the importance of water availability in determining microbiome composition” Why was the effect of drought on soil microbiomes negligible in this study?

“To probe for possible relative contributions of genotype and drought, we performed mixed effect modeling and estimated the marginal R² for PDSI to be 50%.” And what about the marginal R² for genotype? Earlier in the paper, genotype was stated to be the dominant force shaping the leaf microbiomes (>50% of variance)?

“Drought might do so directly, by affecting the physiological state of the plant, indirectly by shaping host genetics, or both.” This dataset could have been used to investigate whether the physiological state of the plant is a more proximate cause of microbiome variation - by including stage, size, etc. as predictors in a model.

“To determine the relative effect sizes of drought, latitude and plant identity on MDS loadings, phenotypes were modeled with the gaston package in R. The model was $Y_i \sim PDSI_i + Lati + k_{i,j} + \epsilon_i$ where Y_i is the phenotype for each i th accession, $k_{i,j}$ is the genetic relatedness between the i th and j th accessions...” I am very confused by this model. The response variable (MDS loading) is solely a property of accession i . Yet, one of the predictors describes the genetic relationship between accessions i and j . How was accession j chosen or defined?

Author Rebuttal to Initial comments

Dear Detlef,

10Thank you for your patience while your manuscript "Drought selection on Arabidopsis populations and their microbiomes" was under peer-review at Nature Microbiology. It has now been seen by 3 referees, whose expertise and comments you will find at the end of this email. Although they find your work of some potential interest, they have raised a number of concerns that will need to be addressed before we can consider publication of the work in Nature Microbiology.

In particular, referees #1 and #3 both ask for additional experimental evidence to support the main conclusion. These referees give some suggestions to test causation. We also asked referee #2 for further comments and specific advice regarding the suggestion from referees #1 and #3 for an additional causation experiment. They said "The causation experiment is feasible but not easy (establishing microbiome inoculation system in the lab needs a lot of time and experience). Two major points may be clarified by experiments : 1) dissecting influences on leaf microbiome by environmental factors and plant genetics; 2) demonstrating the function of certain leaf bacteria in Arabidopsis drought tolerance. For point 1, if the authors did not harvest Arabidopsis seeds when collecting leaf samples, they could perform the experiment using representative ecotypes (e.g. 10~20) across longitudes from the Arabidopsis resources in Weigel lab. For point 2, the authors could examine Arabidopsis growth (one or two ecotypes) with or without bacterial inoculation (e.g. ASV5). If the authors could manage one of these experiments, the manuscript would be much stronger." In addition, the referees ask for some additional bioinformatics analyses and some rewriting of the paper to make it more focused.

Should further experimental data allow you to address these criticisms, we would be happy to look at a revised manuscript.

Thank you for the useful comments and critiques. We have added experimental data in line with experiments recommended in the reviews as well as synthesis with experimental results from two other studies of European *A. thaliana*. Together, these data indicate that *A. thaliana* natural genetic variation influences the majority of the ASVs we survey in our study, and that one of the most abundant ASVs (ASV5 as recommended) affects *A. thaliana*'s tolerance of drought. We address the comments in greater detail below, and will summarize the major changes briefly here.

With our revisions to the manuscript, we aimed to add several pieces of information. Firstly, we aimed to test whether the ASVs we identified in our study were heritably controlled in *A. thaliana*. Secondly, we aimed to explicitly test whether drought influenced the abundance and effect of a common ASV.(1) For the first aim, we ascertained heritability estimates for the ASVs in our study by identifying the matching ASVs in a recent heritability study of *A. thaliana* leaf microbiomes carried out at four field sites near natural *A. thaliana* populations (Brachi et al. PNAS 2022). We found that 64% of our 575 ASVs had a match in the Brachi study. Using the published field data, we obtained heritability estimates for 251 ASVs. Of these, almost all (247, 98.4%) were heritable in at least one of the four common gardens. While probably only few respond directly to *A. thaliana* genetics (and the rest are indirectly influenced by these hub microbes), these results clearly indicate that *A. thaliana* has the capacity to affect the core phyllosphere microbiome variation we observe across Europe.

(2) We also tested how drought affects the growth of the common pseudomonad ASV5 (ATUE5). We conducted a controlled drought experiment in growth chambers in which we exposed *A. thaliana* to drought and performed inoculations of the drought-stressed plants. We found that drought reduces the capacity of the ATUE5 strain to grow *in planta*, and also reduces the negative effect of ATUE5 on plant growth and health, suggesting that the fitness effects of ATUE5 likely differ between European populations since the likelihood of drought greatly differs across Europe.

Altogether, we are hopeful that the revisions and additional experiments sufficiently address the comments and suggestions of the reviewers.

Reviewer Expertise:

Referee #1: plant microbiome, multi-omics, bioinformatics

Referee #2: plant-microbe interactions, multi-omics

Referee #3: plant genetics and microbiome

Reviewer Comments:

Reviewer #1 (Remarks to the Author):

Karasov and colleagues present in this manuscript a continental-scale characterization of the leaf microbiome of 300 *Arabidopsis thaliana* populations across Europe. They relate changes in community composition estimated from 16S rRNA gene amplicon profiling to variation in host genetics and environmental factors. They identify latitude and drought-associated readouts as the factors with the highest explanatory power, tightly correlated with the abundances of some (but not all) of the most abundant bacterialphylotypes (ASVs). Although previous studies have explored the variation in plant microbiome composition across populations, the depth and breadth of the dataset presented, as well as some potentially interesting new findings, raises the interest of this manuscript. As far as I can judge, the data analysis and interpretation are more than appropriate, and I think the authors should be praised for making code and intermediate data publicly available at this stage. Although I can clearly see the relevance of this manuscript, I have a lingering concern that this study might represent only an incremental step towards improving our understanding of microbiome variation in nature and its link to host genetics. Below I list a few points for the authors to consider and suggest an additional experiment which could help address this concern and that I believe might improve the study.

We thank the reviewer for the critical reading and useful suggestions. We have addressed the concerns below, and added several analyses that hopefully clarify the results.

Major points:

1. The authors find that, among abundant bacterial phylotypes, some of them appear to have a patchy distribution, while others correlate with climate and geographical variables such as latitude or PDSI. One possible interpretation is that the first group of bacteria are differentially selected by genetic variation in immune genes, which are not geographically structured (e.g., ATU5). In contrast, members of the second group (e.g., Sphingomonadaceae or Rhizobiaceae) may be selected by environmental variation directly, or indirectly by geographically-structured and climate-driven variation in host genetics.

This is an interesting hypothesis — that ASVs less structured by the environment are more likely to be structured by host genetics. We investigated this hypothesis with data and analyses we describe below. In brief, we did not find this association, but our results do not rule out this association when more fine-scaled data will become available.

To test the hypothesis that ASVs less structured by the environment are more likely to be influenced by heritability, we ascertained heritability estimates for the abundance of ASVs identified in our study. We did this by leveraging the results from a recent common garden

15study of *A. thaliana* leaf microbiomes, in which 200 *A. thaliana* genotypes were grown over several seasons at four locations in Southern and Northern Sweden (Brachi et al. 2022). The experimental data from this study allowed us to obtain heritability estimates and confidence intervals for almost half of the ASVs

To test for the association of these ASVs with latitude, we determined the association of heritability estimates with two spatial metrics from our study: (i) latitude, and (ii) genetic

differentiation between regions. We found that nearly all (98.5%) considered ASVs are heritable (pseudo- h^2 significantly greater than 0) in at least one of the four locations. The fact that nearly all of the ASVs can be influenced by host genetics indicates that there is not a qualitative difference between ASVs that show a spatial pattern vs. those that do not. The small magnitude of heritability estimates in this study precludes making quantitative comparisons

The authors do an excellent job in presenting these alternative hypotheses, but ultimately the answer is unsatisfactory since the data from this continental survey cannot differentiate between a direct impact of climate on abundant, not pathogenic microbes, and the effect of climate-driven variation in host genetics (e.g., drought). To address this important question, perhaps the authors could consider a greenhouse experiment, where a few ecotypes from representative populations along a latitude/PDSI gradient are grown under controlled conditions and drought treatment. These plants can be inoculated with an ATU5 and/or an *Sphingomonas* isolate(s) from an abundant phylogroup that shows latitudinal variation (or a close relative from the *A. thaliana* leaf culture collection if not available). If the final abundances of the *Sphingomonas* isolate(s) show variation across ecotypes in the control condition, an effect of host genetics is likely. If they only show variation in comparison with the drought-treated plants, this could be taken as evidence of direct influence of the environment on the microbes, independently of the host.

As described above, we have now added analyses that determine the genetic capacity of the plants to select for specific ASVs. We have also added an experiment that specifically tests the effect of drought on the abundance and fitness impact of colonization by the most common *Pseudomonas* ASV. These results underscore the ability of the host to influence nearly all of the microbiome through its genetic composition.

2. I find the clustering into $k=2$ 'microbiome types' somewhat problematic. Firstly, I think that clustering approaches that use pairwise distances or dissimilarities are better suited for compositional data than methods that need an embedding into a metric space such as k -means. Why do k -means on projected coordinates in 2 components (which capture only a fraction of all variance) instead of simply doing hierarchical clustering on the pairwise dissimilarity matrix? Secondly, it is not at all clear to me that $k=2$ is a good choice, by

looking at Fig. 2c and Fig. S4, the latter of which I don't find very informative. What are the silhouette scores for $k=3, 4, \dots$? I understand one of the main reasons for this analysis is to map Fig. 2c onto 2d, but this could be easily done by coloring 2c according to latitude, for instance. I think fitting the data into two microbiome types is rather artificial, given what looks to me like a rather smooth distribution of samples along MDS1. I would worry that the authors might be repeating the same error made in gut microbiota field, when the concept of discrete 'enterotypes' was introduced (Arumugam, et al., 2011),

and then proven to be somewhat flawed years later, when more data showed a smooth gradient in the abundances a small subset of key taxa, rather than well-defined clusters (Koren et al., 2013). My suggestion would be to remove this clustering analysis altogether, color Fig. 2c according to latitude (or show a biplot), and perhaps use hierarchical clustering as an alternative (e.g., on top of Fig. 2e).

We appreciate and agree with the concern that the microbiome type classification may not be entirely appropriate for the continuous distribution observed in the MDS plots. To clarify one point: the k-means clustering was performed on the relative abundance counts table, and not the dimensionally reduced MDS loadings. We have also clarified this point in the text. We also include regression analyses throughout the manuscript that ignore microbiome type classifications and instead use loadings on the first two principal coordinate axes as quantitative response variables. We would like to preserve a subset of the clustering/classification analysis because the qualitative classification of microbiomes enables several contrasts (in ASV abundance for example such as Figure S5) that are more salient than are correlative associations. We include the quantitative correlations also for reference. The results do not differ in substance.

Minor points:

3. Soil pH doesn't seem to be among the climate variables used for the random forest classification and feature ranking. However, pH was reported to be one of the largest contributors to (root) microbiota variation across European *A. thaliana* populations (Thiergart et al., 2020). Is this also an important factor for predicting leaf community composition?

This is an interesting hypothesis. Unfortunately, we did not take pH measurements of the soil. We have made note of the suggestion, and will include these measurements in future collections.

4. Abstract: "52-68% of variance in the first two principal coordinates of microbiome type explained by host genotype" If I get this right, this is a percentage of a percentage – what is the fraction of overall variance explained by genotype?

19We have added text to the manuscript to clarify this point. We now write:

“The observed host genetic diversity was consistent with previous surveys (Figure S6), indicating that we had sampled a broad diversity of *A. thaliana* genotypes across Europe. To determine the relationship between host genotype and microbiome composition, we fitted a mixed-effect model that included relatedness as a random effect and the loading on the first axis of the decomposition of the microbiome composition as the phenotypic

response variable. We found that plant genotype alone explains 68% of the variance in the loading on the first principle coordinate axis, MDS1, and 52% of the variance in the loading on MDS2 (pseudo- $h^2 = 0.68$, s.e.=0.10 for MDS1 and pseudo- $h^2 = 0.52$, s.e.=0.12 for MDS2). MDS1 and MDS2 explain 8% and 5% of the variance in microbiome composition, respectively, consistent with host genetics playing only a minor role in the structuring of the microbiome.”

5. “500 major bacterial phylotypes” – use the term ‘amplicon variant’ or define ‘bacterial phylotype’ the first time is used. I was unsure at this point of whether any clustering or abundance filtering had been done to group sequences into phylotypes.

We have now changed the text to read “500 major bacterial amplicon variants (phylotypes)”.

6. The first four paragraphs of the ‘Results’ section read more as an Introduction. The results should probably start in the middle of page three instead.

We have shifted the beginning of the “Results” section to the recommended location in the manuscript.

7. Page 3, paragraph 4: “We assessed the microbial composition of the leaf and soil samples by sequencing the v3-v4 region of the 16S rRNA locus (hereafter rDNA) and classifying distinct 16S sequences as alternative sequence variants (ASV) using DADA2 (ref. 12)” As far as I know, ASV stands for ‘amplicon sequence variant’ (not ‘alternative’). Also, “ref. 12” can be removed.

Text changed to “Amplicon” and ref. removed.

8. Page 4: “Modeling the effect of host versus soil on the abundance of core microbial phylotypes revealed that 91% (524/575) of phylotypes were differentially abundant between the *A. thaliana* phyllosphere and soil”. Surely the analysis yielded more than 575 ASVs and these are abundant or prevalent ones? If so, this might be good to mention when defining how the authors use the term bacterial phylotype. In addition, perhaps it would clearer to write

“modeling the effect of compartment” or “microhabitat” or “leaf versus soil” rather than “host versus soil”.

We have changed the text to read “Modeling the effect of compartment (host leaf versus soil) on the abundance of core microbial phylotypes core in the phyllosphere revealed that 91% (524/575) of phylotypes were differentially abundant between the *A. thaliana*phyllosphere and soil (False Discovery Rate [FDR] <0.01), indicating filtering of different microbes between the plant and the soil.”

9. Figure S3: This is a very minor issue but I would say that phylogenies built from partial DNA sequences such as this one should not be ultrametric, unless a model with a clock has been used (which I don't think was the case here).

We added details to the Figure S3 Figure specifying that “The central tree in the Circos plot represents the maximum likelihood tree, plotted without inferred branch lengths (Schliep 2011).”

10. Page 5: “indicating extensive filtering of microbes that colonize the plant, whether from soil or from another environmental source” and again in the discussion “colonization of *A. thaliana* leaves serves as a strong filter from the surrounding environment”. With the current data one can only conclude this for neighboring soil and plant species, not for other environmental sources (from which they might be less or partial filtering) such as root/rhizosphere, leaves of nearby trees, rainwater, etc. In fact, roots might be an important environmental reservoir of leaf-colonizing bacteria (see for example the observation from Wagner et al., 2016, that a significant fraction of the leaf microbiota of another Brassicaceae is found in roots of the same plant).

We have changed the text to read “...filtering of different microbes between the plant and the soil...” and “colonization of *A. thaliana* leaves serves as a strong filter from the surrounding soil and other plant species...”

11. Figure 2: what dissimilarity measure was used for these beta-diversity analyses? Not stated in the figure caption or methods, as far as I can tell.

We transformed the ASV count matrices using a Hellinger transformation and performed multidimensional scaling on these transformed matrices. We have added these details to the figure caption.

12. Fig. 3a. Also Rhizobiaceae, an abundant and ubiquitous bacterial taxon in the leaf, seems

to have a relatively strong latitude component. I find this intriguing, but not mentioned in the text.

We have added the text “ASVs of Rhizobiaceae also exhibited a latitudinal distribution.”13. The analysis based on F_{st} in gene alleles linked to immunity is very interesting, and the fact that the top SNPs were identified in the ACD6 gene I guess is reassuring. Perhaps a similar analysis on drought-related genes might be relevant as well?

Thank you for this interesting suggestion. We have expanded our F_{st} analyses beyond immunity genes accordingly. Several studies have established that a large fraction of *A. thaliana* SNPs are correlated with water availability and are spatially structured (Expositio-Alonso et al., 2019, for example) and these polymorphisms show spatial gradients. To narrow our analysis to specifically probe SNPs associated with microbial community, we asked whether SNPs that control microbiome composition tend to show geographic structure.

To test the spatial distribution of polymorphisms associated with overall microbiome composition, we obtained SNPs that were previously shown to be significantly associated with microbiome composition across European *A. thaliana* (Bergelson et al., 2019) and compared spatial structuring of these SNPs. The distribution of F_{st} values for these SNPs is significantly different from that of the rest of the genome, on average higher, suggesting that populations of *A. thaliana* are differentiated in their genetic control of microbiota. The low coverage of the genomes in our data precluded measuring global F_{st} genomewide for SNPs between the Northern and Southern clusters, so we limited the analysis to global F_{st} from the genomes studied in ref. (1001 Genomes Consortium 2016). We have added this analysis to the revision in Figure S9.

14. Page 6: “While ATUE5 was ubiquitously present, its distribution was not uniform.” It is not clear from Fig. 3c that this ASV was ubiquitous. To me, it appears almost completely absent from Spain, England and southern Italy.

We find ATUE5 in 56% of samples. To be more precise in the text we write “While ATUE5 was found in the majority of samples...”

15. Analysis of climate variables as predictors of community composition. I think the random forest analysis (e.g, Fig. 4a) is very informative but I think biplots on a projection such as the one shown on Fig. 2c might be useful as well.

25We have included biplots in Figure S11, which shows the correlation of environmental variables with the MDS loadings.

16. Page 11 (methods): “918 samples had a sufficient number of reads”. How many reads was considered a sufficient number?Text changed to “918 samples had a sufficient number of reads (>1,000 reads)”.

Reviewer #2 (Remarks to the Author):

The manuscript by Karasov et al investigated the composition and driving force of the leaf microbiome of *Arabidopsis thaliana* populations in their native sites across Europe. First, the authors found a remarkable pattern that the leaf microbiomes separated along a latitude gradient, southern and northern parts of the European continent. Second, association analysis revealed that host genetics, especially SNPs of an immune regulator ACD6 in *Arabidopsis* correlated with leaf microbiome composition. In the end, the statistical model revealed that drought indices were the strongest predictors for leaf microbiome, but not for soil microbiome. This is a comprehensive sequencing study covering diverse environmental and plant genetic factors. Although it is a pity that the authors did not show functions leaf microbiome in *Arabidopsis* drought tolerance, the continent-scale data alone provide valuable and solid insight on leaf microbiome assembly and potential co-adaption of *Arabidopsis* accessions and leaf microbiome. However, several aspects are needed to be further explored or clarified. My concerns are as below:

We have now included data that indicates that one of the ASVs, the most abundant *Pseudomonas* ASV, improves drought tolerance of *A. thaliana*, although at the same time it is thriving less in drought conditions (Figure S12).

1) it is unclear why the microbiome clusters along latitude showed taxonomic patterns only at the phylotype (ASV) level, but not at higher levels (family or genus). The Southern cluster and Northern clusters significantly separate in the first dimension of MDS analysis (Figure 2c), which is very likely to be driven by a group of bacteria in closely related taxa (e.g. *Bacteroides* and *Prevotella* genera determine the enterotype in human gut) or some bacterial ASVs with high-abundance. The authors need to perform the association in Figure 2e at the level of family or genus, and showed all biomarker ASVs to distinguish the Southern and Northern clusters. The results would not only

27help to explain the data, but also serve as the basis to speculate or test leaf microbiome's function for Arabidopsis drought tolerance.

We have added statistical contrasts of the abundance of ASVs in the Northern and Southern clusters, broken down by bacterial genus (Figure S5).

2) A large proportion of the manuscript shows data on distribution analysis of Arabidopsis pathogens and association of immune genes with leaf microbiome composition, but the results do not explain host genetic regulation on leaf microbiome clustering on latitude,

because common *Arabidopsis* pathogens are patchily distributed and do not correlated with latitude. I would suggest the authors explore the data with more thoughts on commensals. The interactions between commensals and host plants are key questions in microbiome study, and may drive the plant evolution. The authors could search the association of latitude with genes that are reported to regulate *Arabidopsis* leaf or root commensal bacteria, which may improve the result of host genetic regulation on leaf microbiome in this large-scale data.

We agree with this comment and have rewritten several paragraphs throughout the manuscript to focus on the broader microbial community, and interactions of immune genes with both pathogenic and non-pathogenic organisms.

3) In the last part of the story, the authors focus on drought indexes PDSI as the strongest predictors of leaf microbiome composition. What about others, such as srad and vpd? They also show a considerable contribution in the statistic model (Figure 4a). I understand that it may be easy to focus on the strongest predictor, but it is also very important show a model using a few (e.g. three) predictors to obtain a largely correct leaf microbiome. I think, this is the power of such large-scale data. It would be great if the authors could extend the modeling part of the manuscript, and explain the influence of environmental factors on leaf microbiome in a more systematical way.

We have expanded the modeling in the manuscript in several dimensions. Firstly, we now include biplots to show the correlations between different environmental variables as well as the environmental variables and the principal coordinates.

4) The authors should test the association of leaf microbiome with drought-related SNPs with their geographic distribution. Since environmental factors are strong drivers for both microbiome and plant genetics, it is likely that drought-related genes evolve with the environment and plant microbiome. A similar analysis with the immune genes would be helpful to elucidate potential co-adaptation of *Arabidopsis* genomes and leaf microbiome under variable drought conditions in nature.

This question is similar to one posed by reviewer #1, and we provide the same answer

here. Several studies have established that a large fraction of the *A. thaliana* genome is correlated with water availability (Exposito-Alonso et al., 2019, for example). In comparison to selection on drought tolerance, selection on immunity in *A. thaliana* shows less geographic structure (Bakker et al., 2008, or Karasov et al., 2014, for example), which should improve our power to distinguish true positive associations from background.To test the spatial distribution of polymorphisms associated with microbiome composition overall, we obtained SNPs that were previously shown to be significantly associated with microbiome composition across European *A. thaliana* (Bergelson et al. 2019) and compared spatial structuring of these SNPs. The distribution of F_{st} values for these SNPs is significantly different from the rest of the genome, being on average higher, suggesting that populations of *A. thaliana* are differentiated in their genetic control of microbiota. The low coverage of the genomes in our data precluded measuring global F_{st} genomewide for SNPs between the Northern and Southern clusters, so we limited the analysis to global F_{st} from the genomes studied in ref. (1001 Genomes Consortium 2016). We have added this analysis to the revision in Figure S9.

Reviewer #3 (Remarks to the Author):

The manuscript by Karasov et al. describes the results of a large-scale sampling of *Arabidopsis thaliana* rosette microbiomes across Europe. A strong latitudinal pattern was observed, where microbiome composition clustered into two distinct groups roughly corresponding to “north” and “south”. Multiple environmental variables associated with each sample were either measured directly or acquired from databases, and a few plant traits (size, disease status, herbivory status) were also estimated. Shotgun sequencing reads were also used to characterize plant genotype. Overall this is a rich dataset with high potential value.

While the sampling effort and some of the analyses were quite impressive, and the paper was well-written and presented, in my opinion the data do not sufficiently support the authors' conclusion stated in the title, that drought pressure has been an agent of natural selection shaping the phyllosphere microbiome. The paper demonstrates correlations between several sets of variables - microbiome composition, latitude, environment (esp. drought), and host genotype - but ultimately, causal relationships between these factors cannot be identified from this type of sampling design. Instead, the data simply demonstrate that all of these things are spatially structured by latitude - a pattern with

31numerous possible alternative explanations. To convincingly show a role for leaf microbiomes in host adaptation to drought, I would expect to see this dataset bolstered by manipulative experiments of some kind, such as a greenhouse experiment with inoculated and uninoculated plants under simulated drought. Fitness measurements and/or much more thorough population genomic analyses are needed to invoke natural selection as the cause of the observed patterns.

We agree with the reviewer that we did not present sufficient evidence to support the claim that the leaf microbiome plays a role in host adaptation to drought. This larger claim was, however, not the focus of our work. Instead our goal is to present evidence that (i)

there are reproducible geographic patterns in the distribution of *A. thaliana*-associated microbes, (ii) plants have genetic influence over most of these microbes, and (iii) the reproducible differences in microbial community are associated with water availability. We include now a new piece of experimental data indicating that the relationship of *A. thaliana* to one of its most abundant microbes changes across environments. Together, these data suggest that the microbe-associated selective pressure on *A. thaliana* differs reproducibly across regions. We have amended the text in the manuscript to clarify these points.

Even the claim that host genotype is a major factor shaping microbiome composition, is difficult to defend without a way to eliminate the environmental contribution. I was surprised that the authors did not attempt to fit models of leaf microbiome composition that included latitude, environmental variables, and host genotype as predictors, simultaneously. This model would have been very informative, and increased my confidence that some or all of these factors have a marginal effect on the microbiome. Instead there is a series of separate analyses using subsets of these variables, such that it is not clear whether any of them has an effect independent of the others.

Another criticism is that the paper is unfocused. Ostensibly the paper is about the leaf microbiome's role in plant adaptation to drought. However, a very large proportion of the paper actually is focused on spatial distribution of pathogens (ATUE5), disease, and immunity. Granted, allelic variation at immunity genes such as ACD6 may well be important for microbiome composition. Nevertheless, this paper contains no actual experimental evidence of causal links between disease, drought, and leaf microbiomes.

To provide greater focus to the manuscript, we have rearranged it to consolidate sections on plant genotype, and have moved the focus of the analyses away from pathogens specifically to rely on microbes in general. We have also included experimental data (our own and from others) that drought affects the pathogenic effect of a common ASV on *A. thaliana*.

Finally, it is unclear what contribution this paper makes to the broader field. What do these results teach us about the role of leaf microbiomes in adaptation to drought, or even plant-microbiome interactions in general? References to the pre-existing literature are quite

sparse. There is no discussion of possible mechanisms linking phyllosphere microbes to drought resistance, in any direction of causality. It is unclear if links between pathogens, drought, and phyllosphere microbiome have been reported in other systems.

We have added references to several studies that identify relationships between drought and microbial abundances as well as instances in which microbes altered the deleterious effect of drought. For example, we added the following text:

“Such findings are reminiscent of viral infection being able to reduce drought-based mortality in plants (González et al. 2021) and they are in agreement with numerous other studies that had demonstrated plant growth promoting effects of microbes under drought (Ma, Dias, and Freitas 2020).”

And:

“Shaffique et al. (Shaffique et al. 2022) provide a comprehensive review of the diverse mechanisms of microbe-mediated drought tolerance in plants.”

Other comments:

The dataset includes information about microbial load, but it is unclear how this was used.

We have removed references to microbial load in the manuscript.

It is also unclear how the plant phenotype data were used - for example, developmental stage was recorded but not used as a predictor or covariate in the models. Other studies have shown strong changes in leaf microbiomes with developmental stage. Is it possible that genetic variation for developmental rate is at the core of observed correlations between genotype and microbiome?

Developmental state and the other plant phenotypic data were included as variables in feature selection with random forest modeling, but they were not significant predictors of microbiome type in comparison to the climatic predictors from Terraclimate. These results are supported by the biplots we have added in Figure S5.

Is PDSI during the 6 months prior to collection a reliable proxy for drought pressure over evolutionary time (which is when genomic signatures of plant-microbiome adaptation would have been shaped)?

This is an interesting question. PDSI is a relative drought metric, calculated from the deviance of the average water content of a region. Hence, PDSI is not obviously tied to long-term (evolutionary timescales) water availability. We have added text to the manuscript to acknowledge this trend.

While PDSI could have a causative influence on the microbiota, it is less clear that PDSI alone would be a driver of the plant genetic signal. Several other absolute measures of drought are closely correlated with PDSI (Figure S8), however.

What value is added by repeating analyses using logistic regression with microbiome clusters, and also linear regression with latitude?

We included both logistic regression on the cluster classification and linear regression for the loading on MDS1 because we wanted to test whether the association held in the absence of microbiome type clustering. As reviewer 1 points out, it is debatable whether we are observing two microbiome types rather than a continuous distribution of differences between samples. We wanted to test for the association with PDSI in both scenarios.

“Previous work on non-host associated soils supports the importance of water availability in determining microbiome composition” Why was the effect of drought on soil microbiomes negligible in this study?

The current analyses are focused on the plant-associated core phylotypes. We analyse the 575 phylotypes most abundant across *A. thaliana* samples. These are largely not the same phylotypes that are common in the soil. Hence the ordination we performed is not likely to show differences in common soil-associated taxa.

“To probe for possible relative contributions of genotype and drought, we performed mixed effect modeling and estimated the marginal R² for PDSI to be 50%.” And what about the marginal R² for genotype? Earlier in the paper, genotype was stated to be the dominant force shaping the leaf microbiomes (>50% of variance)?

The text now reads “The observed host genetic diversity was consistent with previous surveys (Figure S6), indicating that we had sampled a broad diversity of *A. thaliana* genotypes across Europe. To determine the relationship between host genotype and microbiome composition, we fitted a mixed-effect model that included relatedness as a random effect and the loading on the first axis of the decomposition of the microbiome composition as the phenotypic response variable. We found that plant genotype alone explains 68% of the variance in the loading on the first principle coordinate axis, MDS1, and 52% of the variance in the loading on MDS2 (pseudo-h² = 0.68, s.e.=0.10 for MDS1 and pseudo-h² = 0.52, s.e.=0.12 for MDS2). MDS1 and MDS2 explain 8% and 5% of the variance in microbiome composition, respectively, consistent with host genetics likely playing only a minor role in the structuring of the microbiome.”

“Drought might do so directly, by affecting the physiological state of the plant, indirectly

by shaping host genetics, or both.” This dataset could have been used to investigate whether the physiological state of the plant is a more proximate cause of microbiome variation - by including stage, size, etc. as predictors in a model.

We have now added biplots to Figure S11 to show the correlation of physiological variables vs environmental variables with the loadings on MDS1 and MDS2. These biplots

illustrate that the physiological variables, with the exception of developmental state, are not well correlated with either MDS1 or MDS2.

“To determine the relative effect sizes of drought, latitude and plant identity on MDS loadings, phenotypes were modeled with the *gaston* package in R. The model was $Y_i \sim \text{PDS}_{li} + \text{Lati} + k_{i,j} + \epsilon_i$ where Y_i is the phenotype for each i th accession, $k_{i,j}$ is the genetic relatedness between the i th and j th accessions...” I am very confused by this model. The response variable (MDS loading) is solely a property of accession i . Yet, one of the predictors describes the genetic relationship between accessions i and j . How was accession j chosen or defined?

We agree that the model was not written previously in a cogent manner. To determine the relative effect sizes of the fixed effects under consideration (physiological and environmental measurements), we performed mixed effect modeling with a kinship matrix of plant genotypes. We have removed the confusing model from the text and have written “To determine the relative effect sizes of drought, latitude, and plant identity on MDS loadings, phenotypes were modeled using restricted expectation maximum likelihood with the *lme4* package in R with kinship as a random effect (H. Perdry, Dandine-Roulland, and Bandyopadhyay 2018). The kinship matrix was constructed using several methods including the R package *gaston* as well as the centered kinship matrix in *gemma* (version 0.98.3)(Zhou and Stephens 2012).”

1001 Genomes Consortium. Electronic address: magnus.nordborg@gmi.oeaw.ac.at, and 1001 Genomes Consortium. 2016. “1,135 Genomes Reveal the Global Pattern of Polymorphism in *Arabidopsis Thaliana*.” *Cell* 166 (2): 481–91.

Brachi, Benjamin, Daniele Filiault, Hannah Whitehurst, Paul Darne, Pierre Le Gars, Marine Le Mentec, Timothy C. Morton, et al. 2022. “Plant Genetic Effects on Microbial Hubs Impact Host Fitness in Repeated Field Trials.” *Proceedings of the National Academy of Sciences* 119 (30): e2201285119.

González, Rubén, Anamarija Butković, Francisco J. Escaray, Javier Martínez-Latorre, Ízan Melero, Enric Pérez-Parets, Aurelio Gómez-Cadenas, Pedro Carrasco, and Santiago F. Elena. 2021. “Plant Virus Evolution under Strong Drought Conditions Results in a Transition from Parasitism to Mutualism.” *Proceedings of the National Academy of Sciences* 118 (6): e2020990118.

Ma, Ying, Maria Celeste Dias, and Helena Freitas. 2020. "Drought and Salinity Stress Responses and Microbe-Induced Tolerance in Plants." *Frontiers in Plant Science* 11 (November): 591911.

Perdry, H., C. Dandine-Roulland, and D. Bandyopadhyay. 2018. "Gaston: Genetic Data Handling (QC, GRM, LD, PCA) & Linear Mixed Models." *R Packag.*
https://scholar.google.ca/scholar?cluster=18142910180256059156&hl=en&as_sdt=0,5&sci odt=0,5.

Perdry, Hervé. n.d. "Claire Dandine-Roulland Gaston: Genetic Data Handling (QC, GRM, LD, PCA)." *Linear Mixed Models Version 1*.

Shaffique, Shifa, Muhamad Aaqil Khan, Muhamad Imran, Sang-Mo Kang, Yong-Sung Park, Shabir Hussain Wani, and In-Jung Lee. 2022. "Research Progress in the Field of Microbial Mitigation of Drought Stress in Plants." *Frontiers in Plant Science* 13 (May): 870626.

Zhou, Xiang, and Matthew Stephens. 2012. "Genome-Wide Efficient Mixed-Model Analysis for Association Studies." *Nature Genetics* 44 (7): 821–24.

Decision Letter, first revision:

Message: 1st February 2023

Dear Detlef,

Thank you for your patience while your manuscript "Drought selection on *Arabidopsis* populations and their microbiomes" was under peer review at Nature Microbiology. It has now been seen by our referees, whose expertise and comments you will find at the end of this email. In the light of their advice, we have decided that we cannot offer to publish your manuscript in Nature Microbiology.

From the reports, you will see that while they find your work of some potential interest, the referees raise concerns about the advance your findings represent over earlier work and the strength of the novel conclusions that can be drawn at this stage. In particular, our referees remain unconvinced that the data, including the additional experiments, support conclusions regarding drought adaptation. Unfortunately, these criticisms are sufficiently important as to preclude publication of your work in Nature Microbiology.

As you have opted out of transfer consultations, I am unable to discuss the paper and the reports in confidence with my colleagues at other journals, such as Nature Communications. If you would like me to consult with editors at Nature Communications, please let me know.

I am sorry that we cannot be more positive on this occasion, but hope that you find the

40referees' comments helpful when preparing your paper for resubmission elsewhere.

Yours sincerely,

Reviewer Expertise:

Referee #1: plant microbiome, multi-omics, bioinformatics

Referee #2: plant-microbe interactions, multi-omics

Referee #3: plant genetics and microbiome

Reviewers Comments:

Reviewer #1 (Remarks to the Author):

The authors have convincingly addressed most of my concerns and they have improved the manuscript with this revised version. However, I still think that the current dataset does not allow the authors to assess whether there is a direct link between host adaptation to drought and microbiome variation. This, in my opinion, is an important shortcoming that potentially limits the contribution that this manuscript makes to the broader field.

While it is interesting that ATUE5 treatment dampens the negative impact of drought on *Arabidopsis*, the significant albeit small reduction in bacterial CFUs might have been observed in general, including for other microbial isolates from taxa not correlated with PDSI. Additionally, whether this reduction is at all linked with host genotype is unclear, since only Col-0 was used.

As stated in my first assessment, an experiment under controlled conditions using representative ecotypes and drought treatment could help clarify this point. I note that this comment was also made by reviewer #3, and by reviewer #2, who suggests to select 10-20 representative ecotypes for experiments with or without drought bacterial inoculation.

Reviewer #2 (Remarks to the Author):

The authors have answered all my questions. I have no further requests.

Reviewer #3 (Remarks to the Author):

I have carefully reviewed the revised manuscript and the authors' rebuttal of my comments on the original manuscript. This work certainly provides intriguing evidence of a complex relationship between microbiome composition, pathogen activity, and drought resistance in the leaves of *Arabidopsis thaliana*.

Unfortunately some of my concerns have not been sufficiently addressed. Namely:

I had argued that this work is consistent with, but does not actually demonstrate that drought has exerted selection to shape leaf microbiomes. The authors replied, "We agree with the reviewer that we did not present sufficient evidence to support the claim that the leaf microbiome plays a role in host adaptation to drought. This larger claim was, however, not the focus of our work. Instead our goal is to..." Then why is the title of the paper "Drought selection on Arabidopsis populations and their microbiomes" rather than something that actually describes the authors' intended goal? I worry that the readers of Nature Microbiology, if not already well versed in evolutionary genetics methodology, will come away with the wrong impression. Because from what I can see, only the first complete paragraph on pg 8 (higher F_{st} in microbiome-related vs. average SNPs) directly tests for selection shaping Arabidopsis microbiomes, and even this cannot identify drought as the driver of selection.

On this same point, the authors did add some data from a growth chamber experiment demonstrating that inoculation with a core pseudomonad strain, ATUE5, confers some amount of drought resistance, even while drought reduces the fitness of ATUE5. I appreciate this addition and find the result quite interesting. Unfortunately it does not help much to convince me that "drought selection" is shaping geographic patterns of host genotypic and microbiome variation, in part because (as clearly shown in past work and in Figure 3) ATUE5 has a patchy distribution and is not at all correlated with either latitude or PDSI. Maybe I am missing something.

Overall I think the title of this paper must be changed to match the contents. The last sentence of the abstract, "...raise the possibility that drought not only directly shapes genetic variation in *A. thaliana*, but does so also indirectly through its effects on the leaf microbiome", is a much more accurate description.

I had written, "Even the claim that host genotype is a major factor shaping microbiome composition, is difficult to defend without a way to eliminate the environmental contribution. I was surprised that the authors did not attempt to fit models of leaf microbiome composition that included latitude, environmental variables, and host genotype as predictors, simultaneously. This model would have been very informative, and increased my confidence that some or all of these factors have a marginal effect on the microbiome. Instead there is a series of separate analyses using subsets of these variables, such that it is not clear whether any of them has an effect independent of the others."

As far as I can tell, this comment was not addressed at all (though again, perhaps I am missing something). The analyses in the paper relating genotype to microbiome composition, described in the last full paragraph in p. 7, still do not control for either spatial autocorrelation or environmental factors that may independently structure plant genotypes and microbiomes. I disagree with the authors' assertion that this is an "unbiased" method of assessing the host genetics-microbiome relationship. These are paired data coming from individual plants that occupy a particular location in space. Thus there is ample opportunity for underlying geographic patterns, climate, soil, etc to bias the result toward a positive correlation.

To be clear, I don't doubt that there is some role of host genotype shaping these communities (there are plenty of common garden studies confirming this) but in its current state this analysis does not add rigorously to that body of evidence, and the % of variation contributed by genotype across such a wide environmental range may well be overstated in this study.

I appreciate the authors' efforts to consolidate diverse analyses into a simpler framework. However I admit that the paper does still seem unfocused to me. For example, only 1.5 pages out of 10.5 even mention drought. At least equal attention is paid to host genetic variation, disease, and latitude. So rather than trying to make it one large story about drought selection, why not present this study for what it is: a valuable continental-scale study of the patterns and sources of variance in leaf bacterial microbiomes?

Finally, I must admit that I share Ref. 1's "lingering concern that this study might represent only an incremental step towards improving our understanding of microbiome variation in nature and its link to host genetics". This is an impressive, wide-ranging, and valuable dataset that reveals some intriguing patterns, but I do not quite see what major advance this paper provides.

Two more minor comments:

The abstract should clarify that 52-68% of variance explained for MDS1-2 is actually ~13% of total microbiome variance. This is explained on pg 7 but it is misleading not to do so in the abstract.

On page 7: "In our study we were able to determine microbiome composition and host genotype from the same wild individuals, and we could therefore assess the relationship between host genotype and microbiome composition in an unbiased manner." This seems like a non-sequitur. What does this have to do with the previous sentences that were speculating on possible evolutionary/adaptive explanations for the geographic patterning of microbiomes? Of course heritability being necessary for adaptive evolution, but demonstrating a relationship to host genotype is not at all sufficient to test these hypotheses about selection acting on microbiomes and/or their functions.

** I suggest that you consider Nature Communications as a suitable venue for your work. To transfer your manuscript there, please use our manuscript transfer portal. You will not have to re-supply manuscript metadata and files, unless you wish to make modifications, but please note that this link can only be used once and remains active until used. For more information, please see our manuscript transfer FAQ page.

Note that any decision to opt in to In Review at the original journal is not sent to the receiving journal on transfer. You can opt in to In Review at receiving journals that support this service by choosing to modify your manuscript on transfer. In Review is available for primary research manuscript types only.

Author Rebuttal, first revision:

From the reports, you will see that while they find your work of some potential interest, the referees raise concerns about the advance your findings represent over earlier work and the strength of the novel conclusions that can be drawn at this stage. In particular, our referees remain unconvinced that the data, including the additional experiments, support conclusions regarding drought adaptation. Unfortunately, these criticisms are sufficiently important as to preclude publication of your work in Nature Microbiology.

In response to these critiques we conducted and now include a field study from Spring 2023 that explicitly tests the effects of drought vs. genotype on the abundance of our microbes of interest. This experiment hews closely to what was requested by the external reviewer and two of the main reviewers. This experiment provided sufficient power to identify microbes that were influenced directly by drought and microbes that were influenced by plant genetic adaptations to drought. These results now provide a direct line of evidence connecting the continental-scale patterns of microbial biogeography and plant genetics.

As you have opted out of transfer consultations, I am unable to discuss the paper and the reports in confidence with my colleagues at other journals, such as Nature Communications. If you would like me to consult with editors at Nature Communications, please let me know.

We hope that the addition of our new results is sufficient to warrant another round of review at Nature Microbiology.

Reviewers Comments:

Reviewer #1 (Remarks to the Author):

The authors have convincingly addressed most of my concerns and they have improved the manuscript with this revised version. However, I still think that the current dataset does not allow the authors to assess whether there is a direct link between host adaptation to drought and microbiome variation. This, in my opinion, is an important shortcoming that potentially limits the contribution that this manuscript makes to the broader field.

In response to this critique, we conducted an outdoor common garden drought experiment that tests the effect of *A. thaliana* SNPs associated with drought adaptation on the microbiome in a controlled design. With this substantial experimental effort, we have now identified a set of ASVs that are specifically associated in the common garden with genetic backgrounds that are drought-adapted. We also identified ASVs that are significantly associated with drought treatment without significant impact of host genotype. Together, these results indicate that ~15% of the ASVs for which we had sufficient data are associated with the host-genotypic background, and a similar number of ASVs are associated with interactions between host genetic background and drought treatment. A substantial fraction of ASVs, 10% percent, were independently affected by drought. We describe the experiments in greater detail below and have included additional sections in the manuscript.

While it is interesting that ATUE5 treatment dampens the negative impact of drought on Arabidopsis, the significant albeit small reduction in bacterial CFUs might have been observed in general, including for other microbial isolates from taxa not correlated with PDSI. Additionally, whether this reduction is at all linked with host genotype is unclear, since only Col-0 was used. As stated in my first assessment, an experiment under controlled conditions using representative ecotypes and drought treatment could help clarify this point. I note that

this comment was also made by reviewer #3, and by reviewer #2, who suggests to select 10-20 representative ecotypes for experiments with or without drought bacterial inoculation.

We appreciate this thoughtful suggestion and have finally been able to conduct the suggested experiment. Specifically, we have collected microbiome composition data from a common garden drought experiment using diverse *A. thaliana* accessions that had previously been identified as having polymorphisms associated with drought-adaptation or drought-sensitivity¹. These accessions were randomized and grown outside in either well-watered or drought conditions. We obtained 16S rDNA sequencing data, processing these using the same pipelines that we had developed for the range-wide analysis of wild samples. We found at least 30% of the core ASVs that we identified in Europe were present in the common garden experiment carried out in the US. This sharing of ASVs allowed us to test which ASVs are influenced by drought status and which are influenced by host genetics.

We have added a section in the manuscript to relay the results:

“Host adaptation to drought influences microbial abundance

Our results indicated that the spatial microbiome patterns are correlated both with host genetics and drought metrics. To disentangle the impact of drought from that of differences in plant genetics, we conducted a common garden field experiment at Stanford, California, with a differential watering regime. Using a setup similar to our previous work in Germany and Spain², we grew *A. thaliana* accessions under a high- and low-watering regime. Using accessions that had previously been identified as drought-adapted or drought-susceptible based on a panel of genetic loci associated with adaptation to drought¹, we assessed differences in phyllosphere composition after drought exposure. In this Californian field experiment, we found 154/575 core phylotypes from the European field collections. Twenty

of these phylotypes were found at frequencies that were sufficiently high to enable us to determine the relative influences of genetics and drought treatment on their relative abundances (Table S3). Of these phylotypes, 3/20 were significantly influenced by host genetic classification of drought-adapted versus drought-susceptible and 3/20 showed a significant interaction between drought treatment and host genotype. Two out of twenty showed a significant response to the abiotic drought treatment alone. In total, these results indicate that the drought adaptation status of the plant modifies which microbes colonize.”

Reviewer #2 (Remarks to the Author):

The authors have answered all my questions. I have no further requests. **We thank the reviewer for their useful comments.**

Reviewer #3 (Remarks to the Author):

I have carefully reviewed the revised manuscript and the authors' rebuttal of my comments on the original manuscript. This work certainly provides intriguing evidence of a complex relationship between microbiome composition, pathogen activity, and drought resistance in the leaves of *Arabidopsis thaliana*.

Unfortunately some of my concerns have not been sufficiently addressed. Namely:

I had argued that this work is consistent with, but does not actually demonstrate that drought has exerted selection to shape leaf microbiomes. The authors replied, "We agree with the reviewer that we did not present sufficient evidence to support the claim that the leaf microbiome plays a role in host adaptation to drought. This larger claim was, however, not the focus of our work. Instead our goal is to..." Then why is the title of the paper "Drought selection on *Arabidopsis* populations and their microbiomes" rather than something that actually describes the authors' intended goal?

The field experiments with manipulated rainfall that we have added to the current manuscript hopefully address this concern. Specifically, our new experiments explicitly test whether host genotypes that are drought adapted influence the core microbiota in a specific manner, i.e., whether drought and host genotype select for specific microbial communities. We are, however, very happy to receive editorial suggestions.

I worry that the readers of Nature Microbiology, if not already well versed in evolutionary genetics methodology, will come away with the wrong impression. Because from what I can see, only the first complete paragraph on pg 8 (higher F_{st} in microbiome-related vs. average SNPs) directly tests for selection shaping *Arabidopsis* microbiomes, and even this cannot identify drought as the driver of selection.

With the addition of the new experiment we now include more explicit connections to plant adaptation to drought. Our previous research^{1,3} identified genotypes of *A. thaliana* that were adapted to drought conditions and identified the polygenic basis of this

adaptation. Using this polygenic information to classify genotypes, we find a set of ASVs that are differentially abundant between high and low-drought adapted genotypes, paralleling contrasts observed in our field collections.

On this same point, the authors did add some data from a growth chamber experiment demonstrating that inoculation with a core pseudomonad strain, ATUE5, confers some amount of drought resistance, even while drought reduces the fitness of ATUE5. I appreciate this addition and find the result quite interesting. Unfortunately it does not help much to convince me that “drought selection” is shaping geographic patterns of host genotype and microbiome

variation, in part because (as clearly shown in past work and in Figure 3) ATUE5 has a patchy distribution and is not at all correlated with either latitude or PDSI. Maybe I am missing something.

We chose to include these results as a demonstration that one of the common phylotypes (found in 56% of the plants) has an influence on the effect of drought on the plant. Our previous work demonstrated that this ASV interacts differently with different *A. thaliana* accessions⁴, raising the possibility of microbe x drought x host genetic adaptations. We have reworded this section to clarify its association with plant phenotypic effects. It now reads:

“Common phylotypes can alter the effect of drought on *A. thaliana*

To explicitly test whether water availability can influence the abundance of common phylotypes in *A. thaliana*, we tested the proliferation of the abundant opportunistic pathogen ATUE5 under different watering regimes. In growth chambers, we exposed four-week old *A. thaliana* plants of the Col-0 reference accession to a week-long drought, and syringe-inoculated plants with the p25.c2 strain of ATUE5. Three days after infection, we compared bacterial growth and the amount of remaining green tissue in drought-stressed and well-watered plants. Drought significantly reduced the ability of ATUE5 to proliferate *in planta* (Figure S12b; Wilcoxon

rank-sum test $p = 0.003$), a result consistent with *Pseudomonas* pathogens relying on water availability to spread and multiply⁵. Drought significantly reduced the green, photosynthetically active leaf area (Figure S12), with ATUE5 infection blunting this negative effect of drought.

These results indicate that infection by an opportunistic pathogen may be conditionally beneficial, conferring drought tolerance to *A. thaliana* under specific conditions. ATUE5 was previously shown to influence *A. thaliana* growth in a genotype-specific manner⁴ indicating that the interaction between drought and ATUE5 infection is likely to differ between plant populations. Such findings are reminiscent of viral infection being able to reduce

drought-based mortality in plants⁶ and they are in agreement with numerous other studies that have demonstrated plant growth promoting effects of microbes under drought⁷, as discussed in a recent comprehensive review⁸ of the diverse mechanisms of microbe-mediated drought tolerance in plants. Moreover, there is precedence for cryptic *A. thaliana*

pathogens providing environment-specific fitness benefits⁹.”

Overall I think the title of this paper must be changed to match the contents. The last sentence of the abstract, “...raise the possibility that drought not only directly shapes genetic variation in

A. thaliana, but does so also indirectly through its effects on the leaf microbiome”, is a much more accurate description.

In light of the new field experiment and the corresponding evidence of host genetic control of the spatially distributed ASVs we think the title is appropriate (again given the new results). We are, however, very happy to receive editorial suggestions.

I had written, “Even the claim that host genotype is a major factor shaping microbiome composition, is difficult to defend without a way to eliminate the environmental contribution.

As described above we have included a new field experiment that controls for environment (the common garden with a rainfall manipulation treatment), and we hope that the addition alleviates this specific concern.

I was surprised that the authors did not attempt to fit models of leaf microbiome composition that included latitude, environmental variables, and host genotype as predictors, simultaneously. This model would have been very informative, and increased my confidence that some or all of these factors have a marginal effect on the microbiome. Instead there is a series of separate analyses using subsets of these variables, such that it is not clear whether any of them has an effect independent of the others.”

We have completed the recommended modeling, including latitude, host genetics and PDSI within the same model to measure their relative contributions. The statistics can be found in Table S2 and are described in the manuscript:

“We tested the relative effects of latitude, PDSI and host genetics in a mixed effect model using the first five principal components of the host genotype similarity matrix¹⁰, and found PDSI to be associated with MDS1 in the full model whereas several genetic principal components are associated with MDS2 (Table S2).”

As far as I can tell, this comment was not addressed at all (though again, perhaps I am missing something). The analyses in the paper relating genotype to microbiome composition, described in the last full paragraph in p. 7, still do not control for either spatial autocorrelation or environmental factors that may independently structure plant genotypes and microbiomes. I disagree with the authors’ assertion that this is an “unbiased” method of assessing the host genetics-microbiome relationship. These are paired data coming from individual plants that occupy a particular location in space. Thus there is ample opportunity for underlying geographic patterns, climate, soil, etc to bias the result toward a positive correlation.

We have removed the words “in an unbiased manner” from the text.

To be clear, I don’t doubt that there is some role of host genotype shaping these communities (there are plenty of common garden studies confirming this) but in its current state this analysis does not add rigorously to that body of evidence, and the % of variation

contributed by genotype across such a wide environmental range may well be overstated in this study. I appreciate the authors' efforts to consolidate diverse analyses into a simpler framework.

However I admit that the paper does still seem unfocused to me. For example, only 1.5 pages out of 10.5 even mention drought. At least equal attention is paid to host genetic variation, disease, and latitude.

To focus the manuscript, we have rearranged the text and removed the general discussion on plant-disease interactions from the results section. Our addition of the field experiment has hopefully improved the structure and focus of the manuscript. The manuscript and its arguments flow now as follows:

Spatial patterns of the microbiome -> Environmental correlates for these patterns -> Genetic correlates for these patterns -> Test for genetics vs. drought effect in field experiment.

So rather than trying to make it one large story about drought selection, why not present this study for what it is: a valuable continental-scale study of the patterns and sources of variance in leaf bacterial microbiomes?

The experiment we have added explicitly tests drought and drought adaptation. We hope that the reviewer agrees with us that this new experiment brings together the spatial observations with host genetic observations in the structure we describe in the previous paragraph (Spatial patterns of the microbiome -> Environmental correlates for these patterns -> Genetic correlates for these patterns -> Test for genetics vs. drought effect in field experiment)

Finally, I must admit that I share Ref. 1's "lingering concern that this study might represent only an incremental step towards improving our understanding of microbiome variation in nature and its link to host genetics". This is an impressive, wide-ranging, and valuable dataset that reveals some intriguing patterns, but I do not quite see what major advance this paper provides.

We apologize if it remained unclear what the specific advance of our study is. To our knowledge, it is the first study that brings together information on the spatial variation among the microbes that colonize a wild plant with spatial variation in the genetic makeup of the host plant. In addition, we now provide data that explicitly test the proposed effect of drought treatment x host genetic background. The results of these experiments allow us to begin to disentangle host genetic effects from environmental effects. As the reviewer states, the dataset will be a substantial resource also for others who study interactions between microbes and plant health and how these interactions may evolve in a changing environment.

Two more minor comments:

The abstract should clarify that 52-68% of variance explained for MDS1-2 is actually ~13%

of total microbiome variance. This is explained on pg 7 but it is misleading not to do so in the abstract.

We have removed the reference to 52-68% in the abstract to remove confusion.

On page 7: “In our study we were able to determine microbiome composition and host genotype from the same wild individuals, and we could therefore assess the relationship between host genotype and microbiome composition in an unbiased manner.” This seems like a non-sequitur. What does this have to do with the previous sentences that were speculating on possible evolutionary/adaptive explanations for the geographic patterning of microbiomes? Of course heritability being necessary for adaptive evolution, but demonstrating a relationship to host genotype is not at all sufficient to test these hypotheses about selection acting on microbiomes and/or their functions.

We have removed this sentence from the manuscript.

References

1. Exposito-Alonso, M. *et al.* Natural selection on the *Arabidopsis thaliana* genome in present and future climates. *Nature* **573**, 126–129 (2019).
2. Exposito-Alonso, M. *et al.* A map of climate change-driven natural selection in *Arabidopsis thaliana*. Preprint at <https://doi.org/10.1101/321133>.
3. Exposito-Alonso, M. *et al.* Genomic basis and evolutionary potential for extreme drought adaptation in *Arabidopsis thaliana*. *Nat Ecol Evol* **2**, 352–358 (2018).
4. Duque-Jaramillo, A. *et al.* The genetic and physiological basis of *Arabidopsis thaliana* tolerance to *Pseudomonas viridiflava*. *bioRxiv* 2023.03.18.533268 (2023) doi:10.1101/2023.03.18.533268.
5. Aung, K., Jiang, Y. & He, S. Y. The role of water in plant–microbe interactions. *Plant J.* (2018).
6. González, R. *et al.* Plant virus evolution under strong drought conditions results in a transition from parasitism to mutualism. *Proceedings of the National Academy of Sciences* **118**, e2020990118 (2021).
7. Ma, Y., Dias, M. C. & Freitas, H. Drought and Salinity Stress Responses and Microbe-Induced Tolerance in Plants. *Front. Plant Sci.* **11**, 591911 (2020).
8. Shaffique, S. *et al.* Research Progress in the Field of Microbial Mitigation of Drought Stress in Plants. *Front. Plant Sci.* **13**, 870626 (2022).
9. Hiruma, K. *et al.* Root Endophyte Colletotrichum tofieldiae Confers Plant Fitness Benefits that Are

- Phosphate Status Dependent. *Cell* **165**, 464–474 (2016).
10. Yang, J., Lee, S. H., Goddard, M. E. & Visscher, P. M. GCTA: a tool for genome-wide complex trait analysis. *Am. J. Hum. Genet.* **88**, 76–82 (2011).

Decision Letter, second revision:

Message: 12th January 2024

Dear Detlef,

Thank you for your patience while your manuscript "Drought selection on *Arabidopsis* populations and their microbiomes" was under peer-review at Nature Microbiology. It has now been seen by 3 referees, whose expertise and comments you will find at the of this email. You will see from their comments below that while they find your work of interest, some important points are raised. We are very interested in the possibility of publishing your study in Nature Microbiology, but would like to consider your response to these concerns in the form of a revised manuscript before we make a final decision on publication.

In particular, you will see that while all referees appreciate the additional experiment, they have a few questions regarding the small number of altered ASVs. Specifically, both referees #1 and #3 ask that you determine how abundant these ASVs are within the phyllosphere and discuss the potential significance of these shifts. Please add abundance data for these ASVs and discuss the relevance of these changes.

Given the reviewer feedback and after discussions within the editorial team, we believe that this would be more suited as a RESOURCE given the scale and potential utility of the dataset. This does not require any formatting changes (it is essentially the same as an ARTICLE), but will highlight to our readers that the dataset will have high utility for them. Please also tone down conclusions regarding drought selecting for leaf microbiota composition given the concerns from the referees about the level of evidence for these conclusions. Instead you can say that both drought and genotype can select for the phyllosphere and there may be cross-selection. Please also add a discussion of confounders and limitations to the discussion section (in line with referee #3's previous concerns around confounders). We would suggest the following title: 'Continental-scale characterization of the *Arabidopsis thaliana* phyllosphere and associations with host genotype and drought'.

56If you have not done so already please begin to revise your manuscript so that it conforms to our Article format instructions at <http://www.nature.com/nmicrobiol/info/final-submission/>

The usual length limit for a Nature Microbiology Article is six display items (figures or tables) and 3,000 words. We have some flexibility, and can allow a revised manuscript at 3,500 words, but please consider this a firm upper limit. There is a trade-off of ~250 words per display item, so if you need more space, you could move a Figure or Table to Supplementary Information.

Some reduction could be achieved by focusing any introductory material and moving it to the start of your opening 'bold' paragraph, whose function is to outline the background to your work, describe in a sentence your new observations, and explain your main conclusions. The discussion should also be limited. Methods should be described in a separate section following the discussion, we do not place a word limit on Methods.

Nature Microbiology titles should give a sense of the main new findings of a manuscript, and should not contain punctuation. Please keep in mind that we strongly discourage active verbs in titles, and that they should ideally fit within 90 characters each (including spaces).

Please include a data availability statement as a separate section after Methods but before references, under the heading "Data Availability". This section should inform readers about the availability of the data used to support the conclusions of your study. This information includes accession codes to public repositories (data banks for protein, DNA or RNA sequences, microarray, proteomics data etc...), references to source data published alongside the paper, unique identifiers such as URLs to data repository entries, or data set DOIs, and any other statement about data availability. At a minimum, you should include the following statement: "The data that support the findings of this study are available from the corresponding author upon request", mentioning any restrictions on availability. If DOIs are provided, we also strongly encourage including these in the Reference list (authors, title, publisher (repository name), identifier, year). For more guidance on how to write this section please see:

<http://www.nature.com/authors/policies/data/data-availability-statements-data-citations.pdf>

To improve the accessibility of your paper to readers from other research areas, please pay particular attention to the wording of the paper's opening bold paragraph, which serves both as an introduction and as a brief, non-technical summary in about 150 words. If, however, you require one or two extra sentences to explain your work clearly, please include them even if the paragraph is over-length as a result. The opening paragraph should not contain

references. Because scientists from other sub-disciplines will be interested in your results and their implications, it is important to explain essential but specialised terms concisely. We suggest you show your summary paragraph to colleagues in other fields to uncover any problematic concepts.

If your paper is accepted for publication, we will edit your display items electronically so they conform to our house style and will reproduce clearly in print. If necessary, we will re-size figures to fit single or double column width. If your figures contain several parts, the parts should form a neat rectangle when assembled. Choosing the right electronic format at this stage will speed up the processing of your paper and give the best possible results in print. We would like the figures to be supplied as vector files - EPS, PDF, AI or postscript (PS) file formats (not raster or bitmap files), preferably generated with vector-graphics software (Adobe Illustrator for example). Please try to ensure that all figures are non-flattened and fully editable. All images should be at least 300 dpi resolution (when figures are scaled to approximately the size that they are to be printed at) and in RGB colour format. Please do not submit Jpeg or flattened TIFF files. Please see also 'Guidelines for Electronic Submission of Figures' at the end of this letter for further detail.

Figure legends must provide a brief description of the figure and the symbols used, within 350 words, including definitions of any error bars employed in the figures.

When submitting the revised version of your manuscript, please pay close attention to our [href="https://www.nature.com/nature-research/editorial-policies/image-integrity">Digital Image Integrity Guidelines](https://www.nature.com/nature-research/editorial-policies/image-integrity). and to the following points below:

Please include a statement before the acknowledgements naming the author to whom correspondence and requests for materials should be addressed.

Finally, we require authors to include a statement of their individual contributions to the paper -- such as experimental work, project planning, data analysis, etc. -- immediately after the acknowledgements. The statement should be short, and refer to authors by their initials. For details please see the Authorship section of our joint Editorial policies at http://www.nature.com/authors/editorial_policies/authorship.html

* include a point-by-point response to any editorial suggestions and to our referees. Please include your response to the editorial suggestions in your cover letter, and please upload your response to the referees as a separate document.

* ensure it complies with our format requirements for Letters as set out in our guide to authors at www.nature.com/nmicrobiol/info/gta/

* state in a cover note the length of the text, methods and legends; the number of references; number and estimated final size of figures and tables

*This url links to your confidential homepage and associated information about manuscripts you may have submitted or be reviewing for us. If you wish to forward this e-mail to co-authors, please delete this link to your homepage first.

Please ensure that all correspondence is marked with your Nature Microbiology reference number in the subject line.

Nature Microbiology is committed to improving transparency in authorship. As part of our efforts in this direction, we are now requesting that all authors identified as 'corresponding author' on published papers create and link their Open Researcher and Contributor Identifier (ORCID) with their account on the Manuscript Tracking System (MTS), prior to acceptance. This applies to primary research papers only. ORCID helps the scientific community achieve unambiguous attribution of all scholarly contributions. You can create and link your ORCID from the home page of the MTS by clicking on 'Modify my Springer Nature account'. For more information please visit www.springernature.com/orcid.

We hope to receive your revised paper within three weeks. If you cannot send it within this time, please let us know.

Yours sincerely,

Reviewers Comments:

Reviewer #1 (Remarks to the Author):

I appreciate the authors' comments to my previous assessment and have now read with interest the description of the new common garden experiment. Although the experimental design seems very well suited to fully address my remaining concerns, I was puzzled by the way these results are presented. What I understand from the text is that 3 out of the 575 previously identified bacterial phylotypes show a significant interaction between drought and host genotype. How abundant are these three phylotypes? More importantly, what percentage of variation do these factors and their interaction explain in this dataset? The

59associated supplementary table (S4), which is merely a list of bacterial ASV IDs and p-values, does not provide more information. Perhaps I am missing something here, but in my opinion, the observation that 3 ASVs show a significant interaction in the common garden experiment is not sufficient to support the main claim of the manuscript.

I am afraid that the new data provided in this revised version and the adjustments made by the authors are not sufficient to fundamentally change my previous assessment. I believe this manuscript constitutes an impressive effort and makes a valuable contribution but I remain unconvinced about the main claim that drought selection has significantly shaped leaf microbiota composition.

Reviewer #2 (Remarks to the Author):

I am satisfied with the manuscript. Current plant microbiota research field lacks big scale survey with high scientific standard (like high level studies in human gut microbiota), which is very important for demonstrating microbiota patterns in natural environments, and for illustrating the importance to further investigate molecular mechanisms. This work is an excellent example for such big scale survey. On the other side, I would suggest the authors to adjust or weaken some causal conclusions if no experimental data under controlled conditions are provided. Taken together, I think, the revised manuscript would significantly contribute to the plant microbiota research field.

Reviewer #3 (Remarks to the Author):

I have once again carefully read the revised manuscript and the rebuttal to my previous comments on the first and second versions of this manuscript.

I appreciate the addition of data from a common-garden field experiment and the restructuring of the text. I agree that the new flow of topics is logical and easier to follow.

1. The new field experiment demonstrates a direct effect of water availability on a few of the 20 ASVs that were tested, and evidence for a genotype-drought interaction shaping the abundance of a few others. However, given the focus of this manuscript on demonstrating drought *selection* on leaf microbiomes, I was surprised that the authors did not actually test whether the drought treatment altered the selection gradient on microbiome components. Is this just because they did not measure plant fitness? It seems all the other components are in place to allow such an analysis, which would actually provide direct evidence that drought exerts selection on leaf microbiomes.

2. The revised manuscript still does not acknowledge the issue of spatial autocorrelation / uncontrolled environmental factors that likely shape both plant genetic structure and microbiome composition, independently; which is a confounding bias that potentially inflated the estimate of % microbiome variation explained by host genotype. The new common garden experiment solidifies the evidence for host genetic control of leaf microbiome composition, which helps to overcome this limitation of the continent-scale study. However, the issue still exists and it would be appropriate for a caveat to be added in the discussion or

60results sections.

3. The new field experiment increases the novelty of this work somewhat, by demonstrating a systematic difference in the abundance of ~8 ASVs between drought-adapted and non-adapted genotypes. It is unclear how much of the total community these ASVs represented, or what their significance may be (other than also being found in the Europe dataset).

Minor comments:

– You state that a total of 386 plants were sampled for 16S rRNA gene sequencing, but later you state that your dataset “began with 939 samples”. How many samples were taken from each plant, and how were those non-independent samples (i.e. multiple observations from the same plant) treated during statistical analysis?

– “Twenty of these phylotypes were found at frequencies that were sufficiently high to enable us to determine the relative influences of genetics and drought treatment on their relative abundances (Table S2).” This process is not described in the Methods. What was the threshold for “sufficiently high” frequency that identified these 20 focal ASVs? The citation of Table S2 might be incorrect, I do not see how it is relevant to this statement.

–The Methods include insufficient details about the statistical analysis of the amplicon sequencing data from the new field experiment.

– It looks like Figure 3 was modified (split into more panels) but references to those panels were not updated - please double check the caption and text to make sure the correct panels are being cited.

Author Rebuttal, third revision:

Our responses in red.

Reviewer #1 (Remarks to the Author):

61I appreciate the authors' comments to my previous assessment and have now read with interest the description of the new common garden experiment. Although the experimental design seems very well suited to fully address my remaining concerns, I was puzzled by the way these results are presented. What I understand from the text is that 3 out of the 575 previously identified bacterial phylotypes show a significant interaction between drought and host genotype. **How abundant are these three phylotypes? More importantly, what percentage of variation do these factors and their interaction explain in this dataset?** The associated supplementary table (S4), which is merely a list of bacterial ASV IDs and p-values, does not provide more information. Perhaps I am missing something here, but in my opinion, the observation that 3 ASVs show a significant interaction in the common garden experiment is not sufficient to support the main claim of the manuscript.

Only 20 of the 575 phylotypes we observed in the European collections were found at appreciable frequencies (in at least 10% of the plants) in the California field experiment. Of these 20 phylotypes, four showed a significant association with either plant genotype or the interaction term between genotype and treatment (20% of the measured ASVs). We have added an additional supplementary figure (Figure S12) and the following text to clarify this point.

“The phylotypes that were significantly associated with plant genotype in the California field experiment composed an appreciable fraction of the total microbiome in the European wild collections — an average of 13.2% of the total microbial community in a plant, and as high as 71.9% total relative abundance in a plant (Figure S12). The most abundant phylotype across the European collection (Figure S12) was significantly associated with plant genotypic classification. In total, these results indicate that genetic adaptation to drought has an impact on some of the most abundant bacteria that colonize a plant.”

I am afraid that the new data provided in this revised version and the adjustments made by the authors are not sufficient to fundamentally change my previous assessment. I believe this manuscript constitutes an impressive effort and makes a valuable contribution but I remain unconvinced about the main claim that drought selection has significantly shaped leaf microbiota composition.

In response to this criticism we have changed the title to “Continental-scale associations of

62Arabidopsis thaliana phyllosphere members with host genotype and drought”.Reviewer #2 (Remarks to the Author):

I am satisfied with the manuscript. Current plant microbiota research field lacks big scale survey with high scientific standard (like high level studies in human gut microbiota), which is very important for demonstrating microbiota patterns in natural environments, and for illustrating the importance to further investigate molecular mechanisms. This work is an excellent example for such big scale survey. On the other side, I would suggest the authors to adjust or weaken some causal conclusions if no experimental data under controlled conditions are provided. Taken together, I think, the revised manuscript would significantly contribute to the plant microbiota research field.

As described above, we have adjusted the title to “Continental-scale associations of *Arabidopsis thaliana* phyllosphere members with host genotype and drought”. We believe this title is more agnostic to the role of selection in the observed patterns.

Reviewer #3 (Remarks to the Author):

I have once again carefully read the revised manuscript and the rebuttal to my previous comments on the first and second versions of this manuscript.

I appreciate the addition of data from a common-garden field experiment and the re-structuring of the text. I agree that the new flow of topics is logical and easier to follow.

1. The new field experiment demonstrates a direct effect of water availability on a few of the 20 ASVs that were tested, and evidence for a genotype-drought interaction shaping the abundance of a few others. However, given the focus of this manuscript on demonstrating drought *selection* on leaf microbiomes, I was surprised that the authors did not actually test whether the drought treatment altered the selection gradient on microbiome components. Is this just because they did not measure

64plant fitness? It seems all the other components are in place to allow such an analysis, which would actually provide direct evidence that drought exerts selection on leaf microbiomes.

The reviewer is correct that we are unable to quantify changes in the selection gradient in the current field experiment given that this current experiment did not quantify the plant fitness.

2. The revised manuscript still does not acknowledge the issue of spatial autocorrelation / uncontrolled environmental factors that likely shape both plant genetic structure and microbiome composition, independently; which is a confounding bias that potentially

inflated the estimate of % microbiome variation explained by host genotype. The new common garden experiment solidifies the evidence for host genetic control of leaf microbiome composition, which helps to overcome this limitation of the continent-scale study. However, the issue still exists and it would be appropriate for a caveat to be added in the discussion or results sections.

We have added the text in bold in the Discussion to acknowledge the potential role for spatial autocorrelation in structuring the microbiome composition:

“Our field experiment begins to disentangle the direct contribution of geography-dependent climate differences on the microbiome from those that are mediated by adaptive differences in host genetics.

We note, however, that both genetic population structure and environmental variables exhibit autocorrelation, hence the variance explained by plant genotype is invariably confounded by correlated environmental factors, with the exact extent being difficult to discern.”

3. The new field experiment increases the novelty of this work somewhat, by demonstrating a systematic difference in the abundance of ~8 ASVs between drought-adapted and non-adapted genotypes. It is unclear how much of the total community these ASVs represented, or what their significance may be (other than also being found in the Europe dataset).

To provide context for the potential functional impact of the phylotypes we identified that are associated with host genetic adaptation to drought, we have added Figure S12 and the following corresponding text: “The phylotypes that were significantly associated with plant genotype in the California field experiment accounted for an appreciable fraction of the total microbiome in the European wild collections — an average of 13.2% of the total microbial community in a plant, and as high as 71.9% total relative abundance in a plant (Figure S12). The most abundant phylotype across the European collection (Figure S12) was significantly associated with plant genotypic classification. In total, these results indicate that genetic adaptation to drought has an impact on some of the most abundant bacteria that colonize a plant.”

Minor comments:

– You state that a total of 386 plants were sampled for 16S rRNA gene sequencing, but later you state that your dataset “began with 939 samples”. How many samples were taken from each plant, and how were those non-independent samples (i.e. multiple observations from the same plant) treated during statistical analysis?

Our comparisons of the abundance of ASVs in *A. thaliana* was limited to 386 plants (one pooled sample of two leaves per plant). The remaining 553 samples included soil samples from each location and samples from neighboring non-*A. thaliana* plants. We have included the following text in the manuscript to clarify:

“We began with 939 samples (including soil samples and neighboring non-*A. thaliana* plants), in which we found 195,545 ASVs.”

– “Twenty of these phylotypes were found at frequencies that were sufficiently high to enable us to determine the relative influences of genetics and drought treatment on their relative abundances (Table S2).” This process is not described in the Methods. What was the threshold for “sufficiently high” frequency that identified these 20 focal ASVs? The citation of Table S2 might be incorrect, I do not see how it is relevant to this statement.

–The Methods include insufficient details about the statistical analysis of the amplicon sequencing data from the new field experiment.

Thank you for pointing out these two accidental omissions. We have now added the following text to the “16S rDNA ASV identification” section of the Materials and Methods:

“For the Californian field experiment, we sequenced the 16S rDNA amplicons as above and processed ASVs with the same pipeline used for the European wild samples. In the Californian ASV table, we identified ASVs present in 10% or more of the samples, and merged these ASV identifies with those of the European collections to identify the intersection of observed ASVs.”

– It looks like Figure 3 was modified (split into more panels) but references to those panels were not updated - please double check the caption and text to make sure the correct panels are being cited.

Corrected – thank you!

Decision Letter, third revision:

Message: Our ref: NMICROBIOL-22041005D

3rd May 2024

Dear Dr. Weigel,

Thank you for your patience as we've prepared the guidelines for final submission of your Nature Microbiology manuscript, "Continental-scale associations of *Arabidopsis thaliana* phyllosphere members with host genotype and drought" (NMICROBIOL-22041005D). Please carefully follow the step-by-step instructions provided in the attached file, and add a response in each row of the table to indicate the changes that you have made. Please also check and comment on any additional marked-up edits we have proposed within the text. Ensuring that each point is addressed will help to ensure that your revised manuscript can be swiftly handed over to our production team.

In recognition of the time and expertise our reviewers provide to Nature Microbiology's editorial process, we would like to formally acknowledge their contribution to the external peer review of your manuscript entitled "Continental-scale associations of *Arabidopsis thaliana* phyllosphere members with host genotype and drought". For those reviewers who give their assent, we will be publishing their names alongside the published article.

Nature Microbiology offers a Transparent Peer Review option for new original research manuscripts submitted after December 1st, 2019. As part of this initiative, we encourage our authors to support increased transparency into the peer review process by agreeing to have the reviewer comments, author rebuttal letters, and editorial decision letters published as a Supplementary item. When you submit your final files please clearly state in your cover letter whether or not you would like to participate in this initiative. Please note

2that failure to state your preference will result in delays in accepting your manuscript for publication.

Cover suggestions

COVER ARTWORK: We welcome submissions of artwork for consideration for our cover. For more information, please see our guide for cover artwork.

Nature Microbiology has now transitioned to a unified Rights Collection system which will allow our Author Services team to quickly and easily collect the rights and permissions required to publish your work. Approximately 10 days after your paper is formally accepted, you will receive an email in providing you with a link to complete the grant of rights. If your paper is eligible for Open Access, our Author Services team will also be in touch regarding any additional information that may be required to arrange payment for your article.

Please note that *Nature Microbiology* is a Transformative Journal (TJ). Authors may publish their research with us through the traditional subscription access route or make their paper immediately open access through payment of an article-processing charge (APC). Authors will not be required to make a final decision about access to their article until it has been accepted. Find out more about Transformative Journals

Authors may need to take specific actions to achieve compliance with funder and institutional open access mandates. If your research is supported by a funder that requires immediate open access (e.g. according to Plan S principles) then you should select the gold OA route, and we will direct you to the compliant route where possible. For authors selecting the subscription publication route, the journal's standard licensing terms will need to be accepted, including self-archiving policies. Those licensing terms will supersede any other terms that the author or any third party may assert to any version of the manuscript.

Reviewer #1:

3Remarks to the Author:

I believe the authors have addressed my concerns to the extent that this was possible, particularly in toning down the stronger claims of earlier versions and in the re-structuring of the text. The revised manuscript constitutes a novel contribution and valuable resource to the plant microbiome research field.

Reviewer #3:

Remarks to the Author:

I am satisfied with this version of the manuscript. The authors addressed my concerns and the new title is a much more appropriate fit to the content. I have no further requests for improvement.

Final Decision Letter:

Message: 2nd July 2024

Dear Detlef,

I am pleased to accept your Resource "Continental-scale associations of *Arabidopsis thaliana* phyllosphere members with host genotype and drought" for publication in Nature Microbiology. Thank you for having chosen to submit your work to us and many congratulations.

You may wish to make your media relations office aware of your accepted publication, in case they consider it appropriate to organize some internal or external publicity. Once your paper has been scheduled you will receive an email confirming the publication details. This is normally 3-4 working days in advance of publication. If you need additional notice of the date and time of publication, please let the production team know when you receive the proof of your article to ensure there is sufficient time to coordinate. Further information on our embargo policies can be found here:

<https://www.nature.com/authors/policies/embargo.html>

After the grant of rights is completed, you will receive a link to your electronic proof via

4email with a request to make any corrections within 48 hours. If, when you receive your proof, you cannot meet this deadline, please inform us at rjsproduction@springernature.com immediately. You will not receive your proofs until the publishing agreement has been received through our system

Please note that *Nature Microbiology* is a Transformative Journal (TJ). Authors may publish their research with us through the traditional subscription access route or make their paper immediately open access through payment of an article-processing charge (APC). Authors will not be required to make a final decision about access to their article until it has been accepted. Find out more about Transformative Journals

nature portfolio

With kind regards,